# Learning Probabilistic Symmetrization for Architecture Agnostic Equivariance

**Jinwoo Kim**    **Tien Dat Nguyen**    **Ayhan Suleymanzade**
**Hyeokjun An**    **Seunghoon Hong**
KAIST

## Abstract

We present a novel framework to overcome the limitations of equivariant architectures in learning functions with group symmetries. In contrary to equivariant architectures, we use an arbitrary base model such as an MLP or a transformer and symmetrize it to be equivariant to the given group by employing a small equivariant network that parameterizes the probabilistic distribution underlying the symmetrization. The distribution is end-to-end trained with the base model which can maximize performance while reducing sample complexity of symmetrization. We show that this approach ensures not only equivariance to given group but also universal approximation capability in expectation. We implement our method on various base models, including patch-based transformers that can be initialized from pretrained vision transformers, and test them for a wide range of symmetry groups including permutation and Euclidean groups and their combinations. Empirical tests show competitive results against tailored equivariant architectures, suggesting the potential for learning equivariant functions for diverse groups using a non-equivariant universal base architecture. We further show evidence of enhanced learning in symmetric modalities, like graphs, when pretrained from non-symmetric modalities, like vision. Code is available at `https://github.com/jw9730/lps`.

## 1   Introduction

Many perception problems in machine learning involve functions that are invariant or equivariant to certain symmetry group of transformations of data. Examples include learning on sets and graphs, point clouds, molecules, proteins, and physical data, to name a few [10, 12]. Equivariant architecture design has emerged as a successful approach, where every building block of a model is carefully restricted to be equivariant to a symmetry group of interest [5, 12, 23]. However, equivariant architecture design faces fundamental limitations, as individual construction of models for each group can be laborious or computationally expensive [91, 58, 64, 45], the architectural restrictions often lead to limited expressive power [101, 56, 105, 40], and the knowledge learned from one problem cannot be easily transferred to others of different symmetries as the architecture would be incompatible.

This motivates us to seek a **symmetrization** solution that can achieve group invariance and equivariance with general-purpose, group-agnostic architectures such as an MLP or a transformer. As a basic form of symmetrization, any parameterized function $f_\theta : \mathcal{X} \to \mathcal{Y}$ on vector spaces $\mathcal{X}, \mathcal{Y}$ can be made invariant or equivariant by group averaging [102, 67], *i.e.*, averaging over all possible transformations of inputs $\mathbf{x} \in \mathcal{X}$ and outputs $\mathbf{y} \in \mathcal{Y}$ by a symmetry group $G = \{g\}$:

$$\phi_\theta(\mathbf{x}) = \frac{1}{|G|} \sum_{g \in G} g \cdot f_\theta(g^{-1} \cdot \mathbf{x}), \tag{1}$$

where $\phi_\theta$ is equivariant or invariant to the group $G$. An important advantage is that the symmetrized function $\phi_\theta$ can leverage the expressive power of the base function $f_\theta$; it has been shown that $\phi_\theta$ is a universal approximator of invariant or equivariant functions if $f_\theta$ is a universal approximator [102],

37th Conference on Neural Information Processing Systems (NeurIPS 2023).

# Probabilistic Symmetrization

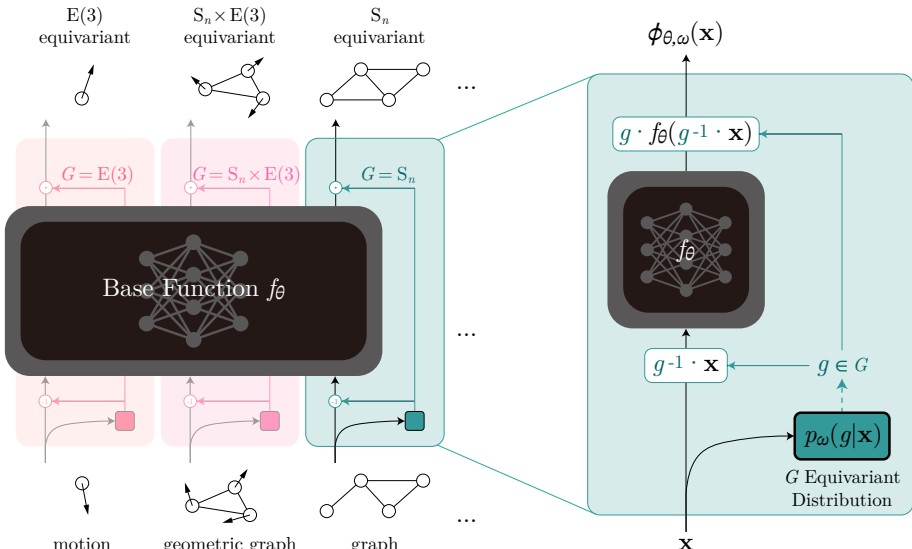

Figure 1: Overview of probabilistic symmetrization. We symmetrize an unconstrained base function $f_\theta$ into an equivariant function $\phi_{\theta,\omega}$ for group $G$ using a learned equivariant distribution $p_\omega(g|\mathbf{x})$.

which includes an MLP [35] or a transformer [104]. On the other hand, an immediate challenge is that for many practical groups involving permutation and rotations, the cardinality of the group $|G|$ is large or infinite, so the exact averaging is intractable. Due to this, existing symmetrization approaches often focus on small finite groups [4, 65, 94, 42], manually derive smaller subsets of the entire group *e.g.*, a frame [74] to average over [67, 68], or implement a relaxed version of equivariance [41, 83].

An alternative method for tractability is to interpret Eq. (1) as an expectation with uniform distribution $\mathrm{Unif}(G)$ over the compact group $G$ [62], and use sampling-based average to estimate it [102, 67, 68]:

$$\phi_\theta(\mathbf{x}) = \mathbb{E}_{g\sim\mathrm{Unif}(G)}\left[g\cdot f_\theta(g^{-1}\cdot\mathbf{x})\right], \tag{2}$$

where $g\cdot f_\theta(g^{-1}\cdot\mathbf{x})$ serves as an unbiased estimator of $\phi_\theta(\mathbf{x})$. While simple and general, this approach has practical issues that the base function $f_\theta$ is burdened to learn all equally possible group transformations, and the expectedly high variance of the estimator can lead to challenges in sampling-based training due to large variance of gradients as well as sample complexity of inference.

Our key idea is to replace the uniform distribution $\mathrm{Unif}(G)$ for the expectation in Eq. (2) with a parameterized distribution $p_\omega(g|\mathbf{x})$ in a way that equivariance and expressive power are always guaranteed, and train it end-to-end with the base function $f_\theta$ to directly minimize task loss. We show that the distribution $p_\omega(g|\mathbf{x})$ only needs to satisfy one simple condition to guarantee equivariance and expressive power: it has to be probabilistically equivariant [9]. This allows us to generally implement $p_\omega(g|\mathbf{x})$ as a noise-outsourced map $(\mathbf{x}, \boldsymbol{\epsilon}) \mapsto g$ with an invariant noise $\boldsymbol{\epsilon}$ and a small equivariant network $q_\omega$, which enables gradient-based training with reparameterization [47]. As $p_\omega$ is trained, it can enhance the learning of $f_\theta$ by producing group transformations of lower variance compared to $\mathrm{Unif}(G)$ so that $f_\theta$ is less burdened, and coordinating with $f_\theta$ to maximize task performance. We refer to our approach as probabilistic symmetrization. An overview is provided in Figure 1.

We implement and test our method with two general-purpose architectures as the base function $f_\theta$: MLP and transformer. In particular, our transformer backbone is architecturally identical to patch-based vision transformers [26], which allows us to initialize most of its parameters from ImageNet-21k pretrained weights [89] and only replace the input and output projections to match task dimensions. We implement the conditional distribution $p_\omega(g|\mathbf{x})$ for a wide range of practical symmetry groups including permutation ($\mathrm{S}_n$) and Euclidean groups ($\mathrm{O/SO}(d)$ and $\mathrm{E/SE}(d)$) and their product combinations (*e.g.*, $\mathrm{S}_n \times \mathrm{O}(3)$), all of which are combinatorial or infinite groups. Empirical tests on a wide range of invariant and equivariant tasks involving graphs and motion data show competitive results against tailored equivariant architectures as well as existing symmetrization methods [74, 41]. This suggests the potential for learning invariant or equivariant functions for diverse groups with a group-agnostic general-purpose backbone. We further show evidence that pretraining from non-symmetric modality (vision) leads to enhanced learning in symmetric modality (graphs).

# 2 Probabilistic Symmetrization for Equivariance

We introduce and analyze our approach called probabilistic symmetrization, which involves an equivariant distribution $p_\omega$ and group-agnostic base function $f_\theta$, in Section 2.1. We then describe implementation of $p_\omega$ for practical groups including permutations and rotations in Section 2.2. Then, we describe our choice of base function $f_\theta$ focusing on MLP and, transformers in particular, in Section 2.3. All proofs can be found in Appendix A.1.

**Problem Setup**  In general, our goal is to construct a function $\phi : \mathcal{X} \to \mathcal{Y}$ on finite vector spaces $\mathcal{X}, \mathcal{Y}$ that is invariant or equivariant to symmetry specified by a group $G = \{g\}$[1]. This is formally described by specifying how the group act as transformations on the input and output. A group representation $\rho : G \to \mathrm{GL}(\mathcal{X})$, where $\mathrm{GL}(\mathcal{X})$ is the set of all invertible matrices on $\mathcal{X}$, associates each group element $g \in G$ to an invertible matrix $\rho(g)$ that transforms a given vector $\mathbf{x} \in \mathcal{X}$ through $\mathbf{x} \mapsto g \cdot \mathbf{x} = \rho(g)\mathbf{x}$. Given that, a function $\phi : \mathcal{X} \to \mathcal{Y}$ is $G$ equivariant if:

$$\phi(\rho_1(g)\mathbf{x}) = \rho_2(g)\phi(\mathbf{x}), \quad \forall \mathbf{x} \in \mathcal{X}, g \in G, \tag{3}$$

where the representations $\rho_1$ and $\rho_2$ are on the input and output, respectively. $G$ invariance is a special case of equivariance when the output representation is trivial, $\rho_2(g) = \mathbf{I}$.

## 2.1 Probabilistic Symmetrization

To construct a $G$ equivariant function $\phi_\theta$, group averaging symmetrizes an arbitrary base function $f_\theta : \mathcal{X} \to \mathcal{Y}$ by taking expectation with uniform distribution over the group (Eq. (2)). Instead, we propose to use an input-conditional parameterized distribution $p_\omega(g|\mathbf{x})$ and symmetrize $f_\theta$ as follows:

$$\phi_{\theta,\omega}(\mathbf{x}) = \mathbb{E}_{p_\omega(g|\mathbf{x})} \left[ \rho_2(g) f_\theta(\rho_1(g)^{-1}\mathbf{x}) \right], \tag{4}$$

where the distribution $p_\omega(g|\mathbf{x})$ itself satisfies probabilistic $G$ equivariance:

$$p_\omega(g|\mathbf{x}) = p_\omega(g'g|\rho_1(g')\mathbf{x}), \quad \forall \mathbf{x} \in \mathcal{X}, g, g' \in G. \tag{5}$$

Importantly, we show that probabilistic symmetrization with equivariant $p_\omega$ guarantees equivariance as well as expressive power of the symmetrized $\phi_{\theta,\omega}$.

**Theorem 1.** *If $p_\omega$ is $G$ equivariant, then $\phi_{\theta,\omega}$ is $G$ equivariant for arbitrary $f_\theta$.*

*Proof.* The proof can be found in Appendix A.1.1. □

**Theorem 2.** *If $p_\omega$ is $G$ equivariant and $f_\theta$ is a universal approximator, then $\phi_{\theta,\omega}$ is a universal approximator of $G$ equivariant functions.*

*Proof.* The proof can be found in Appendix A.1.2. □

While the base function $f_\theta$ that guarantees universal approximation can be chosen in a group-agnostic manner *e.g.*, an MLP [35, 20] or a transformer on token sequences [104], the distribution $p_\omega$ needs to be instantiated group specifically to satisfy $G$ equivariance. A simplistic choice is using uniform distribution $\mathrm{Unif}(G)$ for all inputs $\mathbf{x}$ with no parameterization (reducing to group averaging), which is technically equivariant and therefore guarantees equivariance and universality. However, appropriately parameterizing and learning $p_\omega$ can provide distinguished advantages compared to the uniform distribution, as it can **(1)** learn from data to collaborate with (pre-trained) base function $f_\theta$ to maximize task performance and **(2)** learn to produce more consistent samples $g \sim p_\omega(g|\mathbf{x})$ that can offer more stable gradients for the base function $f_\theta$ during early training.

We now provide a generic blueprint of $G$ equivariant distribution $p_\omega(g|\mathbf{x})$ for any compact group $G$. Our goal is to sample $g \sim p_\omega(g|\mathbf{x})$ to obtain group representation $\rho(g)$ for symmetrization (Eq. (4)) in a differentiable manner so that $p_\omega$ can be trained end-to-end. Since we only need sampling and there is no need to evaluate likelihoods, we simply implement $p_\omega(g|\mathbf{x})$ as a noise-outsourced, differentiable transformation $q_\omega(\mathbf{x}, \boldsymbol{\epsilon})$ of a noise variable $\boldsymbol{\epsilon} \in \mathcal{E}$ that directly outputs a group representation $\rho(g)$:

$$\rho(g) = q_\omega(\mathbf{x}, \boldsymbol{\epsilon}), \quad \boldsymbol{\epsilon} \sim p(\boldsymbol{\epsilon}), \tag{6}$$

---

[1]In this paper, we assume the group $G$ to be compact.

where $q_\omega$ is $G$ equivariant and $p(\boldsymbol{\epsilon})$ is $G$ invariant under an appropriate representation $\rho'$:

$$q_\omega(\rho_1(g)\mathbf{x}, \rho'(g)\boldsymbol{\epsilon}) = \rho(g)q_\omega(\mathbf{x}, \boldsymbol{\epsilon}), \quad p(\boldsymbol{\epsilon}) = p(\rho'(g)\boldsymbol{\epsilon}), \quad \forall \mathbf{x} \in \mathcal{X}, \boldsymbol{\epsilon} \in \mathcal{E}, g \in G. \quad (7)$$

Given above implementation, we can show the $G$ equivariance of $p_\omega$:

**Theorem 3.** *If $q_\omega$ is $G$ equivariant and $p(\boldsymbol{\epsilon})$ is $G$ invariant under representation $\rho'$ that $|\det \rho'(g)| = 1 \forall g \in G$, the distribution $p_\omega(g|\mathbf{x})$ characterized by $q_\omega : (\mathbf{x}, \boldsymbol{\epsilon}) \mapsto \rho(g)$ is $G$ equivariant.*

*Proof.* The proof can be found in Appendix A.1.3. $\qquad\qquad\qquad\qquad\qquad\qquad\qquad\quad$ $\square$

In practice, one can use any available $G$ equivariant neural network to implement $q_\omega$, *e.g.*, a graph neural network for the symmetric group $\mathrm{S}_n$, or an equivariant MLP which can be constructed for any matrix group [30]. Since we expect most of the reasoning to be done by the base function $f_\theta$, the equivariant network $q_\omega$ can be small and relatively less expressive. This allows us to get less affected by their known issues in expressiveness and scaling [101, 70, 14, 40]. For the noise $\boldsymbol{\epsilon} \sim p(\boldsymbol{\epsilon})$, simple choices often suffice for $G$ invariance. For example, standard normal $\boldsymbol{\epsilon} \sim \mathcal{N}(0, \mathbf{I}_n)$ provides invariance for the symmetric group $\mathrm{S}_n$ as well as the (special) orthogonal group $\mathrm{O}(n)$ and $\mathrm{SO}(n)$.

One important detail in designing $q_\omega$ is constraining its output to be a valid group representation $\rho(g)$. For this, we apply appropriate postprocessing to refine neural network features into group representations, *e.g.*, Gram-Schmidt orthogonalization to obtain a representation $\rho(g) \in \mathbb{R}^{n \times n}$ of the orthogonal group $g \in \mathrm{O}(n)$. Importantly, to not break the $G$ equivariance of $q_\omega$, this postprocessing needs to be equivariant itself, *e.g.*, Gram-Schmidt process is itself $\mathrm{O}(n)$ equivariant [41]. Implementations of $p_\omega$ for a range of practical symmetry groups are provided in detail in Section 2.2.

## 2.2 Equivariant Distribution $p_\omega$

We present implementations of the $G$ equivariant distribution $p_\omega(g|\mathbf{x})$ for a range of practical groups demonstrated in our experiments (Section 3). Formal proofs of correctness are in Appendix A.1.4.

**Symmetric Group** $\mathrm{S}_n$    The symmetric group $\mathrm{S}_n$ over a finite set of $n$ elements contains all permutations of the set, which describes symmetry to ordering desired for learning set and graph data. The base representation is given by $\rho(g) = \mathbf{P}_g$ where $\mathbf{P}_g \in \{0, 1\}^{n \times n}$ is a permutation matrix for $g$.

To implement $\mathrm{S}_n$ equivariant distribution $p_\omega(g|\mathbf{x})$ that provides permutation matrices $\mathbf{P}_g$ from graph data $\mathbf{x}$, we use the following design. We first sample invariant noise $\boldsymbol{\epsilon} \in \mathbb{R}^{n \times d}$ from i.i.d. uniform $\mathrm{Unif}[0, \eta]$ with noise scale $\eta$. For the $\mathrm{S}_n$ equivariant map $q_\omega : (\mathbf{x}, \boldsymbol{\epsilon}) \mapsto \rho(g)$, we first use a graph neural network (GNN) as an equivariant map $(\mathbf{x}, \boldsymbol{\epsilon}) \mapsto \mathbf{Z}$ that outputs nodewise scalar $\mathbf{Z} \in \mathbb{R}^n$. Then, assuming $\mathbf{Z}$ is tie-free[2], we use below argsort operator $\mathbf{Z} \mapsto \mathbf{P}_g$ to obtain permutation matrix [73, 98]:

$$\mathbf{P}_g = \mathrm{eq}(\mathbf{Z}\mathbf{1}^\top, \mathbf{1}\mathrm{sort}(\mathbf{Z})^\top), \quad (8)$$

where eq denotes elementwise equality indicator. The argsort operator is $\mathrm{S}_n$ equivariant, *i.e.*, it maps $\mathbf{P}_{g'}\mathbf{Z} \mapsto \mathbf{P}_{g'}\mathbf{P}_g$ for all $\mathbf{P}_{g'} \in \mathrm{S}_n$. To backpropagate through $\mathbf{P}_g$ during training, we use straight-through gradient estimator [6] with an approximate permutation matrix $\hat{\mathbf{P}}_g \approx \mathbf{P}_g$ obtained from a differentiable relaxation of argsort operator [61, 33, 98]; details can be found in Appendix A.3.1.

**Orthogonal Group** $\mathrm{O}(n)$**,** $\mathrm{SO}(n)$    The orthogonal group $\mathrm{O}(n)$ contains all roto-reflections in $\mathbb{R}^n$ around origin, and the special orthogonal group $\mathrm{SO}(n)$ contains all rotations without reflections. These groups describe rotation symmetries desirable in learning geometric data. The base group representation for $\mathrm{O}(n)$ is given by $\rho(g) = \mathbf{Q}_g$ where $\mathbf{Q}_g$ is the orthogonal matrix for $g$. For $\mathrm{SO}(n)$, the representation is $\rho(g) = \mathbf{Q}_g^+$ where $\mathbf{Q}_g^+$ is the orthogonal matrix of $g$ with determinant $+1$.

To implement $\mathrm{O}(n)/\mathrm{SO}(n)$ equivariant distribution $p_\omega(g|\mathbf{x})$ that provides orthogonal matrices given input data $\mathbf{x}$, we use the following design. We first sample invariant noise $\boldsymbol{\epsilon} \in \mathbb{R}^{n \times d}$ from i.i.d. normal $\mathcal{N}(0, \eta^2)$ with noise scale $\eta$. For the $\mathrm{O}(n)/\mathrm{SO}(n)$ equivariant map $p_\omega : (\mathbf{x}, \boldsymbol{\epsilon}) \mapsto \rho(g)$, we first use an equivariant neural network as a map $(\mathbf{x}, \boldsymbol{\epsilon}) \mapsto \mathbf{Z}$ that outputs $n$ features $\mathbf{Z} \in \mathbb{R}^{n \times n}$. Then, assuming $\mathbf{Z}$ is full-rank[3], we use Gram-Schmidt process for orthogonalization, which is differentiable

---

[2]This can be assumed since the invariant noise $\boldsymbol{\epsilon} \in \mathbb{R}^{n \times d}$ serves as tiebreaker between the $n$ nodes.

[3]In practice, we add a random Gaussian matrix of a tiny magnitude to $\mathbf{Z}$ to prevent rank collapse.

and O$(n)$ equivariant [41]. This completes the postprocessing $\mathbf{Z} \mapsto \mathbf{Q}_g$ for O$(n)$. For SO$(n)$, we can further use a simple scale operator $\mathbf{Q} \mapsto \mathbf{Q}_g^+$ to set the determinant of orthogonalized matrix to $+1$:

$$\text{scale} : \left[ \begin{array}{c|c|c} \mathbf{Q}_1 & \cdots & \mathbf{Q}_n \end{array} \right] \mapsto \left[ \begin{array}{c|c|c} \det(\mathbf{Q}) \cdot \mathbf{Q}_1 & \cdots & \mathbf{Q}_n \end{array} \right], \qquad (9)$$

The scale operator is differentiable and SO$(n)$ equivariant, *i.e.*, it maps $\mathbf{Q}_{g'}^+ \mathbf{Q} \mapsto \mathbf{Q}_{g'}^+ \mathbf{Q}_g^+$ for all $\mathbf{Q}_{g'}^+ \in \text{SO}(n)$, thereby completing the postprocessing $\mathbf{Z} \mapsto \mathbf{Q}_g^+$ for SO$(n)$.

**Euclidean Group** E$(n)$**, ** SE$(n)$   The Euclidean group E$(n)$ contains all roto-translations and reflections in $\mathbb{R}^n$ and their combinations, and the special Euclidean group SE$(n)$ contains all roto-translations without reflections. These groups are desired in learning physical systems such as a particle in motion. Formally, the Euclidean group is given as a combination of orthogonal group and translation group E$(n) = $ O$(n) \ltimes$ T$(n)$, and similarly SE$(n) = $ SO$(n) \ltimes$ T$(n)$. As the translation group T$(n)$ is non-compact which violates our assumption for symmetrization, we handle it separately. Following prior work [74, 41], we subtract the centroid $\bar{\mathbf{x}}$ from the input data $\mathbf{x}$ as $\mathbf{x} - \bar{\mathbf{x}}$, and add it to the rotation symmetrized output as $\bar{\mathbf{x}} + g \cdot f_\theta(g^{-1} \cdot (\mathbf{x} - \bar{\mathbf{x}}))$ where $g$ is sampled from O$(n)$/SO$(n)$ equivariant $p_\omega(g|\mathbf{x} - \bar{\mathbf{x}})$. This makes the overall symmetrized function E$(n)$/SE$(n)$ equivariant.

**Product Group** $H \times K$   While we have described several groups individually, in practice we often encounter product combinations of groups $G = H \times K$ that describe joint symmetry to each group $H$ and $K$. For example, S$_n \times$ O$(3)$ describes joint symmetry to permutations and rotations, which is desired in learning point clouds, molecules, and particle interactions. In general, an element of $H \times K$ is given as $g = (h, k)$ where $h \in H$ and $k \in K$, and group operations are applied elementwise $gg' = (h, k)(h', k') = (hh', kk')$. The base group representation is accordingly given as pair of representations $\rho(g) = (\rho(h), \rho(k))$. While a common approach to handling $H \times K$ is *partially* symmetrizing on $H$ and imposing $K$ equivariance on the base architecture (*e.g.*, rotational symmetrization of a graph neural network for S$_n \times$ O$(3)$ equivariance [74, 41]), we extend to *full symmetrization* on $H \times K$ since our goal is not imposing any constraint on the base function $f_\theta$.

To implement $H \times K$ equivariant distribution $p_\omega(g|\mathbf{x})$ that gives $\rho(g) = (\rho(h), \rho(k))$ from data $\mathbf{x}$, we use the following design. We first sample invariant noise $\boldsymbol{\epsilon}$ from i.i.d. normal $\mathcal{N}(0, \eta^2)$ with scale $\eta$. For the $H \times K$ equivariant map $q_\omega : (\mathbf{x}, \boldsymbol{\epsilon}) \mapsto (\rho(h), \rho(k))$, we employ a $H \times K$ equivariant neural network as a map $(\mathbf{x}, \boldsymbol{\epsilon}) \mapsto (\mathbf{Z}_H, \mathbf{Z}_K)$ such that the postprocessing for each group $H$ and $K$ provides maps $\mathbf{Z}_H \mapsto \rho(h)$ and $\mathbf{Z}_K \mapsto \rho(k)$ respectively, leading to full representation $\rho(g) = (\rho(h), \rho(k))$. For this whole procedure to be $H \times K$ equivariant, it is sufficient to have $\mathbf{Z}_H$ be $K$ invariant and $\mathbf{Z}_K$ be $H$ invariant. These are special cases of $H \times K$ equivariance, and is supported by a range of equivariant neural networks especially regarding S$_n \times$ O$(3)$ or S$_n \times$ SO$(3)$ equivariances [23].

### 2.3   Base Function $f_\theta$

We now describe the choice of group-agnostic base function $f_\theta : \mathbf{x} \mapsto \mathbf{y}$. As group symmetry is handled by the equivariant distribution $p_\omega(g|\mathbf{x})$, any symmetry concern is *hidden* from $f_\theta$, allowing the inputs $\mathbf{x}$ and outputs $\mathbf{y}$ to be treated as plain multidimensional arrays. This allows us to implement $f_\theta$ with powerful general-purpose architectures, namely as an MLP that operates on flattened vectors of inputs and outputs, or a transformer as we describe below.

Let inputs $\mathbf{x} \in \mathcal{X} = \mathbb{R}^{n_1 \times \ldots \times n_a \times c}$ and outputs $\mathbf{y} \in \mathcal{Y} = \mathbb{R}^{n_1' \times \ldots \times n_b' \times c'}$ be multidimensional arrays with $c$ and $c'$ channels, respectively. Our transformer base function $f_\theta : \mathbf{x} \mapsto \mathbf{y}$ is given as:

$$f_\theta = \text{detokenize} \circ \text{transformer} \circ \text{tokenize}, \qquad (10)$$

where tokenize $: \mathcal{X} \to \mathbb{R}^{m \times d}$ parses input array to a sequence of $m$ tokens, transformer $: \mathbb{R}^{m \times d} \to \mathbb{R}^{m \times d}$ is a standard transformer encoder on tokens used in language and vision [24, 26], and detokenize $: \mathbb{R}^{m \times d} \to \mathcal{Y}$ decodes encoded tokens to output array. For the tokenizer and detokenizer, we can use linear projections on flattened chunks of the array, which directly extends flattened patch projections in vision transformers to higher dimensions [26, 85, 86]. This enables mapping between different dimensional inputs and outputs, *e.g*, for graph node classification with $\mathbf{x} \in \mathbb{R}^{n \times n \times c}$ and $\mathbf{y} \in \mathbb{R}^{n \times c'}$, it is possible to use 2D patch projection for the input and 1D projection for the output.

Above choice of $f_\theta$ offers important advantages including universal approximation [104] and ability to share and transfer the learned knowledge in $\theta$ over different domains of different group symmetries. Remarkably, this allows us to directly leverage large-scale pre-trained parameters from data-abundant domains for learning on symmetric domains. In our experiments, we only replace the tokenizer and detokenizer of a vision transformer pre-trained on ImageNet-21k [99, 26] and fine-tune it to perform diverse $S_n$ equivariant tasks such as graph classification, node classification, and link prediction.

## 2.4 Relation to Other Symmetrization Approaches

We discuss relation of our symmetrization method to prior ones, specifically group averaging [102, 4], frame averaging [74], and canonicalization [41]. An extended discussion on broader related work can be found in Appendix A.2. Since these symmetrization methods share common formalization $\mathbf{y} = \mathbb{E}_g[g \cdot f_\theta(g^{-1} \cdot \mathbf{x})]$, one can expect a close theoretical relationship between them. We observe that probabilistic symmetrization is quite general; based on particular choices of the $G$ equivariant distribution $p_\omega(g|\mathbf{x})$, it can become most of the related symmetrization methods as special cases. This can be easily seen for group averaging [102], as the distribution $p_\omega$ can reduce to the uniform distribution $\mathrm{Unif}(G)$ over the group. Frame averaging [74] also takes an average, but over a subset of group given by a frame $F : \mathcal{X} \to 2^G \setminus \emptyset$; importantly, it is required that the frame itself is $G$ equivariant $F(\rho(g)\mathbf{x}) = gF(\mathbf{x})$. We can make the following connection between our method and frame averaging, by adopting the concept of stabilizer subgroup $G_\mathbf{x} = \{g \in G : \rho(g)\mathbf{x} = \mathbf{x}\}$:

**Proposition 1.** *Probabilistic symmetrization with $G$ equivariant distribution $p_\omega(g|\mathbf{x})$ can become frame averaging [74] by assigning uniform density to a set of orbits $G_\mathbf{x}g$ for some group elements g.*

*Proof.* The proof can be found in Appendix A.1.5. □

Canonicalization [41] uses a *single* group element for symmetrization, produced by a trainable canonicalizer $C_\omega : \mathcal{X} \to G$. Here, it is required that the canonicalizer itself satisfies *relaxed $G$ equivariance* $C(\rho(g)\mathbf{x}) = gg'C(\mathbf{x})$ up to arbitrary action from the stabilizer $g' \in G_\mathbf{x}$. We now show:

**Proposition 2.** *Probabilistic symmetrization with $G$ equivariant distribution $p_\omega(g|\mathbf{x})$ can become canonicalization [41] by assigning uniform density to a single orbit $G_\mathbf{x}g$ of some group element g.*

*Proof.* The proof can be found in Appendix A.1.5. □

Assuming that stabilizer $G_\mathbf{x}$ is trivial, this can be implemented with our method by removing random noise $\epsilon$, which reduces $p_\omega$ to deterministic map $\rho(g) = q_\omega(\mathbf{x})$. We use this approach to implement canonicalizer for the $S_n$ group, while [41] only provides canonicalizers for Euclidean groups.

## 3 Experiments

We empirically demonstrate and analyze probabilistic symmetrization on a range of symmetry groups $S_n$, $E(3)$ $(O(3))$, and the product $S_n \times E(3)$ $(S_n \times O(3))$, on a variety of invariant and equivariant tasks, with general-purpose base functions $f_\theta$ chosen as MLP and transformer optionally with pretraining from vision domain. Details of the experiments are in Appendix A.3, and supplementary experiments on other base functions and comparisons to other symmetrization approaches are in Appendix A.4.

### 3.1 Graph Isomorphism Learning with MLP

Building expressive neural networks for graphs ($S_n$) has been considered important and challenging, as simple and efficient GNNs are often limited in expressive power to certain Weisfeiler-Lehman isomorphism tests like 1-WL [101, 56]. Since an MLP equipped with probabilistic symmetrization is in theory universal and $S_n$ equivariant, it has potential for graph learning that require high expressive power. To explicitly test this, we adopt the experimental setup of [74] and use two datasets on graph separation task ($S_n$ invariant). GRAPH8c [2] consists of all non-isomorphic connected graphs with 8 nodes, and EXP [1] consists of 3-WL distinguishable graphs that are not 2-WL distinguishable. We compare our method to standard GNNs as well as an MLP symmetrized with group averaging [102], frame averaging [74], and canonicalization [41]. Our method uses the same MLP architecture to symmetrization baselines, and its $S_n$ equivariant distribution $p_\omega$ for symmetrization is implemented

Table 1: Results for $S_n$ invariant graph separation. We use two tasks, one for counting pairs of graphs not separated by a model at random initialization (GRAPH8c and EXP), and one for learning to classify EXP to two classes (EXP-classify). For EXP-classify, we report the test accuracy at best validation accuracy. The columns arch. and sym. denote architectural and symmetrized equivariance, respectively. The results for baselines are from [74] except for MLP-Canonical. which is tested by us.

| method | arch. | sym. | GRAPH8c ↓ | EXP ↓ | EXP-classify ↑ |
|---|---|---|---|---|---|
| GCN [48] | $S_n$ | - | 4755 | 600 | 50% |
| GAT [97] | $S_n$ | - | 1828 | 600 | 50% |
| GIN [101] | $S_n$ | - | 386 | 600 | 50% |
| ChebNet [22] | $S_n$ | - | 44 | 71 | 82% |
| PPGN [56] | $S_n$ | - | 0 | 0 | **100%** |
| GNNML3 [2] | $S_n$ | - | 0 | 0 | **100%** |
| MLP-GA [102] | - | $S_n$ | 0 | 0 | 50% |
| MLP-FA [74] | - | $S_n$ | 0 | 0 | **100%** |
| MLP-Canonical. | - | $S_n$ | 0 | 0 | 50% |
| MLP-PS (Ours), fixed $\epsilon$ | - | $S_n$ | 0 | 0 | 79.5% |
| MLP-PS (Ours) | - | $S_n$ | 0 | 0 | **100%** |

Figure 2: Learned $p_\omega(g|\mathbf{x})$ over time. The entropy of aggregated permutation matrices $\bar{\mathbf{P}} = \sum \mathbf{P}_g/N$ from $\mathbf{P}_g \sim p_\omega(g|\mathbf{x})$ for each input $\mathbf{x}$ drops in early training, indicating that the distribution learns to produce lower-variance permutations as in below visualizations.

using a 3-layer GIN [101] which is 1-WL expressive. We use 10 samples for symmetrization during both training and testing. Further details can be found in Appendix A.3.3, and supplementary results on symmetrization of a different base model can be found in Appendix A.4.1.

The results are in Table 1. At random initialization, all symmetrization methods can provide perfect separation of all graphs, similar to PPGN [56] and GNNML3 [2] that are equivariant neural networks carefully designed to be 3-WL expressive. However, when trained with gradient descent to solve classification problem, naïve symmetrization with group averaging fails, presumably because the MLP fails to adjust to equally possible 64! permutations of 64 nodes in maximum. On the other hand, our method is able to learn the task, achieving the same accuracy to frame averaging that utilizes costly eigendecomposition of graph Laplacian [74]. What makes our method work while group averaging fails? We conjecture this is since the distribution $p_\omega(g|\mathbf{x})$ can learn to provide more consistent permutations during early training, as we illustrate in Figure 2. In the figure, we measured the consistency of samples from $p_\omega(g|\mathbf{x})$ over training progress by sampling $N = 50$ permutation matrices $\mathbf{P}_g \sim p_\omega(g|\mathbf{x})$ and measuring the row-wise entropy of their average $\bar{\mathbf{P}} = \sum \mathbf{P}_g/N$ for each input $\mathbf{x}$. The more consistent the sampled permutations, the sharper their average, and lesser the entropy. As training progresses, $p_\omega$ learns to produce more consistent samples, which coincides with the initial increase in task performance. Given that, a natural question would be: if we enforce the samples $g \sim p_\omega(g|\mathbf{x})$ to be consistent from the first place, would it work? To answer this, we also tested a non-probabilistic version of our model that uses a single permutation per input $\rho(g) = q_\omega(\mathbf{x})$, which is a canonicalizer under relaxed equivariance [41] as described in Section 2.4. As in Table 1, canonicalization fails, suggesting that probabilistic nature of $p_\omega(g|\mathbf{x})$ can be *beneficial* for learning. For another deterministic version of our model made by fixing the noise $\epsilon$ (Eq. (6)) at initialization, the performance drops to 79.5%, further implying that stochasticity of $p_\omega(g|\mathbf{x})$ has a role.

Table 2: Results for $S_n \times E(3)$ equivariant $n$-body problem. The columns arch. and sym. denote architectural and symmetrized equivariance, respectively. We report test MSE at best validation MSE, along with the standard deviation for GA and Ours where predictions are stochastic. The results for baselines are from [41] except symmetrized transformers which are tested by us.

| method | arch. | sym. | Position MSE ↓ |
|---|---|---|---|
| SE(3) Transformer [31] | $S_n \times SE(3)$ | - | 0.0244 |
| TFN [91] | $S_n \times SE(3)$ | - | 0.0155 |
| Radial Field [49] | $S_n \times E(3)$ | - | 0.0104 |
| EGNN [84] | $S_n \times E(3)$ | - | 0.0071 |
| GNN-FA [74] | $S_n$ | E(3) | 0.0057 |
| GNN-Canonical. [41] | $S_n$ | E(3) | 0.0043 |
| Transformer-Canonical. | - | $S_n \times E(3)$ | 0.00508 |
| Transformer-GA | - | $S_n \times E(3)$ | $0.00414 \pm 0.00001$ |
| Transformer-PS (Ours) | - | $S_n \times E(3)$ | $\mathbf{0.00401 \pm 0.00001}$ |

## 3.2 Particle Dynamics Learning with Transformer

Learning sets or graphs attributed with position and velocity in 3D ($S_n \times E(3)$) is practically significant as they universally appear in physics, chemistry, and biology applications. While prior symmetrization methods employ an already $S_n$ equivariant base function and partially symmetrize the $E(3)$ part, we attempt to symmetrize the entire product group $S_n \times E(3)$ and choose the base model $f_\theta$ as a sequence transformer to leverage its expressive power. For empirical demonstration, we adopt the experimental setup of [41] and use the $n$-body dataset [84, 31] where the task is predicting the position of $n = 5$ charged particles after certain time given their initial position and velocity in $\mathbb{R}^3$ ($S_n \times E(3)$ equivariant). We compare our method to $S_n \times E(3)$ equivariant neural networks and partial symmetrization methods applying $E(3)$ symmetrization to GNNs. We also test prior symmetrization methods on the full group $S_n \times E(3)$ along our method, but could not test for frame averaging since equivariant frames for the full group $S_n \times E(3)$ was not available in current literature. Our method is implemented using a transformer with sequence positional encodings with around $2.3\times$ parameters of the baselines, and the $S_n \times E(3)$ equivariant distribution $p_\omega$ for symmetrization is implemented using a 2-layer Vector Neurons [23] that has around $0.03\times$ of parameters to the transformer. We use 20 samples for symmetrization during training, and use $10\times$ sample size for testing since the task is regression where appropriate variance reduction is necessary to guarantee a reliable performance. Further details can be found in Appendix A.3.4, and supplementary results on $E(3)$ partial symmetrization of GNN base model can be found in Appendix A.4.2.

The results are in Table 2. We observe simple group averaging exhibits a surprisingly strong performance, as it achieves 0.00414 MSE and already outperforms previous state of the art 0.0043 MSE. This is because the permutation component of the symmetry is fairly small, with $n = 5$ particles interacting with each other, such that combining it with an expressive base model $f_\theta$ (a sequence transformer) can adjust to $5! = 120$ equally possible permutations and their rotations in 3D. Nevertheless, our method outperforms group averaging and achieves a new state of the art 0.00401 MSE, presumably as the parameterized distribution $p_\omega$ learns to further maximize task performance. On the other hand, the canonicalization approach, implemented by eliminating noise variable $\epsilon$ from our method (Section 2.4), performs relatively poorly. We empirically observe that $f_\theta$ memorizes the per-input canonical orientations provided by $\rho(g) = q_\omega(\mathbf{x})$ that do not generalize to test inputs. This again shows that probabilistic nature of $p_\omega(g|\mathbf{x})$ can be beneficial for performance.

## 3.3 Graph Pattern Recognition with Vision Transformer

One important goal of our approach, and symmetrization in general, is to *decouple* the symmetry of problem from the base function $f_\theta$, such that we can leverage knowledge learned from other symmetries by transferring the parameters $\theta$. We demonstrate an extreme case by transferring the parameters of a vision transformer [26] trained on large-scale image classification (translation invariant) to solve node classification on graphs ($S_n$ equivariant) for the first time in literature. For this, we use the PATTERN dataset [27] that contains 14,000 purely topological random SBM graphs with 44-188 nodes, whose task is finding certain subgraph pattern by binary node classification.

Table 3: Results for $S_n$ equivariant node classification on PATTERN. We report test accuracy at the best validation accuracy, along with the standard deviation for GA and Ours where predictions are stochastic. The results for GNN baselines are from [27].

| method | pretrain. | Accuracy ↑ |
|---|---|---|
| GCN [48], 16 layers | - | 85.614 |
| GAT [97], 16 layers | - | 78.271 |
| GatedGCN [11], 16 layers | - | 85.568 |
| GIN [101], 16 layers | - | 85.387 |
| RingGNN [16], 2 layers | - | 86.245 |
| RingGNN [16], 8 layers | - | diverged |
| PPGN [56], 3 layers | - | 85.661 |
| PPGN [56], 8 layers | - | diverged |
| ViT-GA, 1-sample | - | $76.776 \pm 0.137$ |
| ViT-GA, 10-sample | - | $83.119 \pm 0.048$ |
| ViT-GA, 1-sample | ImageNet-21k | $81.407 \pm 0.101$ |
| ViT-GA, 10-sample | ImageNet-21k | $84.351 \pm 0.053$ |
| ViT-FA | - | 70.063 |
| ViT-FA | ImageNet-21k | 79.637 |
| ViT-Canonical. | - | 85.460 |
| ViT-Canonical. | ImageNet-21k | 86.166 |
| ViT-PS (Ours), 1-sample | - | $85.542 \pm 0.012$ |
| ViT-PS (Ours), 10-sample | - | $85.635 \pm 0.021$ |
| ViT-PS (Ours), 1-sample | ImageNet-21k | $86.226 \pm 0.028$ |
| ViT-PS (Ours), 10-sample | ImageNet-21k | $\mathbf{86.285 \pm 0.015}$ |

Based on pre-trained ViT [26, 89], we construct the base model $f_\theta$ by modifying only the input and output layers to take the flattened patches of 2D zero-padded adjacency matrices of size $188 \times 188$ and produce output as 1D per-node classification logits of length 188 with 2 channels. In addition to standard GNNs[4], we compare group averaging, frame averaging, and canonicalization to our method, and also test whether pre-trained representations from ImageNet-21k is beneficial for the task. For the $S_n$ equivariant distribution $p_\omega$ in our method and canonicalization, we use a 3-layer GIN [101] with only 0.02% of the base model parameters. Further details can be found in Appendix A.3.5.

The results are in Table 3. First, we observe that transferring pre-trained ViT parameters consistently improves node classification for all symmetrization methods. It indicates that some traits of the pre-trained visual representation can benefit learning graph tasks which vastly differ in both the underlying symmetry (translation invariance $\rightarrow$ $S_n$ equivariance) and the data generating process (natural images $\rightarrow$ random process of SBM). In particular, it is somewhat surprising that vision pretraining allows group averaging to achieve 84.351% accuracy, on par with GNN baselines, considering that memorizing all 188! equally possible permutations in this dataset is impossible. We conjecture that group averaged ViT can in some way learn meaningful graph representation internally to solve the task, and vision pretraining helps in acquiring the representation by providing a good initialization point or transferable computation motifs.

On the other hand, frame averaging shows a low performance, 79.637% accuracy with vision pretraining, which is also surprising considering that frames vastly reduce the sample space of symmetrization in general; in fact, the size of frame of each graph in PATTERN is exactly 1. We empirically observe that, unlike group averaging, ViT with frame averaging memorizes frames of training graphs rather than learning generalizable graph representations. In contrast, canonicalization that also uses a single sample per graph successfully learns the task with 86.166% accuracy. We conjecture that the learnable orderings provided by an equivariant neural network $\rho(g) = q_\omega(\mathbf{x})$ is more flexible and generalizable to unseen graphs compared to frames computed from fixed graph Laplacian eigenvectors. Lastly, our method achieves a better performance compared to other symmetrization methods, and the performance consistently improves with vision pretraining and more samples for testing. As a result, our model based on pre-trained ViT and 10 samples for testing achieves 86.285% test accuracy, surpassing all baselines.

---

[4]We note that careful engineering such as graph positional encoding improves performance of GNNs, but we have chosen simple and representative GNNs in the benchmark [27] to provide a controlled comparison.

Table 4: Results for real-world graph tasks. We report test performance at best validation performance.

| method | Peptides-func | Peptides-struct | PCQM-Contact | | | |
|---|---|---|---|---|---|---|
| | AP ↑ | MAE ↓ | Hits@1 ↑ | Hits@3 ↑ | Hits@10 ↑ | MRR ↑ |
| GCN [48] | 0.5930 | 0.3496 | 0.1321 | 0.3791 | 0.8256 | 0.3234 |
| GCNII [15] | 0.5543 | 0.3471 | 0.1325 | 0.3607 | 0.8116 | 0.3161 |
| GINE [36] | 0.5498 | 0.3547 | 0.1337 | 0.3642 | 0.8147 | 0.3180 |
| GatedGCN [11] | 0.5864 | 0.3420 | 0.1279 | 0.3783 | 0.8433 | 0.3218 |
| GatedGCN+RWSE [11] | 0.6069 | 0.3357 | 0.1288 | 0.3808 | 0.8517 | 0.3242 |
| Transformer+LapPE [28] | 0.6326 | 0.2529 | 0.1221 | 0.3679 | 0.8517 | 0.3174 |
| SAN+LapPE [50] | 0.6384 | 0.2683 | 0.1355 | 0.4004 | 0.8478 | 0.3350 |
| SAN+RWSE [50] | 0.6439 | 0.2545 | 0.1312 | 0.4030 | 0.8550 | 0.3341 |
| GraphGPS [76] | 0.6535 | 0.2500 | - | - | - | 0.3337 |
| Exphormer [87] | 0.6527 | **0.2481** | - | - | - | 0.3637 |
| ViT-PS (Ours) | **0.8311** | 0.2662 | **0.3268** | **0.6693** | **0.9524** | **0.5329** |

## 3.4 Real-World Graph Learning with Vision Transformer

Having observed that pre-trained ViT can learn graph tasks well when symmetrized with our method, we now provide a preliminary test of it in real-world graph learning. We use three real-world graph datasets from [28] that involve chemical and biological graphs. PCQM-Contact dataset contains 529,434 molecular graphs with 53 nodes in maximum, and the task is contact map prediction framed as link prediction ($S_n$ equivariant), on whether two atoms would be proximal when the molecule is in 3D space. Peptides-func and Peptides-struct are based on the same set of 15,535 protein graphs with 444 nodes in maximum and the tasks are property prediction ($S_n$ invariant), requiring multi-label classification for Peptides-func and regression for Peptides-struct. The tasks require complex understanding of how the amino acids of the proteins would interact in 3D space. We implement our method using a ViT-Base pre-trained on ImageNet-21k as the base model $f_\theta$ and a 3-layer GIN as the equivariant distribution $p_\omega(g|\mathbf{x})$, following our model in Section 3.3. We use 10 samples for both training and testing. Further details can be found in Appendix A.3.6.

The results are in Table 4. In Peptides-func and PCQM-Contact, the pre-trained ViT symmetrized with our method achieves a significantly higher performance compared to both GNNs and graph transformers, improving previous best by a large margin[5] (0.6527 → 0.8311 for Peptides-func AP, 0.1355 → 0.3268 for PCQM-Contact Hits@1). This demonstrates the scalability of our method as Peptides-func involves 444 maximum nodes, and also its generality as it performs well for both $S_n$ invariant (Peptides-func) and equivariant (PCQM-Contact) tasks. We also note that, unlike some baselines, our method does not require costly Laplacian eigenvectors to compute positional encoding. On the Peptides-struct, our method achieves a slightly lower performance to SOTA graph transformers while still better than GNNs. We conjecture that regression is harder for the model to learn due to its stochasticity in predictions, and leave improving regression performance as future work.

## 4 Conclusion

We presented probabilistic symmetrization, a general framework that learns a distribution of group transformations conditioned on input data for symmetrization of an arbitrary function. By characterizing that the only condition for such distribution is equivariance to data symmetry, we instantiated models for a wide range of groups, including symmetric, orthogonal, Euclidean groups and their product combinations. Our experiments demonstrated that the proposed framework achieves consistent improvement over other symmetrization methods, and is competitive or outperforms equivariant networks on various datasets. We also showed that transferring pre-trained parameters across data in different symmetries can sometimes be surprisingly beneficial. Our approach has weaknesses such as sampling cost, which we further discuss in Appendix A.5; we plan to address these in future work.

**Acknowledgements** This work was supported in part by the National Research Foundation of Korea (NRF2021R1C1C1012540 and NRF2021R1A4A3032834) and IITP grant (2021-0-00537, 2019-0-00075, and 2021-0-02068) funded by the Korea government (MSIT).

---

[5]We note that the baseline architectures are constructed within 500k parameter budget as a convention [28], while we use an identical architecture to ViT-Base to leverage pre-trained representations.

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

# A Appendix

## A.1 Proofs

### A.1.1 Proof of Theorem 1 (Section 2.1)

**Theorem 1.** *If $p_\omega$ is $G$ equivariant, then $\phi_{\theta,\omega}$ is $G$ equivariant for arbitrary $f_\theta$.*

*Proof.* We prove $\phi_{\theta,\omega}(\rho_1(g')\mathbf{x}) = \rho_2(g')\phi_{\theta,\omega}(\mathbf{x})$ for all $\mathbf{x} \in \mathcal{X}$ and $g' \in G$. From Eq. (4), we have:

$$\phi_{\theta,\omega}(\rho_1(g')\mathbf{x}) = \mathbb{E}_{p_\omega(g|\rho_1(g')\mathbf{x})} \left[ \rho_2(g) f_\theta(\rho_1(g)^{-1}\rho_1(g')\mathbf{x}) \right]. \tag{11}$$

Let us introduce transformed random variable $h = g'^{-1}g \in G$ such that $g = g'h$. Since the distribution $p_\omega$ is $G$ equivariant, we can see that $p_\omega(g|\rho_1(g')\mathbf{x}) = p_\omega(g'^{-1}g|\rho_1(g'^{-1})\rho_1(g')\mathbf{x}) = p_\omega(g'^{-1}g|\mathbf{x}) = p_\omega(h|\mathbf{x})$. Thus, we can rewrite the above expectation with respect to $h$ as follows:

$$\begin{aligned}
\phi_{\theta,\omega}(\rho_1(g')\mathbf{x}) &= \mathbb{E}_{p_\omega(h|\mathbf{x})} \left[ \rho_2(g'h) f_\theta(\rho_1(g'h)^{-1}\rho_1(g')\mathbf{x}) \right] \\
&= \mathbb{E}_{p_\omega(h|\mathbf{x})} \left[ \rho_2(g')\rho_2(h) f_\theta(\rho_1(h)^{-1}\rho_1(g')^{-1}\rho_1(g')\mathbf{x}) \right] \\
&= \rho_2(g')\mathbb{E}_{p_\omega(h|\mathbf{x})} \left[ \rho_2(h) f_\theta(\rho_1(h)^{-1}\mathbf{x}) \right] \\
&= \rho_2(g')\phi_{\theta,\omega}(\mathbf{x}),
\end{aligned} \tag{12}$$

showing the $G$ equivariance of $\phi_{\theta,\omega}$ for arbitrary $f_\theta$. $\qquad\square$

### A.1.2 Proof of Theorem 2 (Section 2.1)

**Theorem 2.** *If $p_\omega$ is $G$ equivariant and $f_\theta$ is a universal approximator, then $\phi_{\theta,\omega}$ is a universal approximator of $G$ equivariant functions.*

*Proof.* The proof is inspired by universality proofs of prior symmetrization approaches [102, 74, 41]. Let $\psi : \mathcal{X} \to \mathcal{Y}$ be an arbitrary $G$ equivariant function. By equivariance of $\psi$, we have:

$$\begin{aligned}
\|\psi(\mathbf{x}) - \phi_{\theta,\omega}(\mathbf{x})\| &= \left\| \psi(\mathbf{x}) - \mathbb{E}_{p_\omega(g|\mathbf{x})} \left[ \rho_2(g) f_\theta(\rho_1(g)^{-1}\mathbf{x}) \right] \right\| \\
&= \left\| \mathbb{E}_{p_\omega(g|\mathbf{x})} \left[ \psi(\mathbf{x}) \right] - \mathbb{E}_{p_\omega(g|\mathbf{x})} \left[ \rho_2(g) f_\theta(\rho_1(g)^{-1}\mathbf{x}) \right] \right\| \\
&= \left\| \mathbb{E}_{p_\omega(g|\mathbf{x})} \left[ \rho_2(g)\rho_2(g)^{-1}\psi(\mathbf{x}) \right] - \mathbb{E}_{p_\omega(g|\mathbf{x})} \left[ \rho_2(g) f_\theta(\rho_1(g)^{-1}\mathbf{x}) \right] \right\| \\
&= \left\| \mathbb{E}_{p_\omega(g|\mathbf{x})} \left[ \rho_2(g)\psi(\rho_1(g)^{-1}\mathbf{x}) \right] - \mathbb{E}_{p_\omega(g|\mathbf{x})} \left[ \rho_2(g) f_\theta(\rho_1(g)^{-1}\mathbf{x}) \right] \right\| \\
&= \left\| \mathbb{E}_{p_\omega(g|\mathbf{x})} \left[ \rho_2(g)\psi(\rho_1(g)^{-1}\mathbf{x}) - \rho_2(g) f_\theta(\rho_1(g)^{-1}\mathbf{x}) \right] \right\|.
\end{aligned} \tag{13}$$

As $\mathcal{Y}$ is finite-dimensional, we can assume that the linear operators in $\mathrm{GL}(\mathcal{Y})$ are bounded and so is the induced operator norm of group representation $\|\rho_2(g)\|$ for all $g \in G$. Thus, we have:

$$\begin{aligned}
\|\psi(\mathbf{x}) - \phi_{\theta,\omega}(\mathbf{x})\| &\leq \max_{h \in G} \|\rho_2(h)\| \left\| \mathbb{E}_{p_\omega(g|\mathbf{x})} \left[ \psi(\rho_1(g)^{-1}\mathbf{x}) - f_\theta(\rho_1(g)^{-1}\mathbf{x}) \right] \right\| \\
&\leq c \left\| \mathbb{E}_{p_\omega(g|\mathbf{x})} \left[ \psi(\rho_1(g)^{-1}\mathbf{x}) - f_\theta(\rho_1(g)^{-1}\mathbf{x}) \right] \right\|.
\end{aligned} \tag{14}$$

for some $c > 0$. If $f_\theta$ is a universal approximator, for any compact set $\mathcal{K} \subseteq \mathcal{X}$ and any $\epsilon > 0$, there exists some $\theta$ such that $\|\psi(\mathbf{x}) - f_\theta(\mathbf{x})\| \leq \epsilon$ for all $\mathbf{x} \in \mathcal{K}$. Consider the set $\mathcal{K}_{\mathrm{sym}} = \cup_{g \in G}\rho_1(g)\mathcal{K}$ where $\rho_1(g)\mathcal{K}$ denotes the image of the set $\mathcal{K}$ under linear transformation by $\rho_1(g)$. We use the fact that $\mathcal{K}_{\mathrm{sym}}$ is also a compact set since it is the image of the compact set $G \times \mathcal{K}$ under continuous map $(g, \mathbf{x}) \mapsto \rho_1(g)\mathbf{x}$. As a consequence, for any compact set $\mathcal{K} \subseteq \mathcal{X}$ and any $\epsilon/c > 0$, there exists some $\theta$ such that $\max_{g \in G} \|\psi(\rho_1(g)\mathbf{x}) - f_\theta(\rho_1(g)\mathbf{x})\| \leq \epsilon/c$ for all $\mathbf{x} \in \mathcal{K}$. Since a group is closed under inverse, for any compact set $\mathcal{K} \subseteq \mathcal{X}$ and any $\epsilon > 0$, there exists some $\theta$ such that:

$$\begin{aligned}
\|\psi(\mathbf{x}) - \phi_{\theta,\omega}(\mathbf{x})\| &\leq c \left\| \mathbb{E}_{p_\omega(g|\mathbf{x})} \left[ \psi(\rho_1(g)^{-1}\mathbf{x}) - f_\theta(\rho_1(g)^{-1}\mathbf{x}) \right] \right\| \\
&\leq c \max_{g \in G} \left\| \psi(\rho_1(g)^{-1}\mathbf{x}) - f_\theta(\rho_1(g)^{-1}\mathbf{x}) \right\| \\
&= \epsilon,
\end{aligned} \tag{15}$$

for all $\mathbf{x} \in \mathcal{K}$, showing that $\phi_{\theta,\omega}$ is a universal approximator of $G$ equivariant functions. $\qquad\square$

While we have assumed that the group $G$ is compact in the proof, we conjecture that the results can be extended to non-compact groups if we make an alternative assumption that the distribution $p_\omega(g|\mathbf{x})$ is compactly supported for all $\mathbf{x} \in \mathcal{K}$. We leave proving this as a future work.

### A.1.3 Proof of Theorem 3 (Section 2.1)

**Theorem 3.** *If $q_\omega$ is $G$ equivariant and $p(\epsilon)$ is $G$ invariant under representation $\rho'$ that $|\det \rho'(g)| = 1 \forall g \in G$, the distribution $p_\omega(g|\mathbf{x})$ characterized by $q_\omega : (\mathbf{x}, \epsilon) \mapsto \rho(g)$ is $G$ equivariant.*

*Proof.* We prove $p_\omega(g'g|\rho_1(g')\mathbf{x}) = p_\omega(g|\mathbf{x})$ for all $\mathbf{x} \in \mathcal{X}$ and $g, g' \in G$. In general, we are interested in obtaining a faithful representation $\rho$, *i.e.*, such that $\rho(g)$ is distinct for each $g$. We can interpret the probability $p_\omega(g|\mathbf{x}, \epsilon)$ as a delta distribution centered at the group representation $\rho(g)$:

$$p_\omega(g|\mathbf{x}, \epsilon) = \delta(\rho(g) = q_\omega(\mathbf{x}, \epsilon)). \tag{16}$$

To obtain $p_\omega(g|\mathbf{x})$, we marginalize over $p(\epsilon)$:

$$p_\omega(g|\mathbf{x}) = \int_\epsilon p_\omega(g|\mathbf{x}, \epsilon) p(\epsilon) d\epsilon$$
$$= \int_\epsilon \delta(\rho(g) = q_\omega(\mathbf{x}, \epsilon)) p(\epsilon) d\epsilon. \tag{17}$$

Let us consider $p_\omega(g'g|\rho_1(g')\mathbf{x})$:

$$p_\omega(g'g|\rho_1(g')\mathbf{x}) = \int_\epsilon \delta(\rho(g'g) = q_\omega(\rho_1(g')\mathbf{x}, \epsilon)) p(\epsilon) d\epsilon. \tag{18}$$

Using the $G$ equivariance of $q_\omega$, we have:

$$q_\omega(\rho_1(g')\mathbf{x}, \epsilon) = \rho(g') q_\omega(\rho_1(g'^{-1})\rho_1(g')\mathbf{x}, \rho'(g'^{-1})\epsilon)$$
$$= \rho(g') q_\omega(\mathbf{x}, \rho'(g'^{-1})\epsilon) \tag{19}$$

which leads to the following:

$$p_\omega(g'g|\rho_1(g')\mathbf{x}) = \int_\epsilon \delta(\rho(g'g) = \rho(g') q_\omega(\mathbf{x}, \rho'(g'^{-1})\epsilon)) p(\epsilon) d\epsilon$$
$$= \int_\epsilon \delta(\rho(g) = q_\omega(\mathbf{x}, \rho'(g'^{-1})\epsilon)) p(\epsilon) d\epsilon. \tag{20}$$

Note that the second equality follows from invertibility of $\rho(g')$. We now introduce a change of variables $\epsilon' = \rho'(g'^{-1})\epsilon$ that $\epsilon = \rho'(g')\epsilon'$:

$$p_\omega(g'g|\rho_1(g')\mathbf{x}) = \int_{\epsilon'} \delta(\rho(g) = q_\omega(\mathbf{x}, \epsilon')) p(\rho'(g')\epsilon') \frac{1}{|\det \rho'(g'^{-1})|} d\epsilon'. \tag{21}$$

With $|\det \rho'(g'^{-1})| = 1$, and $G$ invariance of $p(\epsilon)$ which gives $p(\rho'(g')\epsilon') = p(\epsilon')$, we get:

$$p_\omega(g'g|\rho_1(g')\mathbf{x}) = \int_{\epsilon'} \delta(\rho(g) = q_\omega(\mathbf{x}, \epsilon')) p(\epsilon') d\epsilon'$$
$$= p_\omega(g|\mathbf{x}), \tag{22}$$

showing the $G$ equivariance of $p_\omega(g|\mathbf{x})$. $\qquad\square$

### A.1.4 Proof of Validity for Implemented Equivariant Distributions $p_\omega$ (Section 2.2)

We formally show $G$ equivariance of the implemented distributions $p_\omega(g|\mathbf{x})$ presented in Section 2.2. All implementations have a form of noise-outsourced function $q_\omega : (\mathbf{x}, \epsilon) \mapsto \rho(g)$ using distribution $\epsilon \sim p(\epsilon)$ and map $q_\omega$ which is composed of $G$ equivariant neural network and postprocessing to $\rho(g)$. From Theorem 3, for $G$ equivariance of $p_\omega(g|\mathbf{x})$, it is sufficient to show $G$ invariance of $p(\epsilon)$ under a representation $\rho'$ such that $|\det \rho'(g)| = 1$ along with $G$ equivariance of $q_\omega$, which we show below.

**Symmetric Group $S_n$**    We recall that $p_\omega(g|\mathbf{x})$ for the symmetric group $S_n$ is implemented as below:

1. Sample node-level noise $\epsilon \in \mathbb{R}^{n \times d}$ from i.i.d. uniform $\mathrm{Unif}[0, \eta]$.
2. Use a GNN to obtain node-level scalar features $(\mathbf{x}, \epsilon) \mapsto \mathbf{Z} \in \mathbb{R}^n$.

3. Assuming $\mathbf{Z}$ is tie-free, use argsort [98] to obtain group representation $\mathbf{Z} \mapsto \mathbf{P}_g = \rho(g)$.

$$\mathbf{P}_g = \mathrm{eq}(\mathbf{Z}\mathbf{1}^\top, \mathbf{1}\mathrm{sort}(\mathbf{Z})^\top), \tag{23}$$

where eq denotes elementwise equality indicator.

We now show the following:

**Proposition 3.** *The proposed distribution $p_\omega(g|\mathbf{x})$ for the symmetric group $\mathrm{S}_n$ is equivariant.*

*Proof.* Given $p(\boldsymbol{\epsilon})$ is elementwise i.i.d., it is $\mathrm{S}_n$ invariant under the base representation $\rho'(g) = \mathbf{P}_g$ which satisfies $|\det \mathbf{P}_g| = 1$ from orthogonality. As a GNN is $\mathrm{S}_n$ equivariant, we only need to show $\mathrm{S}_n$ equivariance of argsort : $\mathbf{Z} \mapsto \mathbf{P}_g$. This can be shown by transforming $\mathbf{Z}$ with any permutation matrix $\mathbf{P}_{g'}$. Since sort operator and any row replicated matrices are invariant to $\mathbf{P}_{g'}$, we have:

$$\begin{aligned}
\mathrm{eq}(\mathbf{P}_{g'}\mathbf{Z}\mathbf{1}^\top, \mathbf{1}\mathrm{sort}(\mathbf{P}_{g'}\mathbf{Z})^\top) &= \mathrm{eq}(\mathbf{P}_{g'}\mathbf{Z}\mathbf{1}^\top, \mathbf{1}\mathrm{sort}(\mathbf{Z})^\top) \\
&= \mathrm{eq}(\mathbf{P}_{g'}\mathbf{Z}\mathbf{1}^\top, \mathbf{P}_{g'}\mathbf{1}\mathrm{sort}(\mathbf{Z})^\top).
\end{aligned} \tag{24}$$

Since eq commutes with $\mathbf{P}_{g'}$, we have:

$$\begin{aligned}
\mathrm{eq}(\mathbf{P}_{g'}\mathbf{Z}\mathbf{1}^\top, \mathbf{1}\mathrm{sort}(\mathbf{P}_{g'}\mathbf{Z})^\top) &= \mathrm{eq}(\mathbf{P}_{g'}\mathbf{Z}\mathbf{1}^\top, \mathbf{P}_{g'}\mathbf{1}\mathrm{sort}(\mathbf{Z})^\top) \\
&= \mathbf{P}_{g'}\mathrm{eq}(\mathbf{Z}\mathbf{1}^\top, \mathbf{1}\mathrm{sort}(\mathbf{Z})^\top) \\
&= \mathbf{P}_{g'}\mathbf{P}_g,
\end{aligned} \tag{25}$$

showing that argsort is $\mathrm{S}_n$ equivariant, *i.e.*, it maps $\mathbf{P}_{g'}\mathbf{Z} \mapsto \mathbf{P}_{g'}\mathbf{P}_g$ for all $\mathbf{P}_{g'} \in \mathrm{S}_n$. Combining the above, by Theorem 3, the distribution $p_\omega(g|\mathbf{x})$ is $\mathrm{S}_n$ equivariant. $\square$

**Orthogonal Group** $\mathrm{O}(n)$**,** $\mathrm{SO}(n)$      We recall that $p_\omega(g|\mathbf{x})$ for the orthogonal group $\mathrm{O}(n)$ or special orthogonal group $\mathrm{SO}(n)$ is implemented as follows:

1. Sample noise $\boldsymbol{\epsilon} \in \mathbb{R}^{n \times d}$ from i.i.d. normal $\mathcal{N}(0, \eta^2)$.

2. Use an $\mathrm{O}(n)/\mathrm{SO}(n)$ equivariant neural network to obtain $n$ features $(\mathbf{x}, \boldsymbol{\epsilon}) \mapsto \mathbf{Z} \in \mathbb{R}^{n \times n}$.

3. Assuming $\mathbf{Z}$ is full-rank, use Gram-Schmidt process [41] to obtain an orthogonal matrix $\mathbf{Z} \mapsto \mathbf{Q}$.

4. For the $\mathrm{O}(n)$ group, use the obtained matrix as group representation $\mathbf{Q} = \mathbf{Q}_g = \rho(g)$.

5. For the $\mathrm{SO}(n)$ group, use below scale operator to obtain group representation $\mathbf{Q} \mapsto \mathbf{Q}_g^+ = \rho(g)$.

$$\mathrm{scale} : \left[\begin{array}{c|c|c} \mathbf{Q}_1 & \cdots & \mathbf{Q}_n \end{array}\right] \mapsto \left[\begin{array}{c|c|c} \det(\mathbf{Q}) \cdot \mathbf{Q}_1 & \cdots & \mathbf{Q}_n \end{array}\right]. \tag{26}$$

We now show the following:

**Proposition 4.** *The proposed distribution $p_\omega(g|\mathbf{x})$ for the orthogonal group $\mathrm{O}(n)$ is equivariant.*

*Proof.* Without loss of generality, let us omit the scale $\eta$ for brevity, which gives that each column $\boldsymbol{\epsilon}_i \in \mathbb{R}^n$ of the noise $\boldsymbol{\epsilon}$ independently follows multivariate standard normal $\boldsymbol{\epsilon}_i \sim \mathcal{N}(0, \mathbf{I}_n)$. Then, the density $p(\boldsymbol{\epsilon}_i) = (2\pi)^{-n/2} \exp\left(-\|\boldsymbol{\epsilon}_i\|_2^2/2\right)$ is invariant under orthogonal transformation $\mathbf{Q}$ since $\|\mathbf{Q}\boldsymbol{\epsilon}_i\|_2^2 = (\mathbf{Q}\boldsymbol{\epsilon}_i)^\top \mathbf{Q}\boldsymbol{\epsilon}_i = \boldsymbol{\epsilon}_i^\top \mathbf{Q}^\top \mathbf{Q}\boldsymbol{\epsilon}_i = \boldsymbol{\epsilon}_i^\top \boldsymbol{\epsilon}_i = \|\boldsymbol{\epsilon}_i\|_2^2$. Therefore, the distribution $p(\boldsymbol{\epsilon})$ is invariant under the base representation $\rho'(g) = \mathbf{Q}_g$ which satisfies $|\det \rho'(g)| = 1$ from orthogonality. As we use an equivariant neural network to obtain $\mathbf{Z}$, and Gram-Schmidt procedure $\mathbf{Z} \mapsto \mathbf{Q}_g$ is $\mathrm{O}(n)$ equivariant (Theorem 5 of [41]), by Theorem 3, the distribution $p_\omega(g|\mathbf{x})$ is $\mathrm{O}(n)$ equivariant. $\square$

**Proposition 5.** *The proposed distribution $p_\omega(g|\mathbf{x})$ for special orthogonal group $\mathrm{SO}(n)$ is equivariant.*

*Proof.* From the proof of Proposition 4, it follows that the distribution $p(\boldsymbol{\epsilon})$ is invariant under the base representation $\rho'(g) = \mathbf{Q}_g^+$ which satisfies $|\det \rho'(g)| = 1$ due to orthogonality. As we use an equivariant neural network to obtain $\mathbf{Z}$, and Gram-Schmidt procedure $\mathbf{Z} \mapsto \mathbf{Q}$ has $\mathrm{O}(n)$ equivariance which implies $\mathrm{SO}(n)$ equivariance because of $\mathrm{SO}(n) \leq \mathrm{O}(n)$, we only need to show

SO($n$) equivariance of scale : $\mathbf{Q} \mapsto \mathbf{Q}_g^+$. This can be done by transforming $\mathbf{Q}$ with an orthogonal $\mathbf{Q}_{g'}^+$ of determinant $+1$. Since $\det(\mathbf{Q}_{g'}^+\mathbf{Q}) = \det(\mathbf{Q}_{g'}^+)\det(\mathbf{Q}) = \det(\mathbf{Q})$, we have the following:

$$\text{scale}(\mathbf{Q}_{g'}^+\mathbf{Q}) = \left[\; \det(\mathbf{Q}_{g'}^+\mathbf{Q}) \cdot (\mathbf{Q}_{g'}^+\mathbf{Q})_1 \;\middle|\; \cdots \;\middle|\; (\mathbf{Q}_{g'}^+\mathbf{Q})_n \;\right]$$

$$= \left[\; \det(\mathbf{Q}) \cdot (\mathbf{Q}_{g'}^+\mathbf{Q})_1 \;\middle|\; \cdots \;\middle|\; (\mathbf{Q}_{g'}^+\mathbf{Q})_n \;\right]. \tag{27}$$

Also, scaling the first column of the product $\mathbf{Q}_{g'}^+\mathbf{Q}$ with $\det(\mathbf{Q})$ is equivalent to scaling the first column of $\mathbf{Q}$ with $\det(\mathbf{Q})$ then computing the product since $(\mathbf{Q}_{g'}^+\mathbf{Q})_{ij} = \sum_k \mathbf{Q}_{g'ik}^+ \mathbf{Q}_{kj}$. This gives:

$$\text{scale}(\mathbf{Q}_{g'}^+\mathbf{Q}) = \mathbf{Q}_{g'}^+ \left[\; \det(\mathbf{Q}) \cdot \mathbf{Q}_1 \;\middle|\; \cdots \;\middle|\; \mathbf{Q}_n \;\right]$$

$$= \mathbf{Q}_{g'}^+ \text{scale}(\mathbf{Q}), \tag{28}$$

showing that scale operator is SO($n$) equivariant. We also note that scale($\mathbf{Q}$) gives orthogonal matrix of determinant $+1$, as it returns $\mathbf{Q}$ if $\det(\mathbf{Q}) = +1$, otherwise $(\det(\mathbf{Q}) = -1$ since $\mathbf{Q}$ is orthogonal) scales the first column by $-1$ which flips determinant to $+1$ while not affecting orthogonality. Combining the above, by Theorem 3, the distribution $p_\omega(g|\mathbf{x})$ is SO($n$) equivariant. $\qquad\square$

**Euclidean Group** E($n$), SE($n$)   We recall that, unlike the other groups, we handle the Euclidean group E($n$) and special Euclidean group SE($n$) at symmetrization level as the translation component T($n$) in E($n$) = O($n$) $\ltimes$ T($n$) and SE($n$) = SO($n$) $\ltimes$ T($n$) is non-compact. This is done as follows:

$$\phi_{\theta,\omega}(\mathbf{x}) = \mathbb{E}_{p_\omega(g|\mathbf{x}-\bar{\mathbf{x}}\mathbf{1}^\top)}\left[\bar{\mathbf{x}}\mathbf{1}^\top + g \cdot f_\theta(g^{-1} \cdot (\mathbf{x} - \bar{\mathbf{x}}\mathbf{1}^\top))\right], \tag{29}$$

where $\bar{\mathbf{x}} \in \mathbb{R}^n$ is centroid (mean over channels) of data $\mathbf{x} \in \mathbb{R}^{n \times d}$ and distribution $p_\omega$ is O($n$)/SO($n$) equivariant for E($n$)/SE($n$) equivariant symmetrization, respectively. We now show the following:

**Proposition 6.** *The proposed symmetrization $\phi_{\theta,\omega}$ for the Euclidean group E($n$) is equivariant.*

*Proof.* We prove $\phi_{\theta,\omega}(g' \cdot \mathbf{x}) = g' \cdot \phi_{\theta,\omega}(\mathbf{x})$ for all $\mathbf{x} \in \mathcal{X}$ and $g' \in$ E($n$). From Eq. (29), we have:

$$\phi_{\theta,\omega}(g' \cdot \mathbf{x}) = \mathbb{E}_{p_\omega(g|g'\cdot\mathbf{x}-\overline{g'\cdot\mathbf{x}}\mathbf{1}^\top)}\left[\overline{g'\cdot\mathbf{x}}\mathbf{1}^\top + g \cdot f_\theta(g^{-1} \cdot (g' \cdot \mathbf{x} - \overline{g'\cdot\mathbf{x}}\mathbf{1}^\top))\right]. \tag{30}$$

In general, an element of Euclidean group $g' \in$ E($n$) acts on data $\mathbf{x} \in \mathbb{R}^{n \times d}$ via $g' \cdot \mathbf{x} = \mathbf{Q}_{g'}\mathbf{x} + \mathbf{t}_{g'}\mathbf{1}^\top$ where $\mathbf{Q}_{g'} \in$ O($n$) is its rotation component and $\mathbf{t}_{g'} \in \mathbb{R}^n$ is its translation component [74, 41]. With this, the centroid of the transformed data $g' \cdot \mathbf{x}$ is given as follows:

$$\overline{g' \cdot \mathbf{x}} = \overline{\mathbf{Q}_{g'}\mathbf{x} + \mathbf{t}_{g'}\mathbf{1}^\top} = \overline{\mathbf{Q}_{g'}\mathbf{x}} + \mathbf{t}_{g'} = \mathbf{Q}_{g'}\bar{\mathbf{x}} + \mathbf{t}_{g'}, \tag{31}$$

which leads to the following:

$$g' \cdot \mathbf{x} - \overline{g' \cdot \mathbf{x}}\mathbf{1}^\top = \mathbf{Q}_{g'}\mathbf{x} + \mathbf{t}_{g'}\mathbf{1}^\top - \mathbf{Q}_{g'}\bar{\mathbf{x}}\mathbf{1}^\top - \mathbf{t}_{g'}\mathbf{1}^\top$$

$$= \mathbf{Q}_{g'}(\mathbf{x} - \bar{\mathbf{x}}\mathbf{1}^\top). \tag{32}$$

Above shows that subtracting centroid eliminates the translation component of the problem and leaves O($n$) equivariance component. Based on that, we have the following:

$$\phi_{\theta,\omega}(g' \cdot \mathbf{x}) = \mathbb{E}_{p_\omega(g|\mathbf{Q}_{g'}(\mathbf{x}-\bar{\mathbf{x}}\mathbf{1}^\top))}\left[\mathbf{Q}_{g'}\bar{\mathbf{x}}\mathbf{1}^\top + \mathbf{t}_{g'}\mathbf{1}^\top + g \cdot f_\theta(g^{-1} \cdot (\mathbf{Q}_{g'}(\mathbf{x} - \bar{\mathbf{x}}\mathbf{1}^\top)))\right]$$

$$= \mathbb{E}_{p_\omega(g|g'\cdot(\mathbf{x}-\bar{\mathbf{x}}\mathbf{1}^\top))}\left[g' \cdot \bar{\mathbf{x}}\mathbf{1}^\top + g \cdot f_\theta(g^{-1}g' \cdot (\mathbf{x} - \bar{\mathbf{x}}\mathbf{1}^\top))\right] + \mathbf{t}_{g'}\mathbf{1}^\top. \tag{33}$$

Note that, inside the expectation, we interpret the rotation component of $g'$ as an element of the orthogonal group O($n$). Similar as in the proof of Theorem 1, we introduce transformed random variable $h = g'^{-1}g \in$ O($n$) that $g = g'h$. Since the distribution $p_\omega$ is O($n$) equivariant, we can see

that $p_\omega(g|g' \cdot (\mathbf{x} - \overline{\mathbf{x}}\mathbf{1}^\top)) = p_\omega(g'^{-1}g|g'^{-1}g' \cdot (\mathbf{x} - \overline{\mathbf{x}}\mathbf{1}^\top)) = p_\omega(g'^{-1}g|\mathbf{x} - \overline{\mathbf{x}}\mathbf{1}^\top) = p_\omega(h|\mathbf{x} - \overline{\mathbf{x}}\mathbf{1}^\top)$.
Thus we can rewrite the above expectation with respect to $h$ as follows:

$$
\begin{aligned}
\phi_{\theta,\omega}(g' \cdot \mathbf{x}) &= \mathbb{E}_{p_\omega(h|\mathbf{x} - \overline{\mathbf{x}}\mathbf{1}^\top)}\left[ g' \cdot \overline{\mathbf{x}}\mathbf{1}^\top + g'h \cdot f_\theta((g'h)^{-1}g' \cdot (\mathbf{x} - \overline{\mathbf{x}}\mathbf{1}^\top)) \right] + \mathbf{t}_{g'}\mathbf{1}^\top \\
&= \mathbb{E}_{p_\omega(h|\mathbf{x} - \overline{\mathbf{x}}\mathbf{1}^\top)}\left[ g' \cdot \overline{\mathbf{x}}\mathbf{1}^\top + g'h \cdot f_\theta(h^{-1} \cdot (\mathbf{x} - \overline{\mathbf{x}}\mathbf{1}^\top)) \right] + \mathbf{t}_{g'}\mathbf{1}^\top \\
&= \mathbf{Q}_{g'}\mathbb{E}_{p_\omega(h|\mathbf{x} - \overline{\mathbf{x}}\mathbf{1}^\top)}\left[ \overline{\mathbf{x}}\mathbf{1}^\top + h \cdot f_\theta(h^{-1} \cdot (\mathbf{x} - \overline{\mathbf{x}}\mathbf{1}^\top)) \right] + \mathbf{t}_{g'}\mathbf{1}^\top \\
&= \mathbf{Q}_{g'}\phi_{\theta,\omega}(\mathbf{x}) + \mathbf{t}_{g'}\mathbf{1}^\top \\
&= g' \cdot \phi_{\theta,\omega}(\mathbf{x}),
\end{aligned}
\tag{34}
$$

showing the $\mathrm{E}(n)$ equivariance of $\phi_{\theta,\omega}$. $\qquad\square$

**Proposition 7.** *The proposed symmetrization $\phi_{\theta,\omega}$ for special Euclidean group $\mathrm{SE}(n)$ is equivariant.*

*Proof.* We prove $\phi_{\theta,\omega}(g' \cdot \mathbf{x}) = g' \cdot \phi_{\theta,\omega}(\mathbf{x})$ for all $\mathbf{x} \in \mathcal{X}$ and $g' \in \mathrm{SE}(n)$, in an analogous manner to the proof of Proposition 6. From Eq. (29), we have:

$$
\phi_{\theta,\omega}(g' \cdot \mathbf{x}) = \mathbb{E}_{p_\omega(g|g' \cdot \mathbf{x} - \overline{g' \cdot \mathbf{x}}\mathbf{1}^\top)}\left[ \overline{g' \cdot \mathbf{x}}\mathbf{1}^\top + g \cdot f_\theta(g^{-1} \cdot (g' \cdot \mathbf{x} - \overline{g' \cdot \mathbf{x}}\mathbf{1}^\top)) \right].
\tag{35}
$$

In general, an element of special Euclidean group $g' \in \mathrm{SE}(n)$ acts on data $\mathbf{x} \in \mathbb{R}^{n \times d}$ via $g' \cdot \mathbf{x} = \mathbf{Q}_{g'}^+\mathbf{x} + \mathbf{t}_{g'}\mathbf{1}^\top$ where $\mathbf{Q}_{g'}^+ \in \mathrm{SO}(n)$ is rotation component and $\mathbf{t}_{g'} \in \mathbb{R}^n$ is translation [74, 41]. With this, the centroid of the transformed data $g' \cdot \mathbf{x}$ is given as follows:

$$
\overline{g' \cdot \mathbf{x}} = \overline{\mathbf{Q}_{g'}^+\mathbf{x} + \mathbf{t}_{g'}\mathbf{1}^\top} = \overline{\mathbf{Q}_{g'}^+\mathbf{x}} + \mathbf{t}_{g'} = \mathbf{Q}_{g'}^+\overline{\mathbf{x}} + \mathbf{t}_{g'},
\tag{36}
$$

which leads to the following:

$$
\begin{aligned}
g' \cdot \mathbf{x} - \overline{g' \cdot \mathbf{x}}\mathbf{1}^\top &= \mathbf{Q}_{g'}^+\mathbf{x} + \mathbf{t}_{g'}\mathbf{1}^\top - \mathbf{Q}_{g'}^+\overline{\mathbf{x}}\mathbf{1}^\top - \mathbf{t}_{g'}\mathbf{1}^\top \\
&= \mathbf{Q}_{g'}^+(\mathbf{x} - \overline{\mathbf{x}}\mathbf{1}^\top).
\end{aligned}
\tag{37}
$$

Similar as in Proposition 6, subtracting centroid only leaves $\mathrm{SO}(n)$ component. We then have:

$$
\begin{aligned}
\phi_{\theta,\omega}(g' \cdot \mathbf{x}) &= \mathbb{E}_{p_\omega(g|\mathbf{Q}_{g'}^+(\mathbf{x} - \overline{\mathbf{x}}\mathbf{1}^\top))}\left[ \mathbf{Q}_{g'}^+\overline{\mathbf{x}}\mathbf{1}^\top + \mathbf{t}_{g'}\mathbf{1}^\top + g \cdot f_\theta(g^{-1} \cdot (\mathbf{Q}_{g'}^+(\mathbf{x} - \overline{\mathbf{x}}\mathbf{1}^\top))) \right] \\
&= \mathbb{E}_{p_\omega(g|g' \cdot (\mathbf{x} - \overline{\mathbf{x}}\mathbf{1}^\top))}\left[ g' \cdot \overline{\mathbf{x}}\mathbf{1}^\top + g \cdot f_\theta(g^{-1}g' \cdot (\mathbf{x} - \overline{\mathbf{x}}\mathbf{1}^\top)) \right] + \mathbf{t}_{g'}\mathbf{1}^\top,
\end{aligned}
\tag{38}
$$

where, inside the expectation, we interpret the rotation component of $g'$ as an element of the special orthogonal group $\mathrm{SO}(n)$. Similar as in Theorem 1, we introduce $h = g'^{-1}g \in \mathrm{SO}(n)$ that $g = g'h$. As the distribution $p_\omega$ is $\mathrm{SO}(n)$ equivariant, we have $p_\omega(g|g' \cdot (\mathbf{x} - \overline{\mathbf{x}}\mathbf{1}^\top)) = p_\omega(g'^{-1}g|g'^{-1}g' \cdot (\mathbf{x} - \overline{\mathbf{x}}\mathbf{1}^\top)) = p_\omega(h|\mathbf{x} - \overline{\mathbf{x}}\mathbf{1}^\top)$. We then rewrite the expectation with respect to $h$:

$$
\begin{aligned}
\phi_{\theta,\omega}(g' \cdot \mathbf{x}) &= \mathbb{E}_{p_\omega(h|\mathbf{x} - \overline{\mathbf{x}}\mathbf{1}^\top)}\left[ g' \cdot \overline{\mathbf{x}}\mathbf{1}^\top + g'h \cdot f_\theta((g'h)^{-1}g' \cdot (\mathbf{x} - \overline{\mathbf{x}}\mathbf{1}^\top)) \right] + \mathbf{t}_{g'}\mathbf{1}^\top \\
&= \mathbb{E}_{p_\omega(h|\mathbf{x} - \overline{\mathbf{x}}\mathbf{1}^\top)}\left[ g' \cdot \overline{\mathbf{x}}\mathbf{1}^\top + g'h \cdot f_\theta(h^{-1} \cdot (\mathbf{x} - \overline{\mathbf{x}}\mathbf{1}^\top)) \right] + \mathbf{t}_{g'}\mathbf{1}^\top \\
&= \mathbf{Q}_{g'}^+\mathbb{E}_{p_\omega(h|\mathbf{x} - \overline{\mathbf{x}}\mathbf{1}^\top)}\left[ \overline{\mathbf{x}}\mathbf{1}^\top + h \cdot f_\theta(h^{-1} \cdot (\mathbf{x} - \overline{\mathbf{x}}\mathbf{1}^\top)) \right] + \mathbf{t}_{g'}\mathbf{1}^\top \\
&= \mathbf{Q}_{g'}^+\phi_{\theta,\omega}(\mathbf{x}) + \mathbf{t}_{g'}\mathbf{1}^\top \\
&= g' \cdot \phi_{\theta,\omega}(\mathbf{x}),
\end{aligned}
\tag{39}
$$

showing the $\mathrm{SE}(n)$ equivariance of $\phi_{\theta,\omega}$. $\qquad\square$

**Product Group $H \times K$**    For the product group $H \times K$, we assume that the base representation for each element $g = (h, k)$ is given as a pair of representations $\rho(g) = (\rho(h), \rho(k))$. Without loss of generality, we further assume that the representation $\rho(g)$ can be expressed as the Kronecker product $\rho(g) = \rho(h) \otimes \rho(k)$ that acts on flattened data $\mathrm{vec}(\mathbf{x})$ as $\mathbf{x} \mapsto \mathrm{vec}^{-1}(\rho(g)\mathrm{vec}(\mathbf{x}))$. This follows the standard approach in equivariant deep learning [30, 57] that deals with composite representations using direct sum and tensor products of base group representations.

Above approach applies to many practical product groups, including sets and graphs with Euclidean attributes ($\mathrm{S}_n \times \mathrm{O}(d)/\mathrm{SO}(d)$[6]) and sets of symmetric elements ($\mathrm{S}_n \times H$) in general [59]. For

---
[6]This is after handling the translation component of the Euclidean group $\mathrm{E}(d)/\mathrm{SE}(d)$ as in Eq. (29).

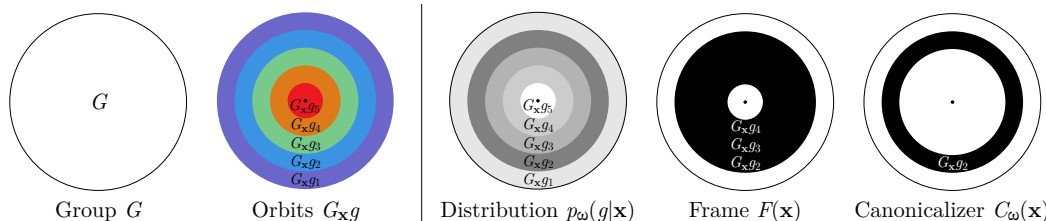

Figure 3: Visual illustration of the symmetrization methods based on probabilities assigned upon the partitioning of the group $G$ into orbits $G_{\mathbf{x}}g$. Note that, while we use concentric circles of different perimeters to illustrate each orbit, all orbits actually have an identical cardinality $|G_{\mathbf{x}}g| = |G_{\mathbf{x}}|$.

example, for the group $\mathrm{S}_n \times \mathrm{O}(d)$ on data $\mathbf{x} \in \mathbb{R}^{n \times d}$, an element $g = (h, k)$ has representation $\rho(g) = \rho(h) \otimes \rho(k) \in \mathbb{R}^{nd \times nd}$ combined from permutation $\rho(h) \in \mathbb{R}^{n \times n}$ and rotation $\rho(k) \in \mathbb{R}^{d \times d}$, which acts by $\mathbf{x} \mapsto \mathrm{vec}^{-1}(\rho(g)\mathrm{vec}(\mathbf{x}))$ or more simply $\mathbf{x} \mapsto \rho(h)\mathbf{x}\rho(k)^{\top}$.

Now we recall that the $p_{\omega}(g|\mathbf{x})$ for the product group $H \times K$ is implemented as follows:

1. Sample noise $\boldsymbol{\epsilon} \in \mathcal{E}$ from i.i.d. normal $\mathcal{N}(0, \eta^2)$ such that $p(\boldsymbol{\epsilon})$ is invariant under representations of $H$ and $K$ that satisfy $|\det \rho'(h)| = 1$ and $|\det \rho'(k)| = 1$, respectively. For example, for $\mathrm{S}_n \times \mathrm{O}(d)$, the noise $\boldsymbol{\epsilon} \in \mathbb{R}^{n \times d}$ that follows i.i.d. normal $\mathcal{N}(0, \eta^2)$ is invariant under base representations of both $\mathrm{S}_n$ and $\mathrm{O}(d)$ which are orthogonal.

2. Use a $H \times K$ equivariant neural network to obtain features $(\mathbf{x}, \boldsymbol{\epsilon}) \mapsto (\mathbf{Z}_H, \mathbf{Z}_K)$ where $\mathbf{Z}_H$ is $K$ invariant and $\mathbf{Z}_K$ is $H$ invariant. For example, for $\mathrm{S}_n \times \mathrm{O}(d)$, we expect node-level scalar features $\mathbf{Z}_{\mathrm{S}_n} \in \mathbb{R}^n$ to be $\mathrm{O}(d)$ invariant and $d$ global rotary features $\mathbf{Z}_{\mathrm{O}(d)} \in \mathbb{R}^{d \times d}$ to be $\mathrm{S}_n$ invariant.

3. Apply postprocessing for $H$ and $K$ groups onto $\mathbf{Z}_H$ and $\mathbf{Z}_K$ respectively to obtain representations $\mathbf{Z}_H \mapsto \rho(h)$ and $\mathbf{Z}_K \mapsto \rho(k)$ of $H$ and $K$ groups respectively. For example, for $\mathrm{S}_n \times \mathrm{O}(d)$, we use argsort in Eq. (23) to obtain $\mathbf{Z}_{\mathrm{S}_n} \mapsto \rho(h)$ and Gram-Schmidt process to obtain $\mathbf{Z}_{\mathrm{O}(d)} \mapsto \rho(k)$.

4. Combine the representations $\rho(g) = (\rho(h), \rho(k))$ to obtain a representation for the $H \times K$ group.

We now show the following:

**Proposition 8.** *The proposed distribution $p_{\omega}(g|\mathbf{x})$ for the product group $H \times K$ is equivariant.*

*Proof.* By assumption, $p(\boldsymbol{\epsilon})$ is invariant under representations of $H$ and $K$ that satisfy $|\det \rho'(h)| = 1$ and $|\det \rho'(k)| = 1$, respectively. This implies $H \times K$ invariance as well, since $p(\boldsymbol{\epsilon}) = p(h \cdot \boldsymbol{\epsilon}) = p(k \cdot \boldsymbol{\epsilon})$ for all $\boldsymbol{\epsilon} \in \mathcal{E}, h \in H, k \in K$ gives $p(k \cdot h \cdot \boldsymbol{\epsilon}) = p(k \cdot (h \cdot \boldsymbol{\epsilon})) = p(h \cdot \boldsymbol{\epsilon}) = p(\boldsymbol{\epsilon})$, and Kronecker product of matrices of determinant 1 gives a matrix of determinant 1. Furthermore, the map $(\mathbf{x}, \boldsymbol{\epsilon}) \mapsto (\rho(h), \rho(k)) = \rho(g)$ is overall $H \times K$ equivariant, since an input transformed with $g' = (h', k')$ is first mapped by the equivariant neural network as $(g' \cdot \mathbf{x}, g' \cdot \boldsymbol{\epsilon}) \mapsto (h' \cdot \mathbf{Z}_H, k' \cdot \mathbf{Z}_K)$, then postprocessed as $(h' \cdot \mathbf{Z}_H, k' \cdot \mathbf{Z}_K) \mapsto (\rho(h')\rho(h), \rho(k')\rho(k)) = (\rho(h'), \rho(k')) \cdot (\rho(h), \rho(k)) = \rho(g')\rho(g)$. Combining the above, by Theorem 3, the distribution $p_{\omega}(g|\mathbf{x})$ is $H \times K$ equivariant. $\square$

### A.1.5    Proof of Proposition 1 and Proposition 2 (Section 2.4)

Before proceeding to proofs, we recall that the stabilizer subgroup $G_{\mathbf{x}}$ of a group $G$ for $\mathbf{x}$ is defined as $\{g' \in G : g' \cdot \mathbf{x} = \mathbf{x}\}$ and acts on a given group element $g \in G$ through left multiplication $g \mapsto g'g$. For some $g \in G$, by $G_{\mathbf{x}}g$ we denote its orbit under the action by $G_{\mathbf{x}}$, *i.e.*, the set of elements in $G$ to which $g$ can be moved by the action of elements $g' \in G_{\mathbf{x}}$. Importantly, we can show the following:

**Property 1.** *Any group $G$ is a union of disjoint orbits $G_{\mathbf{x}}g$ of equal cardinality.*

*Proof.* Let us consider the equivalence relation $\sim$ on $G$ induced by the action of the stabilizer $G_{\mathbf{x}}$, defined as $g \sim h \iff h \in G_{\mathbf{x}}g$. The orbits $G_{\mathbf{x}}g$ are the equivalence classes under this relation, and the set of all orbits of $G$ under the action of $G_{\mathbf{x}}$ forms a partition of $G$ (*i.e.*, the quotient $G/G_{\mathbf{x}}$). Furthermore, since $G_{\mathbf{x}} \leq G$ and right multiplication by some $g \in G$ is a faithful action of $G$ on itself, we have $|G_{\mathbf{x}}g| = |G_{\mathbf{x}}|$ for all $g \in G$, which shows that all orbits $G_{\mathbf{x}}g$ have equal cardinality. $\square$

The partition of group $G$ into disjoint orbits $G_{\mathbf{x}}g$ is illustrated in the first and second panel of Figure 3. We now show the following:

**Property 2.** *$G$ equivariant $p_\omega(g|\mathbf{x})$ assigns identical probability to all elements on each orbit $G_{\mathbf{x}}g$.*

*Proof.* With equivariance, we have $p_\omega(g|\mathbf{x}) = p_\omega(g'g|g'\cdot\mathbf{x})$. Since $g'\cdot\mathbf{x} = \mathbf{x}$ for all $g' \in G_{\mathbf{x}}$, we have $p_\omega(g|\mathbf{x}) = p_\omega(g'g|\mathbf{x})$ for all $g' \in G_{\mathbf{x}}$; all elements on orbit $G_{\mathbf{x}}g$ have an identical probability. $\quad\square$

Property 2 characterizes probability distributions over $G$ that can be expressed with $p_\omega(g|\mathbf{x})$, which we illustrate in the third panel of Figure 3. Intuitively, $p_\omega(g|\mathbf{x})$ assigns constant probability densities over each of the orbit $G_{\mathbf{x}}g$ that partitions $G$ as shown in Property 1. We now prove Proposition 1 and Proposition 2 by showing that $p_\omega(g|\mathbf{x})$ can become frame and canonicalizer as special cases:

**Proposition 1.** *Probabilistic symmetrization with $G$ equivariant distribution $p_\omega(g|\mathbf{x})$ can become frame averaging [74] by assigning uniform density to a set of orbits $G_{\mathbf{x}}g$ for some group elements g.*

*Proof.* A frame is defined as a set-valued function $F : \mathcal{X} \to 2^G \setminus \emptyset$ that satisfies $G$ equivariance $F(g \cdot \mathbf{x}) = gF(\mathbf{x})$ [74]. For some frame $F$, frame averaging is defined as follows:

$$\frac{1}{|F(\mathbf{x})|} \sum_{g \in F(\mathbf{x})} \left[ g \cdot f_\theta(g^{-1} \cdot \mathbf{x}) \right], \tag{40}$$

which can be equivalently written as the below expectation:

$$\mathbb{E}_{g \sim \mathrm{Unif}(F(\mathbf{x}))} \left[ g \cdot f_\theta(g^{-1} \cdot \mathbf{x}) \right]. \tag{41}$$

From Theorem 3 of [74], we have that $F(\mathbf{x})$ is a disjoint union of equal size orbits $G_{\mathbf{x}}g$. Therefore, $\mathrm{Unif}(F(\mathbf{x}))$ is a uniform probability distribution over the union of the orbits. This can be expressed by a $G$ equivariant distribution $p_\omega(g|\mathbf{x})$ by assigning identical probability over all orbits in the frame $F$ and zero probability to all orbits not in the frame (illustrated in the fourth panel of Figure 3). Therefore, probabilistic symmetrization can become frame averaging. $\quad\square$

**Proposition 2.** *Probabilistic symmetrization with $G$ equivariant distribution $p_\omega(g|\mathbf{x})$ can become canonicalization [41] by assigning uniform density to a single orbit $G_{\mathbf{x}}g$ of some group element g.*

*Proof.* A canonicalizer is defined as a (possibly stochastic) parameterized map $C_\omega : \mathcal{X} \to G$ that satisfies relaxed $G$ equivariance $C_\omega(g \cdot \mathbf{x}) = gg'C_\omega(\mathbf{x})$ for some $g' \in G_{\mathbf{x}}$ [41]. For some canonicalizer $C_\omega$, canonicalization is defined as follows:

$$g \cdot f_\theta(g^{-1} \cdot \mathbf{x}), \quad g = C_\omega(\mathbf{x}). \tag{42}$$

From relaxed $G$ equivariance, we have $C_\omega(\mathbf{x}) = g'C_\omega(\mathbf{x})$ for some $g' \in G_{\mathbf{x}}$. A valid choice for the canonicalizer $C_\omega$ is a stochastic map that samples from the uniform distribution over a frame $C_\omega(\mathbf{x}) \sim \mathrm{Unif}(F_\omega(\mathbf{x}))$ where the frame is assumed to always provide a single orbit $F_\omega(\mathbf{x}) = G_{\mathbf{x}}g$. In this case, canonicalization is equivalent to a 1-sample estimation of the below expectation:

$$\mathbb{E}_{g \sim \mathrm{Unif}(F_\omega(\mathbf{x}))} \left[ g \cdot f_\theta(g^{-1} \cdot \mathbf{x}) \right]. \tag{43}$$

Furthermore, uniform distribution over the single-orbit frame $\mathrm{Unif}(F_\omega(\mathbf{x}))$ can be expressed by a $G$ equivariant distribution $p_\omega(g|\mathbf{x})$ by assigning nonzero probability to the single orbit $G_{\mathbf{x}}g$ and assigning zero probability to the rest (illustrated in the last panel of Figure 3). Therefore, probabilistic symmetrization can become canonicalization. $\quad\square$

## A.2 Extended Related Work (Continued from Section 2.4)

Our work draws inspiration from an extensive array of prior research, ranging from equivariant architectures and symmetrization to general-purpose deep learning with transformers. This section outlines a comprehensive review of these fields, spotlighting ideas specifically relevant to our work.

**Equivariant Architectures**    Equivariant architectures, defined by the group equivariance of their building blocks, have been a prominent approach for equivariant deep learning [12, 10]. These architectures have been primarily developed for data types associated with permutation and Euclidean group symmetries, including images [18, 19], sets, graphs, and hypergraphs [5, 57, 8], and geometric graphs [23, 84, 91]. Additionally, they have been extended to more general data types under arbitrary finite group [78] and matrix group symmetries [30]. However, they face challenges such as limited expressive power [101, 56, 64, 105, 40] and architectural issues like over-smoothing [70, 14, 69] and over-squashing [92] in graph neural networks. Our work aims to develop an equivariant deep learning approach that relies less on equivariant architectures, to circumvent these limitations and enhance parameter sharing and transfer across varying group symmetries.

**Symmetrization**    Our approach is an instance of symmetrization for equivariant deep learning which aims to achieve group equivariance using base models with unconstrained architectures. This is in general accomplished by averaging over specific group transformations of the input and output such that the averaged output exhibits equivariance. This allows us to leverage the expressive power of the base model *e.g.*, achieve universal approximation using an MLP [35, 20] or a transformer [104], and potentially share or transfer parameters across different group symmetries. Existing literature has explored the choices of group transformations and base models for symmetrization. A straightforward approach is to average over the entire group [102], which is suitable for small, finite groups [4, 65, 42, 94] and requires sampling-based estimation for large groups such as permutations [67, 68, 88, 21]. Recent studies have attempted to identify smaller, input-dependent subsets of the group for averaging. Frame averaging [74] employs manually discovered set-values functions called frames, which still demand sampling-based estimation for certain worst-case inputs. Canonicalization [41] utilizes a single group transformation predicted by a neural network, but sacrifices strict equivariance. Our approach jointly achieves equivariance and end-to-end learning by utilizing parameterized, input-conditional equivariant distributions. Furthermore, our approach is one of the first demonstrations of symmetrization for the permutation group in real-world graph recognition task. Concerning the base model, previous work mostly examined small base models like an MLP or partial symmetrization of already equivariant models like GNNs. Few studies have explored symmetrizing pre-trained models for small finite groups [4, 3], and to our knowledge, we are the first to investigate symmetrization of a pre-trained standard transformer for permutation groups or any large group generally.

**Transformer Architectures**    A significant motivation of our work is to combine the powerful scaling and transfer capabilities of the standard transformer architecture [96] with equivariant deep learning. The transformer architecture has driven major breakthroughs in language and vision domains [96, 24, 13, 75], and proven its ability to learn diverse modalities [39, 38] or transfer knowledge across them [85, 54, 82, 25, 71, 52]. Although transformer-style architectures have been developed for symmetric data modalities like sets [51], graphs [103, 45, 43, 50, 66, 63], hypergraphs [17, 44], and geometric graphs [31, 55], they often require specific architectural modifications to achieve equivariance to the given symmetry group, compromising full compatibility with transformer architectures used in language and vision domains. Apart from a few studies on linguistic graph encoding with language models [79], we believe we are the first to propose a general framework that facilitates full compatibility of the standard transformer architecture for learning symmetric data. For example, we have shown that a pre-trained vision transformer could be repurposed to encode graphs.

**Learning Distribution of Data Augmentations**    Since our approach parameterizes a distribution $p_\omega(g|\mathbf{x})$ on a group for symmetrization of form $\phi_{\theta,\omega}(\mathbf{x}) = \mathbb{E}_g[g \cdot f_\theta(g^{-1} \cdot \mathbf{x})]$ and learns it from data, one may find similarity to Augerino [7] and related approaches [77, 81, 95, 93, 80, 37] that learn distributions over data augmentations (*e.g.*, $p_\omega(g)$) for a similar symmetrization. The key difference is that, while these approaches aim to discover underlying (approximate) symmetry constraint from data and searches over a space of different group symmetries, our objective aims to obtain an exact $G$ equivariant symmetrization $\phi_{\theta,\omega}(\mathbf{x})$ given the known symmetry group $G$ of data (*e.g.*, $G = \mathrm{S}_n$ for graphs). Because of this, the symmetrizing distribution has to be designed differently. In our case, we parameterize the distribution $p_\omega(g|\mathbf{x})$ itself to be equivariant to a specific given group $G$, while for augmentation learning approaches, the distribution $p_\omega(g)$ is parameterized for a different purpose of covering a range of different group symmetry constraints and their approximations (*e.g.*, a set of 2D affine transformations [7]). This leads to advantages of our approach if the symmetry group $G$ is known, as **(1)** our approach can learn non-trivial and useful distribution $p_\omega(g|\mathbf{x})$ per input data $\mathbf{x}$ while keeping the symmetrized function $\phi_{\theta,\omega}(\mathbf{x})$ exactly $G$ equivariant, while augmentation

Table 5: Overview of the datasets.

| Dataset | Symmetry | Domain | Task | Feat. (dim) |
|---|---|---|---|---|
| GRAPH8c | | | Graph Separation | |
| EXP | $S_n$ Invariant | Graph Isomorphism | | Adj. (1) |
| EXP-classify | | | Graph Classification | |
| $n$-body | $S_n \times E(3)$ Equivariant | Physics | Position Regression | Pos. (3) + Vel. (3) + Charge (1) |
| PATTERN | $S_n$ Equivariant | Mathematical Modeling | Node Classification | Rand. Node Attr. (3) + Adj. (1) |
| Peptides-func | | | Graph Classification | |
| Peptides-struct | $S_n$ Invariant | Chemistry | Graph Regression | Atom (9) + Bond (3) + Adj. (1) |
| PCQM-Contact | $S_n$ Equivariant | Quantum Chemistry | Link Prediction | Atom (9) + Bond (3) + Adj. (1) |

Table 6: Statistics of the datasets.

| Dataset | Size | Max # Nodes | Average # Nodes | Average # Edges |
|---|---|---|---|---|
| GRAPH8c | 11,117 | 8 | 8 | 28.82 |
| EXP | | | | |
| EXP-classify | 1,200 | 64 | 44.44 | 110.21 |
| $n$-body | 7,000 | 5 | 5 | Fully Connected |
| PATTERN | 14,000 | 188 | 117.47 | 4749.15 |
| Peptides-func | | | | |
| Peptides-struct | 15,535 | 444 | 150.94 | 307.30 |
| PCQM-Contact | 529,434 | 53 | 30.14 | 61.09 |

learning does not guarantee equivariance for a given group in general and often has to reduce to trivial group averaging $p_\omega = \text{Unif}(G)$ to be exactly $G$ equivariant, and **(2)** while augmentation learning has to employ regularization [7] or model selection [37] to prevent collapse to trivial symmetry that is the least constrained and would fit the training data most easily [37], our approach fixes and enforces equivariance for the given symmetry group $G$ by construction, which allows us to use regular maximum likelihood objective for training without the need to address symmetry collapse.

### A.3 Experimental Details (Section 3)

We provide details of the datasets and models used in our experiments in Section 3. The details of the datasets from the original papers [2, 1, 27, 28, 31, 84] can be found in Table 5 and Table 6.

### A.3.1 Implementation Details of $p_\omega$ for Symmetric Group $S_n$ (Section 3.1, 3.3, 3.4)

In all experiments regarding the symmetric group $S_n$, we implement the $S_n$ equivariant distribution $p_\omega(g|\mathbf{x})$, *i.e.*, $q_\omega : (\mathbf{x}, \boldsymbol{\epsilon}) \mapsto \mathbf{P}_g$ as a 3-layer GIN with 64 hidden dimensions [101] that has around 25k parameters. Specifically, given a graph $\mathbf{x}$ with node features $\mathbf{X} \in \mathbb{R}^{n \times d_{\text{in}}}$ and adjacency matrix $\mathbf{A} \in \mathbb{R}^{n \times n}$,[7] we first augment a virtual node [32] which is connected to all nodes to facilitate global interaction while retaining $S_n$ equivariance, as follows:

$$\mathbf{X}' = [\mathbf{X}; \mathbf{v}], \quad \mathbf{A}' = \left[ \begin{array}{cc} \mathbf{A} & \mathbf{1} \\ \mathbf{1}^\top & 0 \end{array} \right], \tag{44}$$

where the feature of the virtual node $\mathbf{v} \in \mathbb{R}^{d_{\text{in}}}$ is a trainable parameter. Then, we prepared the input node features $\mathbf{H} \in \mathbb{R}^{(n+1) \times d_{\text{in}}}$ to the GIN as $\mathbf{H} = \mathbf{X}' + \boldsymbol{\epsilon}$ where the noise $\boldsymbol{\epsilon} \in \mathbb{R}^{(n+1) \times d_{\text{in}}}$ is i.i.d. sampled from $\text{Unif}[0, \eta]$ with scale hyperparameter $\eta$. Then, we employ following 3-layer GIN with 64 hidden dimensions to obtain processed node features $\mathbf{H}' \in \mathbb{R}^{(n+1) \times 1}$:

$$\mathbf{H}' = \text{GINConv}_{64,64,1} \circ \text{GINConv}_{64,64,64} \circ \text{GINConv}_{d_{\text{in}},64,64}(\mathbf{H}), \tag{45}$$

where each $\text{GINConv}_{d_1,d_2,d_3}$ computes below with a two-layer elementwise MLP $: \mathbb{R}^{n \times d_1} \to \mathbb{R}^{n \times d_3}$ with hidden dimension $d_2$, ReLU activation, batch normalization, and trained scalar $e$:

$$\mathbf{H} \mapsto \text{MLP}((\mathbf{A}' + (1 + e)\mathbf{I})\mathbf{H}). \tag{46}$$

---

[7]We do not utilize edge attributes in equivariant distribution $p_\omega$, while we utilize them in base model $f_\theta$.

Then, from the processed node features $\mathbf{H}' \in \mathbb{R}^{(n+1)\times 1}$, we finally obtain the features $\mathbf{Z} \in \mathbb{R}^n$ for postprocessing by discarding the feature of the virtual node. Then, postprocessing into a permutation matrix is done with argsort : $\mathbf{Z} \mapsto \mathbf{P}_g \in \mathbb{R}^{n\times n}$ as in Eq. (8).

**Training**  To backpropagate through $\mathbf{P}_g$ for end-to-end training of $p_\omega(g|\mathbf{x})$, we use straight-through gradient estimator [6] with an approximate permutation matrix $\hat{\mathbf{P}}_g \approx \mathbf{P}_g$.[8] For this, we first apply L2 normalization $\mathbf{Z} \mapsto \bar{\mathbf{Z}}$ and use the below differentiable relaxation of the argsort operator [61, 33, 98]:

$$\hat{\mathbf{P}}_g = S(-|\bar{\mathbf{Z}}\mathbf{1}^\top - \mathbf{1}\text{sort}(\bar{\mathbf{Z}})^\top|/\tau), \tag{47}$$

where $S(\cdot/\tau)$ is Sinkhorn operator [61] with temperature hyperparameter $\tau \in \mathbb{R}_+$ that performs elementwise exponential followed by iterative normalization of rows and columns. Following [61], we use 20 Sinkhorn iterations which worked robustly in all our experiments. For the correctness of straight-through gradients, it is desired that $\hat{\mathbf{P}}_g$ closely approximates the real permutation matrix $\mathbf{P}_g$ during training. For this, we choose the temperature $\tau$ to be small, $0.01$ in general, and following prior work [98], employ a regularizer on the mean of row- and column-wise entropy of $\hat{\mathbf{P}}_g$ with a strength of $0.1$ in all experiments. The $\mathrm{S}_n$ equivariance of the relaxed argsort $\mathbf{Z} \mapsto \hat{\mathbf{P}}_g$ can be shown in a similar way to Proposition 3 from the fact that elementwise subtraction, absolute, scaling by $-1/\tau$, exponential, and iterative normalization of rows and columns all commute with $\mathbf{P}_{g'} \in \mathrm{S}_n$.

### A.3.2  Implementation Details of $p_\omega$ for Product Group $\mathrm{S}_n \times \mathrm{E}(3)$ (Section 3.2)

In our $n$-body experiment on the product group $\mathrm{S}_n \times \mathrm{E}(3)$, we implement the $\mathrm{S}_n \times \mathrm{O}(3)$ equivariant distribution $p_\omega(g|\mathbf{x} - \bar{\mathbf{x}}\mathbf{1}^\top)$, i.e., $q_\omega : (\mathbf{x} - \bar{\mathbf{x}}\mathbf{1}^\top, \boldsymbol{\epsilon}) \mapsto (\mathbf{P}_g, \mathbf{Q}_g)$ based on a 2-layer Vector Neurons version of DGCNN with 96 hidden dimensions [23] that has around 7k parameters. Due to the architecture's complexity, we focus on describing input and output of the network and postprocessing, and guide the readers to the original paper [23] for further architectural details. In a high-level, the Vector Neurons receives position $\mathbf{P} \in \mathbb{R}^{n\times 3}$ and velocity $\mathbf{V} \in \mathbb{R}^{n\times 3}$ of the zero-centered input $\mathbf{x} - \bar{\mathbf{x}}\mathbf{1}^\top$ with noises $\boldsymbol{\epsilon}_1, \boldsymbol{\epsilon}_2 \in \mathbb{R}^{n\times 3}$ i.i.d. sampled from normal $\mathcal{N}(0, \eta^2)$ with scale hyperparameter $\eta$, and produces features $\mathbf{H}_{\mathrm{S}_n} \in \mathbb{R}^{n\times 3\times d_1}$ and $\mathbf{H}_{\mathrm{O}(3)} \in \mathbb{R}^{n\times 3\times d_2}$ with $d_1 = 1$ and $d_2 = 3$ as follows:

$$\mathbf{H}_{\mathrm{S}_n}, \mathbf{H}_{\mathrm{O}(3)} = \text{VN-DGCNN}(\mathbf{P} + \boldsymbol{\epsilon}_1, \mathbf{V} + \boldsymbol{\epsilon}_2). \tag{48}$$

Then, we apply $\mathrm{O}(3)$ invariant pooling on $\mathbf{H}_{\mathrm{S}_n}$ and $\mathrm{S}_n$ invariant pooling on $\mathbf{H}_{\mathrm{O}(3)}$, both supported as a part of [23], to obtain features for postprocessing $\mathbf{Z}_{\mathrm{S}_n} \in \mathbb{R}^{n\times 1}$ and $\mathbf{Z}_{\mathrm{O}(3)} \in \mathbb{R}^{3\times 3}$, respectively:

$$\mathbf{Z}_{\mathrm{S}_n} = \text{Pool}_{\mathrm{O}(3)}(\mathbf{H}_{\mathrm{S}_n}), \quad \mathbf{Z}_{\mathrm{O}(3)} = \text{Pool}_{\mathrm{S}_n}(\mathbf{H}_{\mathrm{O}(3)}). \tag{49}$$

Then, postprocessing with argsort : $\mathbf{Z}_{\mathrm{S}_n} \mapsto \mathbf{P}_g \in \mathbb{R}^{n\times n}$ and Gram-Schmidt orthogonalization $\mathbf{Z}_{\mathrm{O}(3)} \mapsto \mathbf{Q}_g \in \mathbb{R}^{3\times 3}$ is performed identically as described in the main text (Section 2.2). For the straight-through gradient estimation of the argsort operator, we use relaxed argsort described in Appendix A.3.1, with the only difference of using the temperature $\tau = 0.1$.

### A.3.3  Graph Isomorphism Learning with MLP (Section 3.1)

**Base Model $f_\theta$**  For EXP and EXP-classify, the model is given adjacency matrix $\mathbf{A} \in \mathbb{R}^{64\times 64}$ and binary node features $\mathbf{X} \in \mathbb{R}^{64}$ which are zero-padded to maximal number of nodes $64$. For GRAPH8c, the input graphs are all of size $8$ without node features, and the model is given adjacency matrix $\mathbf{A} \in \mathbb{R}^{8\times 8}$. For EXP-classify, the prediction target is a scalar binary classification logit.

For the base model for EXP-classify, we use a 5-layer MLP $f_\theta : \mathbb{R}^{64\times 64+64} \to \mathbb{R}$ on flattened and concatenated adjacency matrix and node features, with an identical architecture to other symmetrization baselines (MLP-GA and MLP-FA [74]) as in below:

$$f_\theta = \text{FC}_{1,10} \circ \text{FC}_{10,2048} \circ \text{FC}_{2048,4096} \circ \text{FC}_{4096,2048} \circ \text{FC}_{2048,4160}, \tag{50}$$

where $\text{FC}_{d_2,d_1} : \mathbb{R}^{d_1} \to \mathbb{R}^{d_2}$ denotes a fully-connected layer and ReLU activation is omitted. For EXP, we drop the last layer to obtain 10-dimensional output. For GRAPH8c, we use the following architecture $f_\theta : \mathbb{R}^{8\times 8} \to \mathbb{R}^{10}$ that takes flattened adjacency to produce 10-dimensional output [74]:

$$f_\theta = \text{FC}_{10,64} \circ \text{FC}_{64,128} \circ \text{FC}_{128,64}. \tag{51}$$

---

[8]In PyTorch [72], one can simply replace $\mathbf{P}_g$ with $(\mathbf{P}_g - \hat{\mathbf{P}}_g)$.detach() $+ \hat{\mathbf{P}}_g$ during forward passes.

**Training**  For EXP-classify, we train our models with binary cross-entropy loss using Adam optimizer [46] with batch size 100 and learning rate 1e-3 for 2,000 epochs, which takes around 30 minutes on a single RTX 3090 GPU with 24GB using PyTorch [72]. We additionally apply 200 epochs of linear learning rate warm-up and gradient norm clipping at 0.1, which we found helpful for stabilizing the training. For the equivariant distribution $p_\omega$, we use noise scale $\eta = 1$. Since EXP and GRAPH8c concern randomly initialized models, we do not train the models for these tasks.

### A.3.4  Particle Dynamics Learning with Transformer (Section 3.2)

**Base Model $f_\theta$**  The model is given zero-centered position $\mathbf{P} \in \mathbb{R}^{5 \times 3}$ and velocity $\mathbf{V} \in \mathbb{R}^{5 \times 3}$ of 5 particles at a time point with pairwise charge difference $\mathbf{C} \in \mathbb{R}^{5 \times 5}$ and squared distance $\mathbf{D} \in \mathbb{R}^{5 \times 5}$. We set the prediction target as difference of position $\Delta \mathbf{P} \in \mathbb{R}^{5 \times 3}$ after a certain time.

For the base model, we use a 8-layer transformer encoder $f_\theta : \mathbb{R}^{25 \times 8} \to \mathbb{R}^{25 \times 3}$ that operates on sequences of length 25 with dimension 8. At each prediction, we first organize the input into a single tensor $\in \mathbb{R}^{5 \times 5 \times 8}$ by placing $\mathbf{P}$ and $\mathbf{V}$ on the diagonals of $\mathbf{C}$ and $\mathbf{D}$, and then turn the tensor into a sequence of 25 tokens $\in \mathbb{R}^{25 \times 8}$ by flattening the first two axes. Analogously, we organize the output of the model into a tensor $\in \mathbb{R}^{5 \times 5 \times 3}$ and take the diagonal entries as the predictions. For the transformer, we use the standard implementation provided in PyTorch [72, 96], with 64 hidden dimensions, 4 attention heads, GELU activation [34] in feedforward networks, PreLN [100], learnable 1D positional encoding, and an MLP prediction head with 1 hidden layer. The model has around 208k trainable parameters, around $2.3\times$ compared to the GNN backbones of E(3) symmetrization baselines in the benchmark (GNN-FA and GNN-Canonical.) with 92k parameters.

**Training**  We train our models with MSE loss using Adam optimizer [46] with batch size 100 and learning rate 1e-3 for 10,000 epochs, which takes around 8.5 hours on a single RTX 3090 GPU with 24GB using PyTorch [72]. We use weight decay with strength 1e-12 and dropout on the distribution $p_\omega$ with probability 0.08. For the equivariant distribution $p_\omega$, we use noise scale $\eta = 1$.

### A.3.5  Graph Pattern Recognition with Vision Transformer (Section 3.3)

**Base Model $f_\theta$**  The model is given adjacency matrix $\mathbf{A} \in \mathbb{R}^{188 \times 188}$ and node features $\mathbf{X} \in \mathbb{R}^{188 \times 3}$ zero-padded to maximal 188 nodes. The prediction target is node classification logits $\mathbf{Y} \in \mathbb{R}^{188 \times 2}$.

For the base model, we use a transformer with an identical architecture to ViT-Base [26] that operates on $224 \times 224$ images with $16 \times 16$ patch, using configuration from HuggingFace [99] model hub. We first remove the input patch projection and output head layers, which gives us a backbone transformer : $\mathbb{R}^{(14 \times 14) \times 768} \to \mathbb{R}^{(14 \times 14) \times 768}$ on sequences of $(224/16) \times (224/16) = 14 \times 14$ tokens. Then, we use the following as the base model $f_\theta : (\mathbf{A}, \mathbf{X}) \mapsto \mathbf{Y}$:

$$f_\theta(\mathbf{A}, \mathbf{X}) = \text{detokenize}\left(\text{transformer}\left(\text{tokenize}(\mathbf{A}, \mathbf{X})\right)\right), \tag{52}$$

where, for tokenize : $\mathbb{R}^{188 \times 188 \times 1} \times \mathbb{R}^{188 \times 3} \to \mathbb{R}^{(14 \times 14) \times 768}$ we organize the input into a single tensor $\in \mathbb{R}^{188 \times 188 \times 4}$ by placing $\mathbf{X}$ on the diagonals of $\mathbf{A}$ and apply 2D convolution with kernel size and stride 14, and for detokenize : $\mathbb{R}^{(14 \times 14) \times 768} \to \mathbb{R}^{188 \times 2}$ we apply transposed 2D convolution with kernel size and stride 14 to obtain a tensor $\in \mathbb{R}^{188 \times 188 \times 2}$ and take its diagonal entries as output.

**Training**  We train our models with binary cross-entropy loss weighted inversely by class size [27] using AdamW [53] optimizer with batch size 128, learning rate 1e-5, and weight decay 0.01. We train the models for 25k steps under learning rate warm-up for 5k steps then linear decay to 0 with early stopping based on validation loss, which usually takes less than 5 hours on 8 RTX 3090 GPUs with 24GB using PyTorch Lightning [29]. For the equivariant distribution $p_\omega$ we use noise scale $\eta = 1$ and dropout with probability 0.1. For probabilistic symmetrization that involves sampling-based estimation, we use sample size 1 for training. For group averaging, sample size 1 for training led to optimization challenges, and therefore we use sample size 10 for training which yielded better results.

### A.3.6  Real-World Graph Learning with Vision Transformer (Section 3.4)

**Base Model $f_\theta$**  For Peptides-func/struct, the model is given adjacency matrix $\mathbf{A} \in \mathbb{R}^{444 \times 444}$, node features $\mathbf{X} \in \mathbb{R}^{444 \times 64}$, and edge features $\mathbf{E} \in \mathbb{R}^{444 \times 444 \times 7}$, zero-padded to maximal 444 nodes. The prediction target is binary classification logits $\mathbf{Y} \in \mathbb{R}^{10}$ for Peptides-func, and regression targets

Table 7: Supplementary results for $S_n$ invariant graph separation with $S_n$ symmetrized GIN-ID base function. Baseline scores for GIN-ID-GA and GIN-ID-FA are taken from [74].

| method | arch. | sym. | GRAPH8c ↓ | EXP ↓ | EXP-classify ↑ |
|---|---|---|---|---|---|
| GIN-ID-GA | - | $S_n$ | 0 | 0 | 50% |
| GIN-ID-FA | - | $S_n$ | 0 | 0 | **100%** |
| GIN-ID-Canonical. | - | $S_n$ | 0 | 0 | 84% |
| GIN-ID-PS (Ours) | - | $S_n$ | 0 | 0 | **100%** |

Table 8: Supplementary results for $S_n \times E(3)$ equivariant $n$-body with $E(3)$ symmetrized GNN base function. Baseline scores for GNN-FA and GNN-Canonical. are from [74] and [41], respectively.

| method | arch. | sym. | Position MSE ↓ |
|---|---|---|---|
| GNN-FA | $S_n$ | $E(3)$ | 0.0057 |
| GNN-Canonical. | $S_n$ | $E(3)$ | 0.0043 |
| GNN-Canonical. (Reproduced) | $S_n$ | $E(3)$ | 0.00457 |
| GNN-GA | $S_n$ | $E(3)$ | $0.00408 \pm 0.00002$ |
| GNN-PS (Ours) | $S_n$ | $E(3)$ | $\mathbf{0.00386 \pm 0.00001}$ |

$\mathbf{Y} \in \mathbb{R}^{11}$ for Peptides-struct. For PCQM-Contact, the model is given adjacency matrix $\mathbf{A} \in \mathbb{R}^{53 \times 53}$, node features $\mathbf{X} \in \mathbb{R}^{53 \times 68}$, and edge features $\mathbf{E} \in \mathbb{R}^{53 \times 53 \times 6}$, zero-padded to maximal 53 nodes. The prediction target is binary edge classification logit $\mathbf{Y} \in \mathbb{R}^{53 \times 53 \times 1}$.

For the base model, we use a transformer with an identical architecture to ViT-Base that operates on $14 \times 14$ tokens, same as in Appendix A.3.5. For Peptides-func and Peptides-struct, we use the following as the base model $f_\theta : (\mathbf{A}, \mathbf{X}, \mathbf{E}) \mapsto \mathbf{Y}$:

$$f_\theta(\mathbf{A}, \mathbf{X}, \mathbf{E}) = \text{detokenize}_{\texttt{[cls]}} \left( \text{transformer} \left( \text{tokenize}_{2D}(\mathbf{A}, \mathbf{E}) + \text{tokenize}_{1D}(\mathbf{X}) \right) \right), \quad (53)$$

where $\text{tokenize}_{2D} : \mathbb{R}^{444 \times 444 \times (1+7)} \rightarrow \mathbb{R}^{(14 \times 14) \times 768}$ is 2D convolution with kernel size and stride 32, $\text{tokenize}_{1D} : \mathbb{R}^{444 \times 64} \rightarrow \mathbb{R}^{196 \times 768}$ is 1D convolution with kernel size and stride 3, and $\text{detokenize}_{\texttt{[cls]}}$ performs linear projection of the global $\texttt{[cls]}$ token [26] to the target dimensionality. For PCQM-Contact, we use the following as base model $f_\theta : (\mathbf{A}, \mathbf{X}, \mathbf{E}) \mapsto \mathbf{Y}$:

$$f_\theta(\mathbf{A}, \mathbf{X}, \mathbf{E}) = \text{detokenize}_{2D} \left( \text{transformer} \left( \text{tokenize}_{2D}(\mathbf{A}, \mathbf{E}) + \text{tokenize}_{1D}(\mathbf{X}) \right) \right), \quad (54)$$

where $\text{tokenize}_{2D} : \mathbb{R}^{53 \times 53 \times (1+6)} \rightarrow \mathbb{R}^{(14 \times 14) \times 768}$ is 2D convolution with kernel size and stride 4, $\text{tokenize}_{1D} : \mathbb{R}^{53 \times 64} \rightarrow \mathbb{R}^{196 \times 768}$ is 1D convolution with kernel size and stride 1, and $\text{detokenize}_{2D} : \mathbb{R}^{(14 \times 14) \times 768} \rightarrow \mathbb{R}^{53 \times 53 \times 1}$ is transposed 2D convolution with kernel size and stride 4.

**Training**  We train our models with cross-entropy for classification and L1 loss for regression using AdamW [53] optimizer with batch size 128, learning rate 1e-5 except for PCQM-Contact where we use 5e-5, and weight decay 0.01. We train the models for 50k steps under learning rate warm-up for 5k steps then linear decay to 0 with early stopping based on validation loss, which usually takes less than 12 hours on 8 RTX 3090 GPUs with 24GB using PyTorch Lightning [29]. For the equivariant distribution $p_\omega$, we use noise scale $\eta = 1$, and use dropout with probability 0.1 except for PCQM-Contact where we do not use dropout. We use 10 samples for estimation during training.

## A.4  Supplementary Experiments (Continued from Section 3)

In this section, we present additional experimental results that supplement the experiments in Section 3 but could not be included in the main text due to space constraints.

### A.4.1  Graph Isomorphism Learning (Section 3.1)

In our experiments on graph isomorphism learning in Section 3.1, we mainly experimented for $S_n$ symmetrization of an MLP. Here, we provide supplementary results on $S_n$ symmetrization of a GIN base model with node identifiers, following [74]. The results can be found in Table 7. In accordance with Section 3.1, our approach successfully performs $S_n$ symmetrization of GIN-ID.

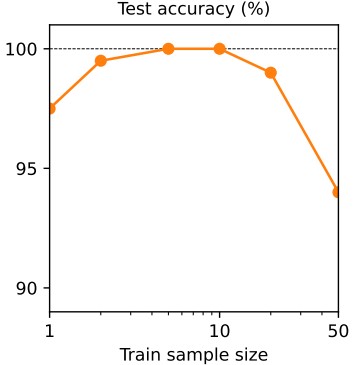

Figure 4: Test accuracy of MLP $f_\theta$ symmetrized by equivariant distribution $p_\omega(g|\mathbf{x})$ trained on EXP-classify dataset across a range of training sample sizes. Inference sample size is set to 10.

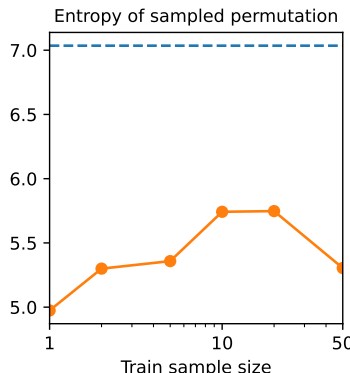

Figure 5: Row- and column-wise entropy of aggregated permutation matrices $\mathbf{P}_g \sim p_\omega(g|\mathbf{x})$ after trained on EXP-classify dataset across a range of training sample sizes. Dashed line indicates entropy measured with random permutation matrices from $\mathrm{Unif}(G)$.

### A.4.2 Particle Dynamics Learning (Section 3.2)

In our experiments on $n$-body dataset in Section 3.2, we experimented for $\mathrm{S}_n \times \mathrm{E}(3)$ symmetrization using a 1D sequence transformer architecture which has $2.3\times$ parameters compared to baselines. To provide parameter-matched comparison against baselines in literature, we apply our approach for $\mathrm{E}(3)$ symmetrization of $\mathrm{S}_n$ equivariant GNN base model that is widely used in literature [74, 41]. We faithfully follow [74, 41] on the experimental setups including training hyperparameters and the configuration of GNN base model, and only add $\mathrm{E}(3)$ equivariant distribution $p_\omega(g|\mathbf{x} - \bar{\mathbf{x}}\mathbf{1}^\top)$, *i.e.*, $q_\omega : (\mathbf{x} - \bar{\mathbf{x}}\mathbf{1}^\top, \boldsymbol{\epsilon}) \mapsto \mathbf{Q}_g$ by utilizing the 2-layer Vector Neurons architecture described in Appendix A.3.2 using only its $\mathrm{O}(3)$ prediction head. We use 20 samples for training and testing. The results can be found in Table 8. In accordance with the results in Section 3.2, our approach outperforms other symmetrization approaches and achieves a new state-of-the-art of 0.00386 MSE.

### A.4.3 Effect of Sample Size on Training and Inference

In this section, we provide additional analysis on how the sample size for estimation of symmetrized function (Eq. (4)) affects training and inference. We use the experimental setup of EXP-classify (Section 3.1; $\mathrm{S}_n$ invariance) and analyze the behavior of MLP-PS with identical initialization and hyperparameters, only controlling sample sizes $\in \{1, 2, 5, 10, 20, 50\}$ for training. Specifically, we analyze **(1)** variance of permutation matrices $\mathbf{P}_g \sim p_\omega(g|\mathbf{x})$ measured indirectly by the entropy of their aggregation $\bar{\mathbf{P}} = \sum \mathbf{P}_g / N$ as in Section 3.1, **(2)** sample variance of the unbiased estimator $g \cdot f_\theta(g^{-1} \cdot \mathbf{x})$ of the symmetrized function $\phi_{\theta,\omega}(\mathbf{x})$ as in Eq. (4), and **(3)** sample mean and variance

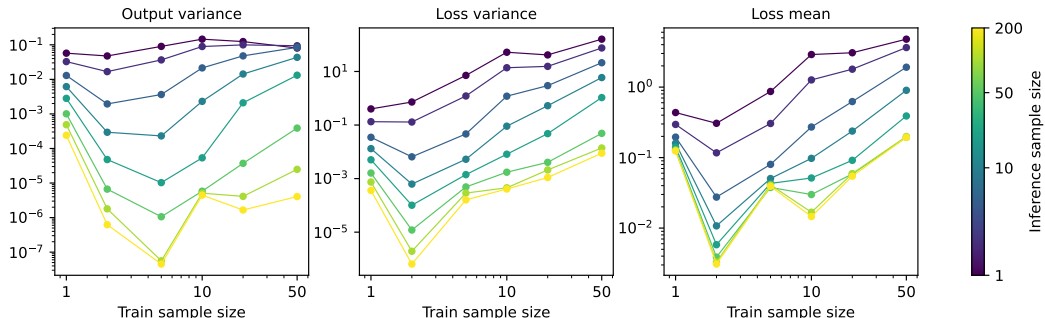

Figure 6: Variance of estimation of MLP $f_\theta$ symmetrized by equivariant distribution $p_\omega(g|\mathbf{x})$ and trained on EXP-classify dataset for a range of training and inference sample sizes.

of the estimated task loss $\mathcal{L}(\mathbf{y}, g \cdot f_\theta(g^{-1} \cdot \mathbf{x}))$ where $\mathcal{L}$ is binary cross entropy. All measurements are repeated 100 times and averaged over the inputs and labels $(\mathbf{x}, \mathbf{y})$ of the validation dataset.

Observations are as follows. First, models trained with smaller sample sizes need more iterations to converge, but after sufficient training (2000 epochs), all achieve $> 95\%$ test accuracy when evaluated with 10 samples (Figure 4). Second, models trained with smaller sample sizes tend to be more sample efficient, *i.e.*, tend to perform a lower variance estimation. Their distribution $p_\omega(g|\mathbf{x})$ tend to learn more low-variance permutations (Figure 5), and the models tend to learn low-variance estimation of output and loss (left and center panels of Figure 6). This indicates that small sample size may serve as a regularizer that encourages lower variance of the estimator. However, this regularization effect is not always beneficial in terms of task loss (right panel of Figure 6), as training sample size 1 achieves a poor task loss for all sample sizes presumably due to the optimization challenge caused by over-regularization. In other words, the sample size for training introduces a tradeoff; a small sample size takes more training iterations to converge, but serves as a regularizer that encourages lower variance of the estimator and thus a better inference time sample efficiency. On the other hand, larger sample sizes for inference consistently benefits all models (Figure 6).

Interestingly, this observed tendency is consistent with the theoretical claims in literature [67, 68] on the sampling based training of symmetrized models, which we reprise here. When training the symmetrized model $\phi_{\theta,\omega}(\mathbf{x})$ in Eq. (4), we cannot directly observe $\phi_{\theta,\omega}(\mathbf{x})$, but observe samples of its unbiased estimator $g \cdot f_\theta(g^{-1} \cdot \mathbf{x})$. Thus, it can be questionable what objective we are actually optimizing during the sampling-based training. Based on [67, 68], it turns out that minimizing a convex loss function $\mathcal{L}$ on the estimated output $g \cdot f_\theta(g^{-1} \cdot \mathbf{x})$ is equivalent to minimizing an upper bound to the true objective on the symmetrized output $\phi_{\theta,\omega}(\mathbf{x})$. This is because our estimation is no longer unbiased when computing loss, as we have the following from Jensen's inequality:

$$\mathbb{E}_{p_\omega(g|\mathbf{x})}[\mathcal{L}(\mathbf{y}, g \cdot f_\theta(g^{-1} \cdot \mathbf{x}))] \geq \mathcal{L}(\mathbf{y}, \mathbb{E}_{p_\omega(g|\mathbf{x})}[g \cdot f_\theta(g^{-1} \cdot \mathbf{x})]) = \mathcal{L}(\mathbf{y}, \phi_{\theta,\omega}(\mathbf{x})). \quad (55)$$

That is, minimizing the sampling-based loss is minimizing an upper-bound surrogate to the true objective. It has been claimed that optimizing this upper bound has an implicit low-variance regularization effect [67, 68], which is consistent with our observations. This also roughly explains why our distribution $p_\omega(g|\mathbf{x})$ does not collapse to uniform distribution although we do not impose any low-variance regularization explicitly; training to directly minimize the task loss with samples implicitly nudges the distribution towards low-variance solutions.

### A.4.4 Additional Comparison to Group Averaging

In this section, we provide additional analysis of our approach in comparison to sampling-based group averaging in terms of sample variance and convergence. We use the experimental setup of EXP-classify (Section 3.1; $S_n$ invariance) and experiment with MLP-PS and MLP-GA.

We first analyze whether using the equivariant distribution $p_\omega(g|\mathbf{x})$ for symmetrization offers a lower variance estimation, *i.e.*, a better sample efficiency, compared to group averaging with $\text{Unif}(G)$. This supplements the results in Figure 2 that $p_\omega(g|\mathbf{x})$ learns to produce low-variance permutations compared to $\text{Unif}(G)$. Specifically, we fix a randomly initialized MLP $f_\theta$ and symmetrize it using our approach and group averaging. We then measure **(1)** the sample variance of the unbiased estimator

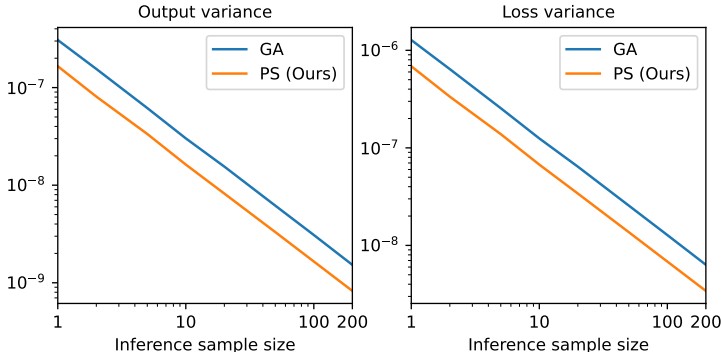

Figure 7: Sample variance of output $g \cdot f_\theta(g^{-1} \cdot \mathbf{x})$ (left) and loss $\mathcal{L}(\mathbf{y}, g \cdot f_\theta(g^{-1} \cdot \mathbf{x}))$ (right) of an identical MLP $f_\theta$ symmetrized by equivariant distribution (PS) and uniform distribution (GA).

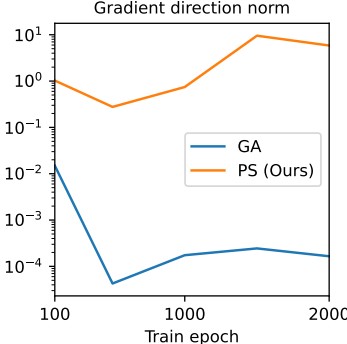

Figure 8: Norm of full gradient over training epochs with respect to the parameters of an identically initialized MLP symmetrized by equivariant distribution (PS) and uniform distribution (GA).

$g \cdot f_\theta(g^{-1} \cdot \mathbf{x})$ of the symmetrized function (Eq. (1) and Eq. (4)), and **(2)** the sample variance of the estimated task loss $\mathcal{L}(\mathbf{y}, g \cdot f_\theta(g^{-1} \cdot \mathbf{x}))$ where $\mathcal{L}$ is binary cross entropy. All measurements are repeated 100 times and averaged over the inputs and labels $(\mathbf{x}, \mathbf{y})$ of the validation dataset. The results are in Figure 7, showing that symmetrization with equivariant distribution $p_\omega(g|\mathbf{x})$ consistently offers a lower variance estimation than group averaging across inference sample sizes.

In addition, we analyze whether the equivariant distribution $p_\omega(g|\mathbf{x})$ for symmetrization offers more stable gradients for the base function $f_\theta$ during training compared to group averaging, as conjectured in Section 2. For this, we fix a randomly initialized MLP $f_\theta$ and symmetrize it using our approach and group averaging. For every few training epochs, we measure the full gradient of the task loss over the entire training dataset with respect to the parameters of the base MLP $f_\theta$. This averages out the variance from individual data points and provides the net direction of the gradient on the base function offered by $p_\omega(g|\mathbf{x})$ or $\mathrm{Unif}(G)$. The results are in Figure 8, showing that that symmetrization with equivariant distribution $p_\omega(g|\mathbf{x})$ offers a consistently larger magnitude of the net gradient, while group averaging with $\mathrm{Unif}(G)$ leads to near-zero net gradients. This indicates, for $\mathrm{Unif}(G)$, the gradients from each training data instances are oriented in a largely divergent manner and therefore the training signal is collectively not very informative, while using $p_\omega(g|\mathbf{x})$ for symmetrization leads to more consistent gradient across training data instances, *i.e.*, it offers a more stable training signal.

### A.4.5 Additional Comparison to Canonicalization

In this section, we provide additional analysis of our approach in comparison to canonicalization [41] that uses a single group element $g$ from an equivariant canonicalizer $C_\omega : \mathbf{x} \mapsto \rho(g)$. The main claim is that there always exist certain inputs that canonicalization fails to guarantee exact $G$ equivariance, while our approach guarantees equivariance for all inputs in expectation as in Theorem 1.

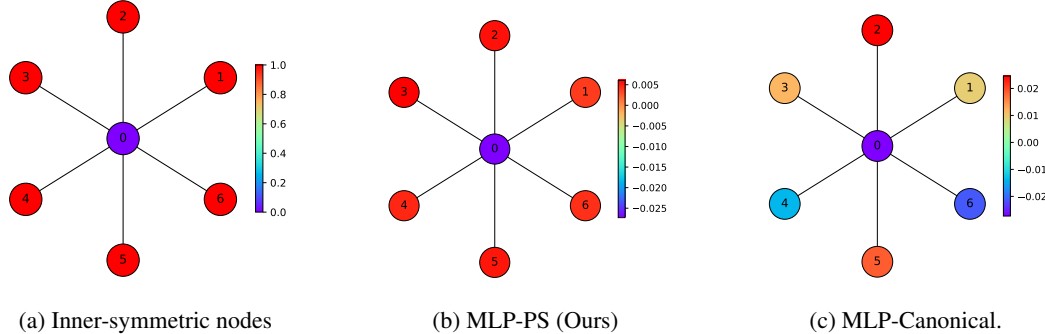

(a) Inner-symmetric nodes        (b) MLP-PS (Ours)        (c) MLP-Canonical.

Figure 9: A graph $\mathbf{x}$ with stabilizer subgroup $G_{\mathbf{x}} \cong S_6$.

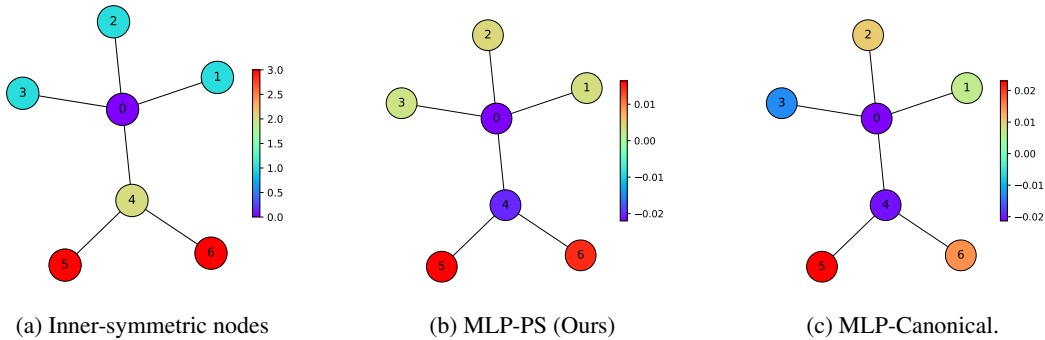

(a) Inner-symmetric nodes        (b) MLP-PS (Ours)        (c) MLP-Canonical.

Figure 10: A graph $\mathbf{x}$ with stabilizer subgroup $G_{\mathbf{x}} \cong S_3 \times S_2$.

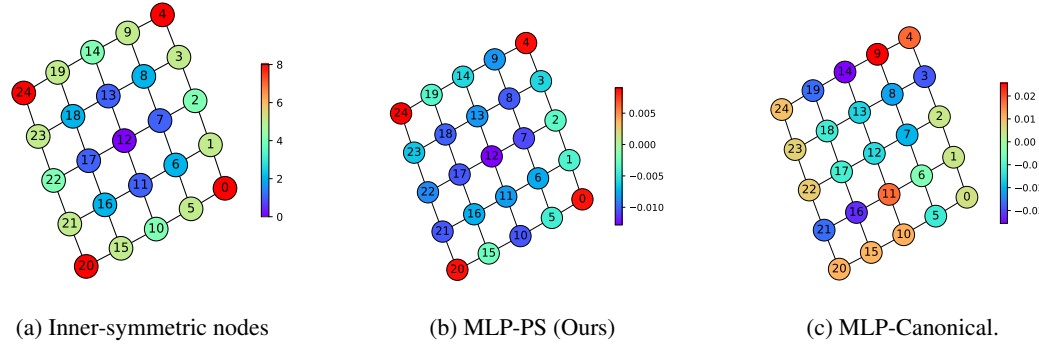

(a) Inner-symmetric nodes        (b) MLP-PS (Ours)        (c) MLP-Canonical.

Figure 11: A graph $\mathbf{x}$ with stabilizer subgroup $G_{\mathbf{x}} \cong D_4$.

More specifically, let us recall the definition of $G$ equivariant canonicalizer from [41]. A canonicalizer $C_\omega$ is $G$ equivariant if $C_\omega(g \cdot \mathbf{x}) = \rho(g) C_\omega(\mathbf{x})$ for all $g \in G$ and $\mathbf{x} \in \mathcal{X}$. Consider an input $\mathbf{x}$ which has a non-trivial stabilizer $G_{\mathbf{x}} = h \in G | h \cdot \mathbf{x} = \mathbf{x}$, *i.e.*, has inner symmetries. It can be shown that equivariant canonicalizers are ill-defined for these inputs. To see this, let $g_1 = gh_1$ and $g_2 = gh_2$ for some $g \in G$ and any $h_1, h_2 \in G_{\mathbf{x}}$ where $h_1 \neq h_2$. Then we have $C_\omega(g_1 \cdot \mathbf{x}) = C_\omega(gh_1 \cdot \mathbf{x}) = C_\omega(g \cdot \mathbf{x}) = C_\omega(gh_2 \cdot \mathbf{x}) = C_\omega(g_2 \cdot \mathbf{x})$, implying that $\rho(g_1)C_\omega(\mathbf{x}) = \rho(g_2)C_\omega(\mathbf{x})$. Since $g_1 \neq g_2$, this contradicts the group axiom, and thus an equivariant canonicalizer cannot exist for inputs with non-trivial inner-symmetries. To handle all inputs, canonicalization [41] adopts relaxed equivariance: a canonicalizer $C_\omega$ satisfies relaxed equivariance if $C_\omega(g \cdot \mathbf{x}) = \rho(gh)C_\omega(\mathbf{x})$ up to arbitrary action from the stabilizer $h \in G_{\mathbf{x}}$. As a result, the symmetrization $\phi_{\theta,\omega}(\mathbf{x}) = g \cdot f_\theta(g^{-1} \cdot \mathbf{x})$ performed using a relaxed canonicalizer $C_\omega$ only guarantees relaxed equivariance $\phi_{\theta,\omega}(g \cdot \mathbf{x}) = gh \cdot \phi_{\theta,\omega}(\mathbf{x})$ up to arbitrary action from the stabilizer $h \in G_{\mathbf{x}}$ (Theorem A.2 of [41]). In other words, canonicalization does not guarantee equivariance for inputs with inner symmetries.

To visually demonstrate this, we perform a minimal experiment using several graphs $\mathbf{x}$ with non-trivial stabilizers $G_{\mathbf{x}}$, *i.e.*, inner symmetries, taken from [90]. We fix a randomly initialized MLP

$f_\theta : \mathbb{R}^{n \times n} \to \mathbb{R}^n$ and symmetrize it using our approach and canonicalization. When symmetrized, the MLP is expected to provide scalar embedding of each node, which we color-code for visualization. The results are in Figures 9, 10, and 11. For each graph, we illustrate three panels: the leftmost one illustrates the color-coding of the inner symmetry of nodes (automorphism), the middle one illustrates node embedding from MLP-PS, and the rightmost one illustrates embedding from MLP-Canonical. If a method is $G$ equivariant, it is expected to give identical embeddings for automorphic nodes, since an equivariant model cannot distinguish them in principle [88]. As in the Figures 9, 10, and 11, in the presence of inner symmetry (left panels), MLP with probabilistic symmetrization (middle panels) achieves $G$ equivariance and produces close embeddings for automorphic nodes. However, the same MLP with canonicalization produces relatively unstructured embeddings (right panels). The result illustrates a potential advantage of probabilistic symmetrization over canonicalization when learning data with inner symmetries, which is often found in applications such as molecular graphs [60].

## A.5 Limitations and Broader Impacts (Continued from Section 4)

While the equivariance, universality, simplicity, and scalability of our approach offers a potential for positive impact for deep learning for chemistry, biology, physics, and mathematics, it also has limitations and potential negative impacts. The main limitation of our work is that it trades off certain desirable traits in equivariant deep learning in favor of achieving architecture agnostic equivariance. For example, **(1)** our approach is less interpretable compared to equivariant architectures due to less structured computations in the base model, **(2)** our approach is presumably less parameter and data efficient compared to equivariant architectures due to less imposed prior knowledge on parameterization, and **(3)** our approach is expected to be challenged when input size generalization is required, partially because the maximum input size has to be specified in advance. Another genuine weakness compared to canonicalization is that, our method is stochastic and therefore incurs $\mathcal{O}(N)$ cost when using $N$ samples for estimation. These limitations might lead to potential negative environmental impacts, since less interpretability and lower efficiency implies higher reliance on larger models with more computation cost. We acknowledge the aforementioned limitations and impacts of our work, and will make effort to address them in follow-up research. For example, we believe data efficiency of our approach could improve with pretrained knowledge transfer from other domains, as it would impose a strong prior on the hypothesis space and may work similarly to architectural priors that benefit data efficiency. Also, for the sampling cost, since the sampling is completely parallelizable and analogous to using a larger batch size, we believe it can be overcome to some degree by leveraging parallel computing techniques developed for scaling batch size.

