# OpenReview forum: "Learning Probabilistic Symmetrization for Architecture Agnostic Equivariance"
_NeurIPS.cc/2023/Conference — NeurIPS 2023 spotlight_

### Official Review · Reviewer_SdiQ · 2023-07-06

**Soundness:** 3 good
**Presentation:** 4 excellent
**Contribution:** 2 fair
**Rating:** 7
**Confidence:** 4

**Summary:**

The paper proposes a probabilistic version of the canonization method to construct equivariant architectures from generic non-equivariant backbones.
This approach is inspired by the symmetrization solution, which involves averaging a non-equivariant function to obtain an equivariant one, but replaces the uniform averaging distribution with an input-conditional one to achieve a lower sample complexity.



**Strengths:**

The paper is clearly written and well motivated.

The proposed idea overcomes some important limitations of both the symmetrisation and the canonicalisation methods.

I liked the idea of using a general-purpose transformer architecture, which makes the proposed method very flexible and easy to apply in more generic contexts.
The surprising performance of pre-trained vision models on non-vision tasks is also a very interesting result and I wonder if it can tell us something more general about these architectures.

Despite the generality and flexibility of the approach, the experiments show it can also achieve competitive performance.


**Weaknesses:**


With respect to the canonicalization method from [26], the proposed approach essentially averages over $N>1$ samples and, therefore, is $N$ times more expensive than canonicalization.
Since this work is partially motivated by computational efficiency, I think this aspect has not been sufficiently discussed in the manuscript.

Moreover, I imagine the number of samples chosen can play an important role on 1) the model's final performance, 2) the training dynamic, 3) the variance of the learned distributions and 4) the training time.
I think this point also requires some additional discussion.
I would encourage the authors to include a short experiment to study the effect of the sample size on (at least some of) these aspects.


**Questions:**


Why is it necessary to assume the group is compact?

line 96-97:  "2) learn to produce low-variance samples $g \sim p_\omega(g|x)$ that can offer more stable gradients for the base function $f_\theta$ during training."
I think this claim requires some additional comments to justify it. Why should the network learn to produce low-variance samples?

On a related note, do you train the models with some low-variance regularization? what prevents the model from outputting always a uniform distribution?

lines 114-117: this seems more a problem of parameterizing the group G itself, not its representations. The representation $\rho(g)$ can be computed in a differentiable way from $g$ when $G$ is a Lie group at least.

Eq. 8: it is not really clear how you output a permutation matrix. You use the notation $P_g \approx \hat{P}_g$ but you don't explain what it means. The matrix $\hat{P}_g$ is a real valued matrix and doesn't belong to $(0, 1)^{n \times n}$. Do you apply rounding on it?

If so, isn't it possible to just use the matrix $\hat{P}_g$? Is there any benefit in using the boolean $P_g$ instead?


You compare your probabilistic method with the deterministic canonicalization method.
It is not clear to me if the benefit is coming from the fact the backbone is averaged over $N>1$ samples or if the stochasticity of $p_\omega$ plays a role.
Could you try a version of your model which outputs $N$ samples but deterministically (e.g. sample the noise variables only once before training and keep them fixed)?


While this is already mentioned by the authors among the limitations as a future work, it would be interesting to understand if this kind of architectures enjoys the same data-efficiency property of other equivariant networks.
Do the authors have any insights about the effect of the training set size on the ability of the model to learn the symmetrising distribution?


The paper is missing a related works section. Moreover, I think an important part of the literature about equivariant networks using group-convolution is not cited.


**Limitations:**

Some limitations are discussed in the supplementary material.
I encourage the authors to move the main points mentioned there in the main paper.
See also my comments under Weaknesses.

---

> ### Author Rebuttal · Authors · 2023-08-10
>
> Q1. The proposed approach essentially averages over N > 1 samples and, therefore, is N times more expensive than canonicalization.
>
> A1. The O(N) cost with N samples is a genuine weakness, and we will add clarification n the main text. Nevertheless, as the sampling is completely parallelizable and analogous to using a larger batch size, we believe this can be overcome to some degree by leveraging parallelism [5] developed for scaling batch size.
>
> Q2. I would encourage the authors to include a short experiment to study the effect of the sample size on training.
>
> A2. We agree that controlling the number of samples used at training time would offer interesting results. We are planning to add a small empirical analysis on how the sample size affects the optimization dynamics and will post the result during the discussion period.
>
> Q3. Why is it necessary to assume the group is compact?
>
> A3. The compactness of group G is necessary for the proof in Theorem 2. Except that, the group does not need to be compact. We can extend Theorem 2 to non-compact groups if we make additional assumptions, such as the support of p(g|x) being always compact.
>
> Q4. Why should the network learn to produce low-variance samples? Do you train the models with some low-variance regularization? what prevents the model from always outputting a uniform distribution?
>
> A4. At some point, certain group elements g ~ p(g|x) can be favored more than others in terms of task loss; this is likely due to the specific configuration of the parameters of the base function f coming from random initialization or pre-training. In either case, it will signal the distribution p(g|x) to favor generating these group elements to minimize task loss, which eventually leads to learning to produce those samples consistently.
>
> In all experiments in the paper, we directly minimize the task loss and do not use low-variance regularization. In our early exploration, we experimented with two variance regularization techniques, which are simple equivariance loss [3] and entropy estimation on the distribution p(g|x) [4]. However, equivariance loss often resulted in collapsed trivial predictions, and entropy estimator did not bring performance gain. We conjecture that the task loss provides a strong signal to nudge the distribution p(g|x) to favor specific group elements g ~ p(g|x) and this leads to a reduced variance of group elements without collapsing to Unif(G).
>
> Q6. Lines 114-117: this seems more like a problem of parameterizing the group G itself, not its representations.
>
> A6. Indeed, the distribution p(g|x) does not fundamentally need to output a representation, i.e., it may produce alternative specifications of g (e.g., a Lie algebra element) from which representation is computed (e.g., with exponential map). In our work, we produce group representation as it provides a convenient medium for sampling and backpropagation through p(g|x) while guaranteeing the G equivariance of p(g|x). Alternatives, e.g., based on Lie algebra [5], is possible. We will revise the main text to clarify this.
>
> Q7. Eq. 8: it is not really clear how you output a permutation matrix.
>
> A7. We apologize for the confusion. Given nodewise scalar Z ∈ Rn from GNN, we first perform argsort to directly obtain the hard permutation matrix Pg [6]. And then, we compute the soft permutation matrix hat{P}_g following Eq. (8) [6]. hat{P}_g does not affect how the hard permutation Pg is computed, but it affects training as it is used for straight-through gradient.
>
> Q8. Isn't it possible to just use the matrix hat{P}_g? Is there any benefit in using the boolean P_g instead?
>
> A8. The first reason we use P_g is, since hat{P_g} is not a valid representation, symmetrization using it would not be G equivariant. The second reason is, using hard permutation is often critical for convergence. We conjecture this is due to the smoothing of soft permutations; in early training, hat{P}_g is close to a dense matrix, and applying it to input would smooth the information of nodes. Such smoothing is known to underlie oversmoothing of GNNs [8]. We added an experimental result of using the soft permutation for training, please see Table 4 below.
>
> Q9. Could you try a version of your model that outputs N samples but deterministically?
>
> A9. We ran a variant of our approach in the EXP-classify (Section 3.1), where the noise are sampled once before training and fixed. The results are in Table 4. The model quickly reaches 82% test accuracy, but then severely overfits the training data. This implies that the effect of averaging is insufficient to explain our approach's performance, and the stochasticity of p(g|x) plays an important role.
>
> Table 4. Ablation-extended results for EXP-classify.
>
> | method | EXP-classify ↑ |
> |:---:|:---:|
> | MLP-GA [7] | 50% |
> | MLP-FA [7] | 100% |
> | MLP-Canonical. | 50% |
> | MLP-PS (Ours) | 100% |
> | MLP-PS (Ours; soft permutation) | 50% |
> | MLP-PS (Ours; fixed noise) | 82% |
>
> Q11. The paper is missing a related works section. I encourage the authors to move the limitation sections in the main paper.
>
> A11. We will revise the main text accordingly.
>
> [1] Kaba et al., Equivariance with learned canonicalization functions (2022)
>
> [2] You et al., Scaling SGD batch size to 32k for ImageNet training (2017)
>
> [3] Murphy et al., Janossy pooling: Learning deep permutation invariant functions for variable-sized inputs (2019)
>
> [4] Seo et al., State entropy maximization with random encoders for efficient exploration (2021)
>
> [5] Finz et al., Generalizing convolutional neural networks for equivariance to Lie groups on arbitrary continuous data (2020)
>
> [6] Prillo et al., Softsort: A continuous relaxation for the argsort operator (2020)
>
> [7] Puny et al., Frame averaging for invariant and equivariant network design (2021)
>
> [8] Cai et al., A note on over-smoothing for graph neural networks (2020)

---

> > ### Comment · Reviewer_SdiQ · 2023-08-15
> >
> > I thank the authors for the detailed answer and for quickly producing the suggested experiments.
> >
> > I maintain my (positive) score and recommend the authors to include these answers in the final version of the manuscript.

---

> > > ### Author Response · Authors · 2023-08-18
> > > **Further Response**
> > >
> > > Dear Reviewer SdiQ,
> > >
> > > Thank you again for the positive and constructive comments. We will indeed include the response in the final revision of the paper.
> > >
> > > Following common requests, we have conducted an in-depth empirical analysis of the convergence and sample complexity of the algorithm using the EXP-classify dataset (Sn invariance; Section 3.1). The results can be found in the latest common response: https://openreview.net/forum?id=phnN1eu5AX&noteId=ci75ubQbgK.
> > >
> > > We think the findings might be of your interest, in particular the analysis on how the sample size for estimation affects training (**Q4**). A notable finding is that smaller sample size for training regularizes the model towards low-variance estimation, which is **beneficial** in terms of inference time sample complexity.
> > >
> > > In addition, we think we have missed one of the questions in the initial rebuttal. We apologize for this, and would like to provide our response below.
> > >
> > > > Q. While this is already mentioned by the authors among the limitations as a future work, it would be interesting to understand if this kind of architectures enjoys the same data-efficiency property of other equivariant networks. Do the authors have any insights about the effect of the training set size on the ability of the model to learn the symmetrising distribution?
> > >
> > > A. For a given task, assuming underfitting is not a problem, we conjecture our method (and symmetrization approaches in general) would not be able to enjoy the same level of data efficiency as equivariant architectures when trained from random initialization. This is because the hypothesis space of symmetrized function would be larger in general (G equivariant universal, Theorem 2) than equivariant architectures. Based on the traditional notion of model flexibility and overfitting, symmetrized models would require more data to reach a generalizable solution. However, if we consider knowledge transfer from other domains through pre-trained base function, these transferred parameters (knowledge) would impose a strong prior on the hypothesis space; in this case, transferred knowledge could possibly improve the data efficiency of symmetrized models. The symmetrizing distribution p(g|x) here would serve as an aligner between the pretrained knowledge and target task. While we currently do not have a clear conjecture on the data efficiency of learning p(g|x) as an aligner, we plan to investigate it in future work.
> > >
> > > Sincerely,
> > >
> > > Authors of submission 12717

---

### Official Review · Reviewer_GZXj · 2023-07-07

**Soundness:** 3 good
**Presentation:** 3 good
**Contribution:** 2 fair
**Rating:** 5
**Confidence:** 3

**Summary:**

The paper presents a method to learn to symmetrize a neural network using data. The method considers a learnable probability distribution over the group and uses group averaging to enforce (relaxed) equivariance w.r.t. the learned distribution.

**Strengths:**

Over recent years there has been a growing interest in the community to learn relaxed symmetries (invariance and equivariance) from data. The paper addresses this important problem and provides a practical method that scales to interesting model classes and data problems (e.g. the use of Transformers, particle dynamics, and graphs is noteworthy).

**Weaknesses:**

* Missed prior work

There have been several approaches that aim to learn symmetrizing distributions (invariance or equivariance). Placing a probability distribution on the group over which a function is averaged and learning this distribution from data is not new. For instance, see approaches listed in survey Sec. 6 of [2].

* Proposed method already exists

The paper misses this literature and it seems that the method has been published before (Equivariant Augerino, Sec. 3.1 of [1]).
An argument could be made that although the Augerino paper describes the method for general groups, it only considers very simple affine groups in practice. This paper does consider more interesting group structures and domains (e.g. graphs). However, if the same method has been published before and is applied to new domains/data, this should at least be credited and discussed. This would make it more clear what the contributions of the work are.

* Objective function

A well-known difficulty with learning symmetry distribution is not necessarily describing a probability distribution on the group but rather the objective to learn the distribution over the group. Although the method claims to be probabilistic, it is not clear what objective is used to train the model (or whether inference is being performed). If regular maximum likelihood loss is used, it seems there is no encouragement in the objective that prevents the collapse of the symmetrization distribution into a delta peak at the identity. In such a case, it would result in not learning the symmetrization that generalises well. Prior works often consider more sophisticated losses, such as regularization [1], lower bounds or model selection, to prevent this. The paper seems to skip over this issue.

[1] Benton, Gregory, et al. "Learning invariances in neural networks from training data." Advances in neural information processing systems 33 (2020): 17605-17616.

[2] Rath, Matthias, and Alexandru Paul Condurache. "Boosting deep neural networks with geometrical prior knowledge: A survey." arXiv preprint arXiv:2006.16867 (2020).

**Questions:**

a) How does the method differ from equivariant Augerino (Sec. 3.1 of [1])?

b) What objective is used, or how is inference over the probability distribution being performed?

c) If a maximum likelihood objective is used, what mechanism prevents the symmetrization from collapsing into the identity? Did authors notice such a collapse in experiments, or is this mitigated somehow?



[1] Benton, Gregory, et al. "Learning invariances in neural networks from training data." Advances in neural information processing systems 33 (2020): 17605-17616.


**Limitations:**

- The paper seems to miss prior works that consider learning similar parameterizations over group structures.
- The proposed method seems to be proposed already (equivariant Augerino, Sec. 3.1 of [1])
- It is not clear what objective is being used to train the model and what prevents collapse of the symmetrization distribution.

---

> ### Author Rebuttal · Authors · 2023-08-10
>
> Q1. The proposed method already exists.
>
> A1. Since our approach parameterizes a distribution p(g|x) on a group G for symmetrization ϕ(x) = E_g[g⋅f(g-1⋅x)] and learns it from data, one may find some similarity to Augerino [1] and other approaches in [2] such as [3-7] that learn distributions over augmentations (e.g., p(g)) for a similarl symmetrization. However, we would like to clarify that our method is distinguished from all these approaches [1-7], including Augerino [1], as our method is the only one that guarantees G equivariance of symmetrized ϕ(x) while able to learn useful distribution p(g|x) from data to maximize performance. Let us elaborate below.
>
> First, it has to be noted that the objectives of the approaches are different. Our approach aims to obtain an exact G equivariant symmetrization ϕ(x) given the known symmetry group G of data (e.g., G=S_n for graphs), while [1-7] aim to discover underlying (approximate) symmetry constraint from data. Because of this, the symmetrizing distribution on G has to be designed differently: In our case, we parameterize the distribution p(g|x) itself to be G equivariant (see Theorem 1), while for [1-7], the distribution p(g) is parameterized for a completely different purpose of covering a range of different symmetry constraints and their approximations (e.g., a set of 2D affine transformations [1]).
>
> Importantly, this allows our approach to learn non-trivial and useful distribution p(g|x) per data x while keeping the symmetrized ϕ(x) exactly G equivariant. This is a key advantage distinguished from [1-7] if the group G is given, since [1-7] does not guarantee G equivariance, nor can they learn useful p(g|x) in the case they learn to be G equivariant. For example, Augerino [1] puts an unconditional distribution p(g) on group G; in order to achieve exact G equivariance, p(g) has to learn a right invariant distribution, i.e., p(gh) = p(g) for all h ∈ G (see Sec. 3.1 of [1]). Since such p(g) is a Haar distribution, i.e., the uniform distribution Unif(G), it can be seen that Augerino has to reduce to basic group averaging [8] in order to achieve exact G equivariance. Please note that we have already extensively discussed the advantages of our approach compared to group averaging in the main text; for example, see Lines 42-45 and 90-97. The same argument also applies to [3-6]: see Sec. 3.1 of [3], Appendix A of [4], Sec. 3 of [5], and Sec. 3.4 of [6]. LILA [7] is closer to our approach as it defines an input-conditional augmented distribution p(x’|x) (see Eq. (1) of [7]), but still does not guarantee G equivariance of ϕ(x) as the distribution p(x’|x) is unconstrained.
>
> We will revise the main text and add the above citations and discussion to make this more clear.
>
> Q2. It is not clear what objective is being used to train the model and what prevents the collapse of the symmetrization distribution.
>
> A2. In symmetrization approaches that aim to discover symmetry or its extent from data [1-7], encouragement in the objective, e.g., through regularization [1] or model selection [7], has to be used to prevent the collapse of the symmetry to the trivial group (i.e., p(g|x) = \delta(g=id) for all x). At a high level, this is because the symmetrized ϕ(x) = E_g[g⋅f(g-1⋅x)] is allowed to search over the space of symmetries, which includes the trivial group; if one uses the maximum likelihood objective, the model would likely favor the trivial symmetry because it is the least constrained and would fit the training data most easily.
>
> However, in our case, the goal is not to search over the space of symmetry groups but to build an exact G equivariant function for a given symmetry group G (e.g., G=S_n for graphs). For this, as in Theorem 1, we constrain the symmetrizing distribution p(g|x) to be G equivariant, which leads to a strong provable guarantee that the symmetrized function ϕ(x) = E_g[g⋅f(g-1⋅x)] is always G equivariant regardless of how the (G equivariant) distribution p(g|x) is trained. In other words, the symmetrized model ϕ(x) cannot collapse to trivial symmetry, as it is enforced to be equivariant for the given symmetry group G. This allows us to use regular maximum likelihood objective for training without the need to address symmetry collapse.
>
> As a more bottom-up interpretation, we can show that if a symmetrizing distribution p(g|x) is G equivariant (as in Theorem 1), it cannot technically collapse to a delta peak at the identity p(g|x) = \delta(g=id) for all x. To prove this, assume that a G equivariant p(g|x) has collapsed. Recall G equivariance p(g|x) = p(g’g|g’⋅x) for all x and g’ ∈ G (see Eq. (5)); transforming input has to transform the distribution accordingly. This yields a contradiction, as transforming an input x to g’⋅x has to transform the distribution p(g|x) with g’ as well, and as a result, p(g|x’) for x’ = g’⋅x is no longer a delta peak at identity.
>
> Q3. How is inference performed?
>
> A3. Since we use regular maximum likelihood objective, we only need to perform sampling g ~ p(g|x) for MC estimation of ϕ(x) to obtain loss, and inference or density estimation on the symmetrizing distribution p(g|x) is not necessary.
>
> [1] Benton, Gregory, et al. "Learning invariances in neural networks from training data." (2020)
>
> [2] Rath, Matthias, and Alexandru Paul Condurache. "Boosting deep neural networks with geometrical prior knowledge: A survey." (2020).
>
> [3] ​​Rommel, Cédric, et al. “Deep invariant networks with differentiable augmentation layers.” (2022).
>
> [4] Wilk, Mark van der, et al. “​​Learning invariances using the marginal likelihood.” (2018).
>
> [5] Wilk, Mark van der, et al. “Learning invariant weights in neural networks.” (2022).
>
> [6] Romero, David W., et al. “Learning partial equivariances from data.” (2021).
>
> [7] Immer, Alexander, et al. “Invariance learning in deep neural networks with differentiable Laplace approximations.” (2022).
>
> [8] Yarotsky, Dmitry. ”Universal approximations of invariant maps by neural networks.” (2018).

---

> > ### Comment · Reviewer_GZXj · 2023-08-14
> >
> > I would like to thank the authors for their detailed reply and explanations. I have updated my rating.

---

### Official Review · Reviewer_wgQS · 2023-07-10

**Soundness:** 3 good
**Presentation:** 3 good
**Contribution:** 2 fair
**Rating:** 6
**Confidence:** 3

**Summary:**

The paper suggests a probabilistic approach to symmetrization, where an input conditional distribution is used to replace the untractable haar measure distribution of infinite groups. In turn, the paper identifies what are the conditions on the conditional distribution under which the symmetrization yields an equivariant function.  The method is evaluated on several benchmarks involving permutation and rotation symmetries.

**Strengths:**

Overall the paper is well-written and easy to follow.
The formulation of the method seems to be adequate.
The proposed idea is simple, and a natural extension of previous methods.

**Weaknesses:**

Relation to previous works

The paper in the introduction states that existing approaches focus on either manually deriving smaller subsets [1], or implementing a relaxed version of equivariance [2]. It is not clear what manually means in this context. In fact, [1] identifies a similar condition to eq(5) when the function p(g|x) is a delta function. I noticed section 2.4 relates to these previous works more thoroughly. More importantly, I feel [2] and the suggested approach are very similar, as [2] does not only suggest an approximate equivariant but rather suggests a learnable frame function (group subset), satisfying the equivariance constraint (similar to eq (5)). Thus, I feel this work should convince us of the necessity of the probabilistic model for learnable frames. To this end, I would have expected an ablation study testing “apples-to-apples” architectures where the only difference is the addition of the input noise term.
It would also be beneficial to separate the test of averaging over S_n to O(3) in the experiment in table (2) and use symmetrization only to the O(3) symmetries with S_n equivariant networks.

Missing discussion

In fact, the network prediction in eq(4) is only approximated. Thus I assume, the loss terms used for training only estimate the true loss (gradients). How stable are these estimates? How well do they approximate the true loss?

**Questions:**

I would appreciate any response from the authors regarding the weakness stated above.

**Limitations:**

Could not find a discussion on limitations

---

> ### Author Rebuttal · Authors · 2023-08-10
>
> Q1. The introduction states that [1] focuses on manually deriving smaller subsets, but it is not clear.
>
> A1. We intended to describe that one needs to manually solve G equivariant set-valued frame, and it cannot be discovered from data. In contrast, the G equivariant distribution p(g|x) requires less hand design and can be learned entirely from data.
>
> Q2. [2] and the suggested approach is very similar.
>
> A2. Theoretically, we can show that there always exist certain input data that canonicalization fails to guarantee exact G equivariance, while PS guarantees G equivariance to all input data (as in Theorem 1).
>
> To see this, let us recall the definition of a G equivariant canonicalizer C: x ↦ ρ(g) [2]. In the paper, it is stated that a canonicalizer C is G equivariant if C(g⋅x) = ρ(g)C(x) for all g ∈ G and x ∈ X. Now consider an input x which has a non-trivial stabilizer subgroup G_x = {h ∈ G | h⋅x = x}, i.e., has inner symmetries. It can be seen that G equivariant canonicalizer is ill-defined for these inputs. Specifically, let g1 = gh1 and g2 = gh2 for some g ∈ G and any h1, h2 ∈ G_x where h1 ≠ h2. Then we have C(g1⋅x) = C(gh1⋅x) = C(g⋅x) = C(gh2⋅x) = C(g2⋅x), implying that ρ(g1)C(x) = ρ(g2)C(x). Since g1 ≠ g2, this contradicts the group axiom, and therefore a G equivariant canonicalizer C cannot exist for inputs x with non-trivial G_x. For a more detailed discussion of this problem, please see Appendix A of [1].
>
> To handle all possible inputs, canonicalization [1] adopts relaxed equivariance: a canonicalizer C satisfies relaxed equivariance if C(g⋅x) = ρ(gh)C(x) up to arbitrary action from the stabilizer h ∈ G_x. As a result, the symmetrization ϕ(x) = g⋅f(g-1⋅x) performed using a relaxed canonicalizer C only guarantees relaxed equivariance ϕ(g⋅x) = gh⋅ϕ(x) up to arbitrary action from the stabilizer h ∈ G_x (proof is given in [1]). Intuitively, this means canonicalization does not guarantee G equivariant processing for input data x with inner symmetries G_x.
>
> To visually demonstrate this, we performed a minimal experiment using several graphs x with non-trivial stabilizers G_x (inner symmetries [2]). The results are in Figures 1-3 of the common response. We fixed a randomly initialized MLP f: Rn x n → Rn and symmetrized it using our approach and canonicalization. When symmetrized, the MLP is expected to provide scalar embedding of each node, which we color-code for visualization. For each graph, we illustrate three panels: the leftmost one illustrates the color-coding of the inner symmetry of nodes (automorphic nodes), the middle one illustrates node embedding from MLP-PS, and the rightmost one illustrates node embedding from MLP-Canonical. If a method is G equivariant, it is expected to give identical embeddings for automorphic nodes because an equivariant model cannot distinguish between the identities of individual automorphic nodes in principle [3].
>
> As  in Figures 1-3, in the presence of inner symmetry (left panels), MLP with PS (middle panels) is able to perform G equivariant processing and produce almost identical embeddings for automorphic nodes. However, the same MLP with canonicalization fails and produces unstructured embeddings (right panels). The result illustrates a potential advantage of PS over canonicalization when learning data with inner symmetries, which is found in applications such as molecular graph processing [4].
>
> Q3. I expected an ablation study where the only difference is the addition of the input noise.
>
> A3. All our main experiments in Sections 3.1-3.3 are conducted in the way the reviewer suggested. Please see Lines 214-216 in the main text; all canonicalization models are constructed by deleting the noise term from the probabilistic mode using identical architectures.
>
> Q4. It would be beneficial to use symmetrization only to the O(3) symmetries with S_n equivariant networks.
>
> A4. We additionally conducted an experiment on E(3) symmetrization of an S_n equivariant GNN for n-body task (Section 3.2). The results are in Table 2 of the common response, and our approach obtains state-of-the-art performance and significantly improves over all other symmetrization approaches.
>
> Q5. The loss terms used for training only estimate the true loss (gradients). How stable are these estimates? How well do they approximate the true loss?
>
> A5. While Eq. (4) gives a G equivariant symmetrized function ϕ(x), we cannot directly observe ϕ(x), but can obtain samples of the unbiased estimator g⋅f(g-1⋅x). Given that, it can be questionable what objective we are actually optimizing when we use estimations of ϕ(x). Fortunately, a theoretical framework for training of S_n symmetrized models has been established [3-4], from which we generalize to the general group. The main message is that minimizing a convex loss function based on the sampled g⋅f(g-1⋅x) is equivalent to minimizing an upper bound to the true objective that involves ϕ(x).
>
> Given a training set D = {(x1, y1), …, (xn, yn)} for a G equivariant task, our true objective would be minimizing the empirical loss L(D; θ, ω) = sum_i l(y_i, ϕ(x_i)) where l is a convex loss. However, in practice, ϕ(x) cannot be observed, and we observe g⋅f(g-1⋅x). While g⋅f(g-1⋅x) serves as an unbiased estimator of ϕ(x), the estimation is no longer unbiased when g⋅f(g-1⋅x) is used to compute the loss: from Jensen’s inequality, we have E_g[l(y, g⋅f(g-1⋅x))] ≥ l(y, E_g[g⋅f(g-1⋅x)]). That is, minimizing the sampling-based loss is minimizing an upper-bound surrogate to the true objective [3].
>
> Q6. Could not find a discussion on limitations.
>
> A6. We will move Appendix A.4 to the main text for visibility.
>
> [1] O. Puny et al., Frame averaging for invariant and equivariant network design (2021)
>
> [2] S. Kaba et al., Equivariance with learned canonicalization functions (2022)
>
> [3] R. L. Murphy et al., Relational pooling for graph representations
>
> [4] R. L. Murphy et al., Janossy pooling: Learning deep permutation invariant functions for variable-sized inputs

---

> > ### Author Response · Authors · 2023-08-18
> > **Further Response**
> >
> > Dear Reviewer wgQS,
> >
> > Thank you again for the insightful and constructive comments. Following common requests, we have conducted an in-depth empirical analysis of the convergence and sample complexity of the algorithm using the EXP-classify dataset (Sn invariance; Section 3.1). The results can be found in the latest common response: https://openreview.net/forum?id=phnN1eu5AX&noteId=ci75ubQbgK.
> >
> > We think the findings might be of your interest, in particular the analysis on how the sample size for estimation affects training (**Q4**). A particularly interesting finding is that smaller sample size for training regularizes the model towards low-variance estimation. This supplements our previous response on sampling-based loss, and provides some understanding of the side effect induced by optimizing the sampling-based upper-bound to the true objective.
> >
> > Sincerely,
> >
> > Authors of submission 12717

---

> > > ### Comment · Reviewer_wgQS · 2023-08-21
> > > **rebuttal**
> > >
> > > I thank the authors for a detailed rebuttal. The rebuttal addressed my concerns, I will raise my score to 6. Thank you.

---

### Official Review · Reviewer_Ry8Q · 2023-07-10

**Soundness:** 4 excellent
**Presentation:** 3 good
**Contribution:** 3 good
**Rating:** 6
**Confidence:** 3

**Summary:**

This paper proposes a new symmetrization method for achieving equivariance for any base function. It absorbs previous methods with the same objective as special cases, including group averaging, frame averaging, and canonicalization. The method uses a learnable, equivariant map to generate the group representation necessary for achieving equivariance from an invariant, external noise variable, followed by necessary post-processing to ensure the validity of the group representation. The authors empirically demonstrate that the learned stochasticity in symmetrization leads to improved model performance than the previous methods, and showcase the benefits of general pre-training for equivariant models.

**Strengths:**

* The proposed method is general, providing a unified perspective for previous methods and may lay the foundation for further theoretical exploration of the "optimal" conditional group distribution.
* The presentation is clear and easy to read.
* The finding of the benefit of non-symemtric pretraining for equivariant models seems interesting and new.

**Weaknesses:**

* My primary concern is that the supriorty of the proposed method compared to the canonicalization method is not clear enough.
    - The advantages of the former over the latter seem to be more evident only in the first experiment, as the canonicalization method appears to have difficulties during optimization (although it is unknown whether this issue can be mitigated through some warm-up techniques). In the other two experiments, the performance of both methods is quite similar.
    - On the other hand, compared to the canonicalization method, the proposed method incurs more computational cost in practice, including the additional parameters and hyperparameters in the learnable module, as well as the sampling cost (the normalization method only requires a single "sample", whereas here we need 10-20 samples).

+ This paper lacks thorough empirical and theoretical analyses for the two claimed superiorities over previous methods. Specifically,
    - "Learn to collaborate with base function to maximize task performance": No sensitivity analysis is provided to demonstrate how this method adapts better to different base functions.
    - "learn to produce low-variance samples that can offer more stable gradients for the base function": No analysis of gradient magnitudes is provided to indicate its stability.

* The limitation is not discusses, including the extra computational cost.
* The post-process seems interesting but also laborious, as it requires different designs for each different (compact) groups.

**Questions:**

* What's the computational cost of this method, including training and testing, compared to the canonicalization method?
* Analyzing the sampling efficiency for estimating expectations would be beneficial, especially in comparison to the uniform distribution, as it is a major claim of the effectiveness of this method.

**Limitations:**

The authors have not discussed the limitations.

---

> ### Author Rebuttal · Authors · 2023-08-10
>
> Q1. The superiority compared to canonicalization is not clear.
>
> A1. Theoretically, we can show that there always exist certain input data that canonicalization fails to guarantee exact G equivariance, while PS guarantees G equivariance to all input data (as in Theorem 1).
>
> To see this, let us recall the definition of G equivariant canonicalizer C: x ↦ ρ(g) [1]. A canonicalizer C is G equivariant if C(g⋅x) = ρ(g)C(x) for all g ∈ G and x ∈ X. Consider an input x which has a non-trivial stabilizer subgroup G_x = {h ∈ G | h⋅x = x}, i.e., has inner symmetries. It can be seen that G equivariant canonicalizer is ill-defined for these inputs. Specifically, let g1 = gh1 and g2 = gh2 for some g ∈ G and any h1, h2 ∈ G_x where h1 ≠ h2. Then we have C(g1⋅x) = C(gh1⋅x) = C(g⋅x) = C(gh2⋅x) = C(g2⋅x), implying that ρ(g1)C(x) = ρ(g2)C(x). Since g1 ≠ g2, this contradicts the group axiom, and therefore a G equivariant canonicalizer C cannot exist for inputs x with non-trivial G_x. For more detailed discussion, please see [1].
>
> To handle all inputs, canonicalization [1] adopts relaxed equivariance: a canonicalizer C satisfies relaxed equivariance if C(g⋅x) = ρ(gh)C(x) up to arbitrary action from the stabilizer h ∈ G_x. As a consequence, the symmetrization ϕ(x) = g⋅f(g-1⋅x) performed using a relaxed canonicalizer C only guarantees relaxed equivariance ϕ(g⋅x) = gh⋅ϕ(x) up to arbitrary action from the stabilizer h ∈ G_x (proof is in [1]). Intuitively, this means canonicalization does not guarantee G equivariance for input data x with inner symmetries G_x.
>
> To visually demonstrate this, we performed a minimal experiment using graphs x with non-trivial stabilizers G_x (inner symmetries [2]). The results are in Figures 1-3 of the common response. We fixed a randomly initialized MLP f: Rn x n → Rn and symmetrized it using our approach and canonicalization. When symmetrized, the MLP is expected to provide scalar embedding of each node, which we color-code for visualization. For each graph, we illustrate three panels: the leftmost one illustrates the color-coding of the inner symmetry of nodes (automorphic nodes), the middle one illustrates node embedding from MLP-PS, and the rightmost one illustrates embedding from MLP-Canonical. If a method is G equivariant, it is expected to give identical embeddings for automorphic nodes as an equivariant model cannot distinguish between the identities of individual automorphic nodes in principle [3].
>
> As in Figures 1-3, in the presence of inner symmetry (left panels), MLP with PS (middle panels) achieves G equivariance and produces almost identical embeddings for automorphic nodes. However, the same MLP with canonicalization fails and produces unstructured embeddings (right panels). The result illustrates a clear advantage of PS over canonicalization when learning data with inner symmetries, which is often found in applications such as molecular graphs [4].
>
> Q2. The advantages over canonicalization are evident only in the first experiment.
>
> A2. Please note that empirical advantage is evident not only in the first experiment (Section 3.1) but also in the second one (n-body; Section 3.2) where Transformer-PS achieves 0.00417 MSE, which significantly improves over Canonicalization with 0.00779 MSE.
>
> To further demonstrate advantage over canonicalization, we added two experiments on graph isomorphism (Section 3.1) with S_n symmetrization of GIN base model with node identifiers (GIN-ID) [7], and on n-body (Section 3.2) with E(3) symmetrization of GNN. In both, we achieved state-of-the-art performance; the results are in Tables 2 and 3 of the common response. Our approach consistently improves over canonicalization as well as other symmetrizations.
>
> Q3. Compared to canonicalization, the method incurs more cost, including parameters and hyperparameters in learnable module, as well as sampling.
>
> A3. The O(N) cost with N samples is a genuine weakness, and we will add clarification in the limitations (Appendix A.4). As the sampling is parallelizable, we believe this can be overcome to some degree by leveraging parallelism [5] developed for scaling batch size.
>
> For the parameters and hyperparameters, we think it adds a negligible overhead to canonicalization. This is because canonicalization also uses a equivariant module C: x ↦ ρ(g), so our approach does not add parameters in principle. For the hyperparameters, our approach only adds scale hyperparameter ŋ of noise, and we find a simple choice ŋ = 1 works robustly in all our experiments.
>
> Q6. No thorough analyses for the claimed superiorities. No analysis on sensitivity and gradients. Analyzing sampling efficiency would be beneficial.
>
> A6. In Lines 90-97, we are making a specific comparison against group averaging rather than claiming superiorities in general. We will revise the main text for clarification.
>
> We are planning a controlled experiment to understand the training dynamics, including behavior of p(g|x), stability of gradients, and sample complexity in comparison to group averaging. We will post the results during the discussion period.
>
> Q7. The post-processing seems laborious.
>
> A7. The post-processing indeed has to be implemented for each group. However, it is not an issue specific to our approach, as it is an issue of canonicalization as well [1]. Also, we think designing it can often be more straightforward compared to alternatives, e.g., a frame which is G equivariant function that has to produce a set of group elements.
>
> [1] Kaba et al., Equivariance with learned canonicalization functions (2022)
>
> [2] Thiede et al., Autobahn: Automorphism-based graph neural nets (2022)
>
> [3] Srinivasan et al., On the equivalence between positional node embeddings and structural graph representations (2019)
>
> [4] MaKey et al., Surge: a fast open-source chemical graph generator (2022)
>
> [5] You et al., Scaling SGD batch size to 32k for ImageNet training (2017)
>
> [6] Puny et al., Frame averaging for invariant and equivariant network design (2021)

---

> > ### Author Response · Authors · 2023-08-18
> > **Further Response**
> >
> > Dear Reviewer Ry8Q,
> >
> > Thank you again for the insightful and constructive comments. Following common requests, we have conducted an in-depth empirical analysis of the convergence and sample complexity of the algorithm using the EXP-classify dataset (Sn invariance; Section 3.1). The results can be found in the latest common response: https://openreview.net/forum?id=phnN1eu5AX&noteId=ci75ubQbgK.
> >
> > We think the findings might be of your interest, in particular the analysis on the stability of gradients (**Q2**) and sample complexity in comparison to the uniform distribution (**Q1 and Q3**).
> >
> > Sincerely,
> >
> > Authors of submission 12717

---

### Official Review · Reviewer_SgC2 · 2023-07-14

**Soundness:** 3 good
**Presentation:** 3 good
**Contribution:** 3 good
**Rating:** 6
**Confidence:** 3

**Summary:**

This work presents a generic approach to symmetrize wide range of base models. Instead of relying on uniform average sampling, the approach introduces a trainable transformation to model the group equivariant distribution. The framework theoretically encompasses existing approaches such as group averaging, frame averaging, and canonical function approaches. Furthermore, the author provides empirical evidence demonstrating competitive performance.




**Strengths:**

1. The proposed method demonstrates the ability to symmetrize architectures in a group-agnostic manner for general purposes, supported by sound theory.

2. The theoretical analysis presented in this work showcases the ability of the proposed approach to encompass interesting literature that assigns distributions on the compact group $G$.

3. The paper is written in a reader-friendly manner, making it easy to follow.






**Weaknesses:**

1. A thorough discussion of the empirical and theoretical advantages and disadvantages of literatures, specifically group average, frame average, and canonicalization, would greatly enhance our understanding, given the shared problem setup with the proposed method.


2. I have reservations about the assertion made in Line 3 that
>we use an arbitrary base model (such as an MLP or a transformer)...

While the argument of this work suggests that the base model $f_{\theta}$ can be arbitrary, in the experiments, only MLP and transformer architectures were explored and evaluated.


3. Typically [1,2], showcasing improved performance on image classification problems is one of the common applications used to demonstrate the effectiveness of a newly proposed equivariant network. It is encouraged to include such experiments in order to comprehensively validate and demonstrate the efficacy of the proposed equivariant network.


[1] S. Basu, P. Sattigeri, K. N. Ramamurthy, V. Chenthamarakshan, K. R. Varshney, L. R. Varshney, and P. Das. Equi-tuning: Group equivariant fine-tuning of pretrained models

[2] S. Kaba, A. K. Mondal, Y. Zhang, Y. Bengio, and S. Ravanbakhsh. Equivariance with learned canonicalization functions



**Questions:**

1. Is it feasible to explore the convergence of the probabilistic equivariant distribution $p_{\omega}$ and assess its dissimilarity to a uniform distribution? This analysis would provide insights into the nature of the mechanism.

2. What is the sensitivity of the proposed method to the architecture scales of MLP and transformer? Can the efficacy of the proposed method be maintained when the base models have a large number of parameters?

3. Is there a general framework or guideline for designing the $G$ equivariant neural network $q_{\omega}$? How significantly does the expressivity of $q_{\omega}$ impact the performance?

4. Why some experimental comparisons between [1-3] are missing? For instance, in ``3.1 Graph Isomorphism Learning with MLP'', what limits the comparison with [3]?


[1] S. Basu, P. Sattigeri, K. N. Ramamurthy, V. Chenthamarakshan, K. R. Varshney, L. R. Varshney, and P. Das. Equi-tuning: Group equivariant fine-tuning of pretrained models

[2] S. Kaba, A. K. Mondal, Y. Zhang, Y. Bengio, and S. Ravanbakhsh. Equivariance with learned canonicalization functions

[3] O. Puny, M. Atzmon, E. J. Smith, I. Misra, A. Grover, H. Ben-Hamu, and Y. Lipman. Frame averaging for
invariant and equivariant network design



**Limitations:**

The authors have discussed limitations and potential societal impact associated with this work.

---

> ### Author Rebuttal · Authors · 2023-08-10
>
> Q1. A thorough discussion of advantages and disadvantages of symmetrization approaches would enhance our understanding.
>
> A1. We will add related work section with added thorough comparison. A tentative discussion is in Table 3. Please note that we performed theoretical analysis in Section 2.4 and Appendix A.2.5, and also included discussions on performances in Sections 3.1-3.3.
>
> Table 3. Overview of symmetrization approaches.
>
> |  | GA | FA | Canonicalization | PS (Ours) |
> |---|---|---|---|---|
> | Symmetrization | g ~ Unif(G) | g ~ Unif(F(x)) | g = C(x) | g ~ p(g\|x) |
> | \|Sample space\| | \|G\| | \|G\_x\| ≤ \|F(x)\| ≤ \|G\| | 1 | \|G\_x\| ≤ \|supp p(g\|x)\| ≤ \|G\| |
> | Advantages | Simple | Efficient if F(x)is small | Efficient, X(x) can be learned | p(g\|x) can be learned, empirically works well |
> | Disadvantages | The group G has to be compact, training challenge, costs linearly to sample size | F(x) requires manual solving, F(x) cannot be learned, costs linearly to sample size if F(x) is large | No exact equivariance for x with \|G\_x\| > 1, training challenge compared to PS | The group G has to be compact, costs linearly to sample size |
>
> Q2. While it is suggested that the base model f can be arbitrary, only MLP and transformer were evaluated.
>
> A2. In principle, we can indeed use an arbitrary base model f, but focused on MLP and transformers as they have universality guarantees as in Theorem 2, and are widely used as general-purpose architectures. We will revise Line 3 to “we use a non-equivariant, general-purpose backbone...” to clarify our contributions.
>
> To demonstrate that a more wide range of base models can be used, we added two experiments on graph isomorphism (Section 3.1) with S_n symmetrization of a GIN base model with node identifiers (GIN-ID) [7], and on particle dynamics (Section 3.2) with E(3) symmetrization of a GNN base model. The base architectures (GIN-ID and GNN) are directly adopted from [3]. In both, we achieved state-of-the-art performance; the results are in Tables 1 and 2 of the common response. The results indicate that our approach works for a wide range of base models.
>
> Q3. In [1,2], showing improved performance on image classification is a common application.
>
> A3. While rotated image datasets are useful to highlight equivariance, we think the groups are overly simple to compare symmetrization methods comprehensively. This is because the involved groups are very small, e.g., c4 contains four elements representing 90-degree rotations. In such cases, even full group averaging is computable [1]. As our method is designed to overcome the sample complexity of group averaging, we believe that it is more suitable to show our approach in more challenging groups, such as combinatorial, infinite groups, and their products, as we presented in experiments (Section 3). Please note that [3] was also not demonstrated for rotated images, likely for similar reasons.
>
> Thus, instead of image classification, we additionally conducted an experiment on E(3) symmetrization of GNN, which is another widely used setup [2, 3]. We have obtained state-of-the-art result, which is in Table 2 of the common response.
>
> Q4. Convergence of the p(g|x) and dissimilarity to uniform distribution?
>
> A4. We have provided an analysis of the convergence of p(g|x) for S_n in Figure 1 of the main text, with comparison to uniform distribution (random permutations). In Figure 1, it can be seen that the distribution learns to produce consistent samples g ~ p(g|x) in early training, diverging from uniform distribution. In later training, the samples' variance g ~ p(g|x) slightly increases, but validation loss keeps improving, indicating that the model is leveraging stochasticity to improve performance.
>
> Q5. Sensitivity to the scales?
>
> A5. Our approach is robust to a wide range of scales, as our smallest base architecture involves 107,523 parameters (Section 3.2), and the largest involves 86M parameters (ViT-Base in Sections 3.3-3.4).
>
> Q6. A guideline for designing the G equivariant q? How significantly does its expressivity impact performance?
>
> A6. We found that having global interactions across input in q is helpful. For example, for S_n, the network q is expected to output a permutation by assigning score to each node. It is expected that it has to perform a kind of comparison over the graph, where global interaction could be useful and was implemented by virtual node and batch normalization. Other than this, the performance was robust across setups for a reasonable q. In all of our experiments on S_n, we use 3-layer GIN, and in all experiments on S_n x E(3) or E(3), we use 2-layer Vector Neurons, and found no issues.
>
> Q7. Some experimental comparisons between [1-3] missing?
>
> A7. We would like to clarify that our approach is compared to [1-3] for all experiments in Sections 3.1-3.3, except in one case where [3] was unavailable. Note that Equi-Tuning [1] is group averaging [5], as clarified in [1]. In this regard, in all experiments in Sections 3.1-3.3, we made comparisons with GA [1, 6], canonicalization [2], and FA [3]. One exception is in Table 2, where we could not experiment FA since frame for the full group S_n x E(3) has not been identified in current literature.
>
> To further provide experimental comparisons to [1-3], we added experiments as mentioned in A1. The results can be found in Tables 2 and 3 of the common response, and they are consistent with our main findings as PS outperforms [1-3].
>
> [1] S. Basu et al., Equi-tuning: Group equivariant fine-tuning of pretrained models
>
> [2] S. Kaba et al., Equivariance with learned canonicalization functions
>
> [3] O. Puny et al., Frame averaging for invariant and equivariant network design
>
> [4] S. Basu et al., Equivariant few-shot learning from pretrained models
>
> [5] D. Yarotsky. Universal approximations of invariant maps by neural networks
>
> [6] B. Sturmfels. Algorithms in invariant theory
>
> [7] R. L. Murphy et al., Relational pooling for graph representations

---

> > ### Comment · Reviewer_SgC2 · 2023-08-17
> >
> > The authors have addressed all my questions and concerns well. Thus, I increased my evaluation.

---

> > > ### Author Response · Authors · 2023-08-18
> > > **Further Response**
> > >
> > > Dear Reviewer SgC2,
> > >
> > > Thank you again for the positive and constructive comments. Following common requests, we have conducted an in-depth empirical analysis of the convergence and sample complexity of the algorithm using the EXP-classify dataset (Sn invariance; Section 3.1). The results can be found in the latest common response: https://openreview.net/forum?id=phnN1eu5AX&noteId=ci75ubQbgK.
> > >
> > > We think the findings might be of your interest, as it includes results on convergence of p(g|x) in comparison to the uniform distribution Unif(G).
> > >
> > > Sincerely,
> > >
> > > Authors of submission 12717

---

### Author Rebuttal · Authors · 2023-08-10

The common response PDF contains the following:

- Table 1, including additional results on $\textnormal{S}\_n$ invariant graph isomorphism learning with $\textnormal{S}\_n$ symmetrized GIN-ID base function.

- Table 2, including additional results on $\textnormal{S}\_n\times\textnormal{E}(3)$ equivariant $n$-body problem with $\textnormal{E}(3)$ symmetrized GNN base function.

- Figure 1 to 3, including visualizations of scalar node embeddings of graphs produced by randomly initialized MLP-PS (Ours) and MLP-Canonical. under the presence of inner-symmetries.

We have added the responses to each reviewer individually.

---

### Author Response · Authors · 2023-08-18
**Additional In-Depth Analysis on Convergence and Sample Sizes (1/2)**

Dear Reviewers,

We sincerely appreciate insightful and constructive comments from all reviewers. Following common requests (SgC2, Ry8Q, wgQS, SdiQ), we have conducted an in-depth empirical analysis of the convergence and sample complexity of our method using the EXP-classify dataset (Sn invariance; Section 3.1). We want to share our main findings that might be of general interest. Since we cannot share the figures directly in the discussion period, we used the tables to present the results. We will update our draft to reflect our new findings with better presentations.

Let us recap that our algorithm uses an equivariant distribution p(g|x) and a base function f(x) to obtain a symmetrized function ϕ(x) = E_g[ g⋅f(g-1⋅x) ].

&nbsp;
&nbsp;
&nbsp;

---

&nbsp;
&nbsp;
&nbsp;

**Q1. Does p(g|x) learn to produce low-variance samples compared to Unif(G)? (SgC2, Ry8Q, SdiQ)**

Experiment: For every few training epochs, we sampled $N = 100$ permutation matrices $\mathbf{P}\_{g} \sim$ p(g|x) for each validation data $\mathbf{x}$ and obtained their average $\bar{\mathbf{P}}\_{\mathbf{x}} = \sum \mathbf{P}\_{g} / N$. Then, we measured the mean of row- and column-wise entropy of $\bar{\mathbf{P}}\_{\mathbf{x}}$ for quantitative analysis. If the distribution p(g|x) has learned low-variance samples, the entropy of $\bar{\mathbf{P}}\_{\mathbf{x}}$ is expected to be low. The results are summarized in Table A. Note that a similar analysis is visually presented in the Figure 1 of the main text.

**Table A.** Entropy of $\bar{\mathbf{P}}\_{\mathbf{x}}$ averaged over validation dataset $\\{\mathbf{x}\\}$.

| Train epoch | 100 | 500 | 1000 | 1500 | 2000 |
| --- | --- | --- | --- | --- | --- |
| p(g\|x) | 6.25 | 6.04 | 5.93 | 5.84 | 5.71 |
| Unif(G) | 7.03 | 7.03 | 7.03 | 7.03 | 7.03 |

Observation: The entropy reduces throughout training and is consistently smaller than that of Unif(G).

**A1. In EXP-classify, p(g|x) learns to produce lower-variance samples than Unif(G).**

&nbsp;
&nbsp;
&nbsp;

---

&nbsp;
&nbsp;
&nbsp;

**Q2. Does p(g|x) offer more stable gradients for the base function than Unif(G)? (Ry8Q, SdiQ)**

Experiment: For every few training epochs, we computed the full gradient of the loss over the entire training dataset with respect to the parameters of the base function f(x). This averages out the variance from individual data point and provides the net direction of the gradient on the base function offered by p(g|x) or Unif(G). For a controlled comparison, we fixed the initialization of the base function to be identical in both settings.

**Table B.** Norm of the full gradient w.r.t. the parameters of the base function trained with p(g|x) or Unif(G).

| Train epoch | 100 | 500 | 1000 | 1500 | 2000 |
| --- | --- | --- | --- | --- | --- |
| p(g\|x) | 1.13 | 1.12 | 0.34 | 2.00 | 3.40 |
| Unif(G) | 1.35e-2 | 4.27e-5 | 1.74e-4 | 2.44e-4 | 1.65e-4 |

Observation: When we use p(g|x) for symmetrization, the net gradient for the base function has a consistently larger magnitude in contrast to Unif(G) that leads to near-zero net gradients. This indicates, for Unif(G), the gradients from each training data instances are oriented in a largely divergent manner and therefore the training signal is collectively not very informative, while using p(g|x) leads to more consistent gradient across training data instances, *i.e.*, it offers a more stable training signal.

**A2. In EXP-classify, p(g|x) offers more stable gradients for the base function.**

&nbsp;
&nbsp;
&nbsp;

---

&nbsp;
&nbsp;
&nbsp;

**Q3. Does p(g|x) offer better sample efficiency (lower variance estimation) than Unif(G)? (Ry8Q, SdiQ)**

Experiment: We fixed a randomly initialized base function f(x) and computed the variance of the estimated output ϕ(x) and loss L(ϕ(x), y) with 100 repeated measurements for a range of inference sample sizes.

**Table C.** Variance of estimated output ϕ(x) under p(g|x) and Unif(G). Scaled by 1e7 for readability. Lower variance for each inference sample size is boldfaced.

| Inference sample size | 1 | 2 | 5 | 10 | 20 | 50 | 100 | 200 |
| --- | --- | --- | --- | --- | --- | --- | --- | --- |
| p(g\|x) | **1.65** | **0.81** | **0.33** | **0.17** | **0.083** | **0.033** | **0.016** | **0.008** |
| Unif(G) | 3.05 | 1.53 | 0.63 | 0.31 | 0.16 | 0.062 | 0.030 | 0.015 |

**Table D.** Variance of estimated loss L(ϕ(x), y) under p(g|x) and Unif(G). Scaled by 1e7 for readability. Lower variance for each inference sample size is boldfaced.

| Inference sample size | 1 | 2 | 5 | 10 | 20 | 50 | 100 | 200 |
| --- | --- | --- | --- | --- | --- | --- | --- | --- |
| p(g\|x) | **6.83** | **3.36** | **1.36** | **0.69** | **0.345** | **0.138** | **0.068** | **0.035** |
| Unif(G) | 12.63 | 6.35 | 2.59 | 1.27 | 0.641 | 0.256 | 0.126 | 0.064 |

Observation: For all measurements, p(g|x) consistently offers smaller variances of estimation than Unif(G).

**A3. In EXP-classify, p(g|x) offers a better sample efficiency than Unif(G).**

---

> ### Author Response · Authors · 2023-08-18
> **Additional In-Depth Analysis on Convergence and Sample Sizes (2/2)**
>
> ---
>
> &nbsp;
>
> **Q4. How does the sample size affect training? (wgQS, SdiQ)**
>
> Experiment: We trained multiple models with same initialization and hyperparameters, only controlling the sample size g ~ p(g|x) for training from 1 to 50. We used early stopping based on val loss.
>
> **Table E.** Test accuracy with inference sample size 10.
>
> | Train sample size | 1 | 2 | 5 | 10 | 20 | 50 |
> | --- | --- | --- | --- | --- | --- | --- |
> | Accuracy | 98.5% | 100% | 100% | 100% | 96% | 96% |
>
> **Table F.** Entropy of $\bar{\mathbf{P}}\_{\mathbf{x}}$ averaged over validation dataset $\\{\mathbf{x}\\}$. Lower value indicates lower variance of p(g|x); please see Q1 for description.
>
> | Train sample size | 1 | 2 | 5 | 10 | 20 | 50 |
> | --- | --- | --- | --- | --- | --- | --- |
> | Entropy | 5.49 | **4.94** | 5.42 | 5.69 | 5.66 | 5.76 |
>
> **Table G.** Variance of estimated output ϕ(x). Lowest variance for each inference sample size is boldfaced.
>
> | Train sample size | 1 | 2 | 5 | 10 | 20 | 50 |
> | --- | --- | --- | --- | --- | --- | --- |
> | Inference sample size 1 | **4.26e-2** | 4.65e-2 | 7.45e-2 | 1.29e-1 | 1.37e-1 | 9.68e-2 |
> | Inference sample size 2 | 2.88e-2 | **1.62e-2** | 3.04e-2 | 8.37e-2 | 1.20e-1 | 1.12e-1 |
> | Inference sample size 5 | 1.36e-2 | **1.96e-3** | 3.11e-3 | 2.24e-2 | 4.98e-2 | 8.14e-2 |
> | Inference sample size 10 | 7.04e-3 | 2.97e-4 | **2.79e-4** | 4.16e-3 | 1.26e-2 | 3.54e-2 |
> | Inference sample size 20 | 3.63e-3 | 4.01e-5 | **6.67e-6** | 1.80e-4 | 1.23e-3 | 8.02e-3 |
> | Inference sample size 50 | 1.41e-3 | 7.70e-6 | **1.44e-6** | 1.58e-6 | 1.89e-5 | 1.89e-4 |
> | Inference sample size 100 | 7.04e-4 | 2.81e-6 | 3.23e-7 | **3.82e-8** | 5.31e-6 | 1.04e-5 |
> | Inference sample size 200 | 3.53e-4 | 1.13e-6 | 3.78e-8 | **1.48e-8** | 2.83e-6 | 5.12e-6 |
>
> **Table H.** Variance of estimated loss L(ϕ(x), y). Lowest variance for each inference sample size is boldfaced.
>
> | Train sample size | 1 | 2 | 5 | 10 | 20 | 50 |
> | --- | --- | --- | --- | --- | --- | --- |
> | Inference sample size 1 | **2.01e-1** | 6.97e-1 | 6.21 | 2.51e+01 | 7.18e+1 | 3.16e+2 |
> | Inference sample size 2 | 1.09e-1 | **1.03e-1** | 1.21 | 7.98 | 2.36e+1 | 1.35e+2 |
> | Inference sample size 5 | 3.65e-2 | **4.72e-3** | 5.24e-2 | 7.07e-1 | 3.48 | 3.05e+1 |
> | Inference sample size 10 | 1.67e-2 | **4.23e-4** | 6.49e-3 | 5.73e-2 | 4.22e-1 | 6.74 |
> | Inference sample size 20 | 7.69e-3 | **4.37e-5** | 1.38e-3 | 7.43e-4 | 4.17e-2 | 7.33e-1 |
> | Inference sample size 50 | 2.74e-3 | 8.07e-6 | 7.78e-4 | **2.76e-6** | 3.27e-3 | 1.17e-2 |
> | Inference sample size 100 | 1.30e-3 | 2.91e-6 | 3.30e-4 | **3.87e-8** | 1.48e-3 | 2.52e-3 |
> | Inference sample size 200 | 6.39e-4 | 1.16e-6 | 2.14e-4 | **1.50e-8** | 1.13e-3 | 1.26e-3 |
>
> **Table I.** Mean of estimated loss L(ϕ(x), y). Best model for each inference sample size is boldfaced.
>
> | Train sample size | 1 | 2 | 5 | 10 | 20 | 50 |
> | --- | --- | --- | --- | --- | --- | --- |
> | Inference sample size 1 | 0.492 | **0.276** | 0.852 | 2.23 | 3.48 | 7.21 |
> | Inference sample size 2 | 0.406 | **9.76e-2** | 0.363 | 1.04 | 1.99 | 5.15 |
> | Inference sample size 5 | 0.329 | **2.28e-2** | 9.22e-2 | 0.220 | 0.564 | 2.10 |
> | Inference sample size 10 | 0.298 | **9.54e-3** | 4.99e-2 | 4.49e-2 | 0.180 | 0.791 |
> | Inference sample size 20 | 0.281 | **5.43e-3** | 4.70e-2 | 4.34e-3 | 8.48e-2 | 0.211 |
> | Inference sample size 50 | 0.269 | 4.09e-3 | 4.13e-2 | **4.36e-4** | 4.44e-2 | 4.49e-2 |
> | Inference sample size 100 | 0.264 | 3.72e-3 | 4.24e-2 | **1.73e-4** | 4.22e-2 | 3.53e-2 |
> | Inference sample size 200 | 0.263 | 3.49e-3 | 4.03e-2 | **1.43e-4** | 4.29e-2 | 3.04e-2 |
>
> Observation 1: All models successfully achieve > 95% test accuracy when evaluated with 10 samples.
>
> Observation 2: Models trained with smaller sample sizes take more training iterations to converge.
>
> Observation 3: Models trained with smaller sample sizes tend to be more sample efficient. Their distribution p(g|x) tend to learn more low-variance permutations (**Table F**), and the models tend to learn more low-variance estimation of output ϕ(x) and loss L(ϕ(x), y) (**Tables G and H**). This indicates that small sample size serves as a regularizer that encourages lower variance of the estimator and thus a better sample efficiency. Interestingly, for training sample size 1, the task loss is high compared to all other models and sample sizes (**Table I**) while variance for inference sample size 1 is the lowest (**Tables G and H**). This suggests that training sample size 1 over-regularizes towards low-variance estimation.
>
> Observation 4: Unlike training, larger sample size for inference consistently benefits all models (**Tables G, H, and I**).
>
> **A4. In EXP-classify, the sample size for training introduces an important tradeoff. A small sample size takes more training iterations to converge, but serves as a regularizer that encourages lower variance of the estimator and thus a better sample efficiency. On the other hand, larger sample sizes are consistently better for inference.**

---

> > ### Author Response · Authors · 2023-08-18
> > **We are happy to provide more discussion and details**
> >
> > Again, we appreciate all reviewers for positive and constructive feedbacks. We believe that the new findings, especially the analysis of the group distribution variance and sample complexity, will make our paper more solid. We are happy to provide more discussion and details on the analysis upon request. We will include all the above results in a more organized form (plots and visualizations) in the final version of the paper.

---

### Decision · Program_Chairs · 2023-09-21

**Decision:**

Accept (spotlight)

**Comment:**

Summary

This work presents an approach to make deep learning models equivariant to a given group. The experiments of this paper on patch based transformer show competitive performance on a variety of equivariant groups such as permutation and rotation. This is valuable given the interest in using transformer based architectures for a variety of recognition tasks. Moreover, the paper also shows applicability of the method on particle dynamics and graphs.

Reviews & Justification

This paper received positive reviews and all the reviewers were unanimous in recognizing its acceptance. The wide applicability of this method across architectures, and domains (vision, particle dynamics, graphs) makes it interesting to a wider audience. The AC also appreciates the in-depth author responses and additional experiments to further bolster the reviewer’s confidence. Please add in the related work suggested by the reviewers.